

# Twisted holography on $AdS_3 \times S^3 \times K3$ & the planar chiral algebra

Víctor E. Fernández[*], Natalie M. Paquette[†] and Brian R. Williams[‡]

Department of Physics, University of Washington, Seattle, USA
Department of Mathematics, Boston University, Boston, USA

[*] vfer@uw.edu , [†] npaquett@uw.edu , [‡] brianwilliams.math@gmail.com

## Abstract

In this work, we revisit and elaborate on twisted holography for $AdS_3 \times S^3 \times X$ with $X = T^4$, K3, with a particular focus on K3. We describe the twist of supergravity, identify the corresponding (generalization of) BCOV theory, and enumerate twisted supergravity states. We use this knowledge, and the technique of Koszul duality, to obtain the $N \to \infty$, or planar, limit of the chiral algebra of the dual CFT. The resulting symmetries are strong enough to fix planar 2 and 3-point functions in the twisted theory or, equivalently, in a 1/4-BPS subsector of the original duality. This technique can in principle be used to compute corrections to the chiral algebra perturbatively in $1/N$.

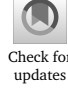

# 1 Introduction & summary

Twisted holography [1–3] is a proposal to access protected quantities on both sides of a holographic duality. While twists of field theory have been studied for a long time, and correspond to restricting to the cohomology of a chosen supercharge, twisting a supergravity or (spacetime) string theory involves turning on a background vev for the bosonic ghost associated to the corresponding supertranslation [1]. More precisely, given a noncompact Calabi-Yau fivefold with asymptotic boundaries, we can specify a vacuum by prescribing asymptotic values of the fields (mathematically, choose an augmentation of the factorization algebra). The corresponding vev of, in particular, the superghost provides a solution to its equations of motion, which in the BV formalism is a solution to the Maurer-Cartan equation. The equation of motion for the superghost tells us that it must be a covariantly constant spinor of square zero. We think of the twist as deforming the field equations by the resulting Maurer-Cartan element, which means (in perturbation theory) studying fluctuations around the resulting field configuration. In the context of AdS/CFT, a choice of twist in the boundary CFT does not uniquely determine a twist in the gravitational theory, but rather gives a boundary value problem to solve for possible covariantly constant spinors in the bulk theory. Working perturbatively around any of these solutions gives different "twists" of the supergravity theory, and in this work we choose one such saddle, corresponding to the twist of empty $\mathrm{AdS}_3$, and work in perturbation theory around it.

In the physical theory, of course, what happens in the interior is determined dynamically after specifying the asymptotic boundary conditions; it is a fascinating open question to understand how one can move beyond working around a given saddle in twisted SUGRA, and, for instance, understand the localized path integral as a suitable sum over twisted gravitational saddles.[1] Many choices of twists are possible, corresponding to the family of nilpotent supercharges available in the supersymmetry or superconformal algebra. One interesting, and relatively accessible, class of twists are those which endow the surviving local operators with the structure of a chiral algebra. In four real dimensions, such a twist has been an area of active recent inquiry [4] and was applied to the twisted holography of 4d $\mathcal{N} = 4$ super Yang-Mills in [2]. In two real dimensions, such a twist is simply the half-twist [5,6], and does not change the effective dimensionality of the twisted field theory or its bulk dual. We will explore this relatively simple twist in the context of (top-down models of) $AdS_3/CFT_2$, in particular $AdS_3 \times S^3 \times K3$. Many similar results for the case when K3 is replaced by $T^4$ have already appeared in the companion paper [3]. It is important to note, however, that the half-twisted theory (equivalently, the minimal, holomorphic twist in two dimensions) is sensitive to nonperturbative corrections, such as worldsheet instantons, which makes studying a global description of the twist of the SCFT on K3 from first principles somewhat challenging. The mathematical version of this statement is that the chiral de Rham complex of a nontrivial compact manifold is given locally as a sheaf of free vertex algebras, but gluing these local descriptions together is not easy. Although we will derive such a local description of the twist on the field theory side, we emphasize a way to circumvent the global challenge: given the description of a twisted supergravity theory, one may apply the operation of Koszul duality to obtain the chiral algebra of the dual field theory. In particular, the global subalgebra of the chiral algebra, which can also be deduced by considering vacuum-preserving diffeomorphisms of the bulk geometry, appears in this construction. That the mathematical operation of Koszul duality may govern part of the holographic dictionary in twisted holography was first suggested in [7] and successfully applied to AdS/CFT in [3]. For a review of Koszul duality and further citations, see [8]. The plan of this paper is as follows. In §2 we will give our description of the holomorphic twist of IIB supergravity in six dimensions (upon compactification on K3).[2] We describe how our twisted action can be obtained by integrating out the vev of a bosonic superghost. We then derive the backreacted geometry in the presence of the twisted D1-D5 system. In §3, we enumerate the states in twisted supergravity and reproduce the elliptic genus computation of [10,11] in this language. In §4, we review the computation of the $N \to \infty$ elliptic genus from the orbifold SCFT $Sym^N(K3)$ and its matching with the supergravity computation, and twist a local model of the B-brane D1-D5 brane system. This twist recovers the expected description of the chiral de Rham complex of $Sym^N(\mathbb{C}^2)$ (i.e. the half-twist of the symmetric orbifold SCFT) in the infrared. The Loday-Quillen-Tsygan theorem, which is a natural tool in the large-$N$ limit of twisted holography (e.g. [12], [13]), gives equivalent results in the $N \to \infty$ limit to this local model of the twist, but has not yet been developed mathematically for the global K3 geometry. Consequently, in §5 and §6 we turn our attention to the determination of the planar chiral algebra of the dual field theory from Koszul duality, first studying the chiral algebra Koszul

---

[1]In a similar vein, if one is only interested in perturbative analyses, the twists of [1] can be studied on compact CY5s, such as $T^{10}$. In this case, we have no choice of asymptotic vacuum and there is typically a unique solution to the BPS equations, so one can formally twist by studying the fluctuations around this solution. In the language of factorization algebras, the cohomology of global sections of the factorization algebra to the ground field is typically one-dimensional on a compact background.

[2]It would also be interesting to study twisted holography for $AdS_3 \times S^3 \times S^3 \times S^1$; see [9] for precise conjectures about the form of the duality and additional references. In a twisted background, the relevant geometry is a deformation of $(\mathbb{C}^3 \smile \mathbb{C}) \times Y$ where $Y$ is a Hopf surface that is diffeomorphic to $S^3 \times S^1$. Since Hopf surfaces are not Kähler, to carefully study this twisted background would require more care than our analysis here. Given a satisfactory formulation of BCOV on Hopf surfaces, it would be of interest to characterize the universal chiral algebra for D-branes in this twisted compactification.

dual of twisted IIB supergravity on $\mathbf{C}^2 \times$ K3 and then incorporating the effects of the D-brane backreaction using a perturbative Feynman diagrammatic approach; while incorporating the effects of backreaction perturbatively from flat space would normally involve the summation of an infinite number of diagrams, the problem simplifies dramatically in twisted holography. There are a finite number of nonzero diagrams at each order in $N$ [3], and only 3 in the planar limit. We also comment on the global subalgebras of the chiral algebras dual to the symmetries of the flat space and backreacted (i.e. holographic) geometries, respectively.

## 2 Twisted supergravity in six dimensions

The compactification of type IIB supergravity on a Calabi–Yau surface results in a supergravity theory which enjoys $\mathcal{N} = (2,0)$ supersymmetry. We concern ourselves with a simplification obtained by twisting the original type IIB supergravity with respect to a particular ten-dimensional supercharge. This supercharge is such that the resulting compactified theory is holomorphic in the sense that it only depends on the complex structure of the six-dimensional spacetime.

As found in [3], in the case that the complex surface is $T^4$, this holomorphic theory is an extended version of the famous Kodaira–Spencer theory introduced in [14] to describe the closed string field theory of the $B$-model on a Calabi–Yau threefold. In this paper we mostly consider the case where the surface we compactify along is a $K3$ surface, referring to [3] for details in the case where the surface is $T^4$. This section outlines the description of this extension of Kodaira–Spencer theory. More generally, we comment on a similar extension of Kodaira–Spencer theory which depends on the data of a commutative super ring equipped with a trace (in the physically meaningful cases, this ring corresponds to the graded cohomology ring of either $K3$ or $T^4$ and trace is integration).

We recall some generalities on twisting supergravity following the foundational work in [1]. In any supergravity theory there are ghosts for both local diffeomorphisms and local supersymmetry. Ghosts for local supersymmetries are bosonic ghosts and are typically realized as sections of a spinor bundle over spacetime. Twisted supergravity is simply supergravity in a background where a particular bosonic ghost for local supersymmetry acquires a nonzero expectation value $Q$. In addition to being part of a consistent background for supergravity, $Q$ must satisfy the Maurer–Cartan equation $[Q, Q] = 0$, where $[-, -]$ is supercommutator in the local supersymmetry algebra. In this sense, for deformations of flat space, the classification of twisting supercharges for supergravity is closely related to twists of ordinary field theories.

The ten-dimensional IIB supersymmetry algebra admits a range of such twisting supercharges. We concern ourselves with a so called 'minimal' (or holomorphic) twisting supercharge $Q$ which has the property that it is stabilized by $SU(5)$ in the Lorentz group $Spin(10)$. Such twists exist whenever the ten-dimensional spacetime is a Calabi–Yau manifold of dimension five. In [1] a conjecture for this twist is given as a certain limit of the string field theory obtained from the topological $B$-model on the Calabi–Yau fivefold. The free limit of this conjecture has been proven in [15].

We remark on a caveat involving this conjecture. First, the topological $B$-model has critical complex dimension three, meaning that genus $g$ amplitudes are nonzero only when the dimension of the Calabi–Yau target manifold is three. On the other hand, there is no $U(1)$ factor of the $R$-symmetry in the ten-dimensional IIB super Poincaré group which is compatible with the choice of a holomorphic supercharge $Q$. These issues are related. On one hand, while there are no nonzero amplitudes for insertions of operators of total ghost number zero, there are nonzero amplitudes involving operators of nonzero ghost number (here we mean ghost number computed from the worldsheet perspective). On the other, the fact that there is no

$U(1)$ within the $R$-symmetry that is compatible with $Q$ means that the fields in the resulting twisted theory do not have a consistent spacetime ghost number, but only a ghost number modulo 2. These two observations are consistent with the fact that Kodaira–Spencer theory defined on a Calabi–Yau manifold of dimension different from three is a theory with ghost number grading by the group $\mathbf{Z}/2$, rather than the typical integral grading. One can think of this $\mathbf{Z}/2$ as fermion parity, so there is no longer an invariant distinction between ghosts and ordinary fermions in this theory. We will observe, nevertheless, that upon compactification of this ten-dimensional Kodaira–Spencer theory to six-dimensions that we are able to lift this $\mathbf{Z}/2$ grading to a fairly natural integral one (but of course this choice is not unique).

## 2.1 Kodaira–Spencer theory and IIB supergravity

We turn to a recollection of the conjectural holomorphic twist of type IIB supergravity in ten dimensions as originally described in [1]. Our discussion largely follows [3]. We refer to these references for more details.

The holomorphic supercharge used to minimally twist supergravity is invariant under $SU(5) \subset Spin(10)$, and so can be defined on any Calabi–Yau fivefold $X$. In [1], as we just recalled, it was conjectured that the holomorphic twist of IIB supergravity is equivalent to a certain truncation of the topological $B$-model on $X$.[3] We will assume this conjecture throughout the paper, and we will provide further justification in section 2.2.

The fields of Kodaira–Spencer theory on the Calabi–Yau fivefold $X$ are given in terms of the Dolbeault complex of polyvector fields on $X$; that is, sections of exterior powers of the holomorphic tangent bundle with values in $(0, \bullet)$ Dolbeault forms:

$$\mathrm{PV}^{i,j}(X) = \Omega^{0,j}(X, \wedge^i TX). \tag{1}$$

In local holomorphic coordinates $z_1, \ldots, z_5$ such a polyvector field can be expressed as

$$\mu = \mu^{\bar{i}_1 \cdots \bar{i}_5}_{j_1 \cdots j_5} \mathrm{d}\bar{z}_{\bar{i}_1} \cdots \mathrm{d}\bar{z}_{\bar{i}_5} \partial_{z_{j_1}} \cdots \partial_{z_{j_5}}.[4] \tag{2}$$

It is convenient to express polyvector fields in terms of a single superfield. To do this, we rename $\mathrm{d}\bar{z}_{\bar{i}}$ as $\overline{\theta}_{\bar{i}}$ and $\partial_{z_j}$ as $\theta^j$. Bear in mind that $\theta$ transforms as a holomorphic vector while $\overline{\theta}$ transforms as an anti-holomorphic covector. With this notation, a general polyvector field

$$\mu \in \mathrm{PV}(X) = \oplus_{i,j} \mathrm{PV}^{i,j}(X), \tag{3}$$

can be thought of as a smooth function

$$\mu = \mu(z_i, \bar{z}_{\bar{i}}, \theta^i, \overline{\theta}_{\bar{i}}), \tag{4}$$

on the superspace $\mathbf{C}^{5|5+5}$ where the odd cordinates are $\theta^i, \overline{\theta}_{\bar{i}}$ for $i, \bar{i} = 1, \ldots, 5$.

The space of fields of Kodaira–Spencer theory is not all polyvector fields: rather, the fields are polyvector fields which satisfy the constraint that they are divergence-free with respect to the holomorphic volume form $\Omega$. Geometrically, this means that $L_\mu \Omega = 0$ where $L_\mu$ is the Lie derivative;[5] equivalently this is the condition $\partial \mu = 0$ where $\partial$ is the divergence operator. In coordinates this reads

$$\partial = \sum_i \partial_{\theta^i} \partial_{z_i}. \tag{5}$$

---

[3]This truncation was referred to as 'minimal' Kodaira–Spencer theory in *loc. cit.*. It effectively throws out the non-propagating fields.

[4]We will always omit the wedge product symbol $\wedge$.

[5]We recall that the Lie bracket on polyvector fields is the Schouten bracket, which reduces to the usual Lie bracket on ordinary vector fields.

In addition to $\partial \mu = 0$ we also require that

$$\partial_{\theta^1} \cdots \partial_{\theta^5} \mu = 0 \,, \tag{6}$$

which effectively throws away the top power of $T_X$. We will justify this additional condition shortly.

To define the action functional we utilize an integration map

$$\int_X^{\Omega} : \mathrm{PV}^{5,5}(X) \simeq C^{\infty}(X)\theta^1 \cdots \theta^5 \overline{\theta}_1 \cdots \overline{\theta}_5 \to \mathbf{C} \,, \tag{7}$$

which is $\int (\mu \vee \Omega) \wedge \Omega$, with $\Omega$ the Calabi–Yau form. This operation simply projects out the $\mathrm{PV}^{5,5}$ component of the Kodaira–Spencer field, to get $(0,5)$ form, then integrates this against the holomorphic volume form. In terms of the superspace description this is the usual integration along $X$ together with the Berezinian integral along the odd directions.

A typical feature of Kodaira–Spencer theory, formulated naively, is that the kinetic part of the Lagrangian contains a non-local expression involving the distributional inverse of the divergence operator $\partial$. While this is not globally well-defined, the condition that $\mu$ be in the kernel of $\partial$ ensures that there exists locally such a polyvector field.

In summary, the fields of Kodaira–Spencer theory are

$$\mathrm{PV}(X) \cap \ker \partial \,. \tag{8}$$

The Lagrangian is

$$\frac{1}{2} \int_X^{\Omega} \mu \overline{\partial} \partial^{-1} \mu + \frac{1}{6} \int_X^{\Omega} \mu^3 \,. \tag{9}$$

The conjecture originally put forth in [1] is that this Lagrangian captures the supersymmetric sector of IIB supergravity as described above. The superfield $\mu$ captures all the original fields, anti-fields, ghosts, etc. of type IIB supergravity after integrating out those fields which become massive in the holomorphic twist. Since the field $\mu$ includes anti-fields and anti-ghosts, we can describe the BV anti-bracket in this notation. The BV anti-bracket of two super-fields is

$$\{\mu(z,\overline{z},\theta,\overline{\theta}), \mu(w,\overline{w},\eta,\overline{\eta})\} = \partial_{z_i} \partial_{\theta^i} \delta(z-w)\delta(\overline{z}-\overline{w})(\overline{\theta}-\overline{\eta})(\theta-\eta)\mathrm{Id} \,. \tag{10}$$

The appearance of the holomorphic derivative $\partial_{z_i}$ in the expression above is one way to understand the appearance of the non-local kinetic term in the Lagrangian.

From this BV anti-bracket it is clear that the component of the super-field $\mu$ proportional to the top polyvector $\partial_{\theta^1} \cdots \partial_{\theta^5}$ does not propagate. It is therefore convenient to impose the additional constraint

$$\partial_{\theta^1} \cdots \partial_{\theta^5} \mu = 0 \,, \tag{11}$$

on the fields of Kodaira–Spencer theory, as mentioned earlier.

We can avoid part of the non-locality appearing in the action by introducing a field $\widehat{\mu}_{i_1 \cdots i_4} \in \mathrm{PV}^{4,\bullet}$ which satisfies

$$(\partial \widehat{\mu})^{\bullet}_{i_1 i_2 i_3} = \mu^{\bullet}_{i_1 i_2 i_3} \,, \tag{12}$$

where the bullet denotes arbitrary anti-holomorphic form type. We can do this because we have the constraint $\partial \mu = 0$. Then, the kinetic term in the Lagrangian above can be written as

$$\int \epsilon^{i_1 \cdots i_5} \epsilon_{\overline{j}_1 \cdots \overline{j}_5} \mu_{i_1} \overline{\partial} \widehat{\mu}_{i_2 \cdots i_5} + \frac{1}{2} \int \epsilon^{i_1 \cdots i_5} \mu_{i_1 i_2} (\overline{\partial} \partial^{-1} \mu)_{i_3 i_4 i_5} \,. \tag{13}$$

This Lagrangian is still non-local, but the only non-locality involves the field $\mathrm{PV}^{2,\bullet}(X)$. We will see the significance of this field from the perspective of supergravity in the next subsection.

## 2.2 Matching supergravity with Kodaira–Spencer theory

At the level of free fields, the match between the holomorphic twist of type IIB supergravity on $\mathbf{R}^{10} = \mathbf{C}^5$ and Kodaira–Spencer theory has been performed in [15]. Here, we spell out a precise relationship between the fields of Kodaira–Spencer theory and those of supergravity, to illustrate how Kodaira-Spencer theory encodes (the twist of) the physical field content. For clarity of presentation we will work on flat space near the flat Kähler metric $g_0^{i\bar{j}} = \delta^{i\bar{j}}$.

The most important bosonic field in supergravity is, of course, the metric tensor. As representations of $SU(5)$, the metric tensor breaks into three components: $g^{ij}, g^{i\bar{j}}, g^{\bar{i}\bar{j}}$. To leading order, the components $g^{ij}, g^{i\bar{j}}$ are rendered massive in the twist and can hence be removed. The remaining component of the metric corresponds to the field $\mu_k^{\bar{j}}$ in Kodaira–Spencer theory via the Kähler form

$$g^{\bar{i}\bar{j}} \mapsto \delta^{k\bar{i}} \mu_k^{\bar{j}}. \tag{14}$$

The fermionic fields of type IIB supergravity include a gravitino. In the untwisted theory the gravitino has a spinor index and a vector index. As an $SU(5)$ representation, the 16-dimensional spinor representation $S_+$ of $SO(10)$ decomposes as a sum of three irreducible representations: the trivial representation, the exterior square of the anti-fundamental representation, and the fourth exterior power of the anti-fundamental representation:

$$S_+ \simeq_{SU(5)} \mathbf{C} \oplus \wedge^2 \overline{\mathbf{C}}^5 \oplus \wedge^4 \overline{\mathbf{C}}^5. \tag{15}$$

The component which survives the twist is the holomorphic vector valued in the exterior square in the above equation, and we denote this field by

$$\lambda_i^{\bar{j}_1\bar{j}_2}, \tag{16}$$

which we can view as an element $\mathrm{PV}^{1,2}(\mathbf{C}^5)$.

The antifield to the component of the gravitino $\lambda_i^{\bar{j}_1\bar{j}_2}$ is a tensor of the form $\lambda^{*k}_{\bar{l}_1\bar{l}_2}$, where the $*$ just indicates that this is an anti-field in the physical theory. Since the gravitino is an odd field, its anti-field has overall even parity. It turns out that it is the derivative of this anti-field that corresponds to a field of Kodaira–Spencer theory

$$\partial_{z_{k_1}} \lambda^{*k_2}_{\bar{l}_1\bar{l}_2} \mapsto \epsilon^{k_1 k_2 i_1 i_2 i_3} \epsilon_{\bar{l}_1\bar{l}_2\bar{j}_1\bar{j}_2\bar{j}_3} \mu_{i_1 i_2 i_3}^{\bar{j}_1\bar{j}_2\bar{j}_3}. \tag{17}$$

That is, we view the derivative of the anti-field as an element of $\mathrm{PV}^{3,3}$. Following the discussion above, we can use the equation $\partial \mu = 0$ to replace the field $\mu_{i_1 i_2 i_3}^{\bar{j}_1\bar{j}_2\bar{j}_3}$ by a field $\widehat{\mu}$ satisfying

$$\mu_{i_1 i_2 i_3}^{\bar{j}_1\bar{j}_2\bar{j}_3} = \partial_{z_j} \widehat{\mu}_{j i_1 i_2 i_3}^{\bar{j}_1\bar{j}_2\bar{j}_3}. \tag{18}$$

Note that $\widehat{\mu}_{j i_1 i_2 i_3}^{\bar{j}_1\bar{j}_2\bar{j}_3}$ is a field of type $\mathrm{PV}^{4,3}$. Using this modified field in Kodaira–Spencer theory, we can more easily match with the anti-gravitino via

$$\lambda^{*k}_{\bar{l}_1\bar{l}_2} \mapsto \epsilon^{k i_1 i_2 i_3 i_4} \epsilon_{\bar{l}_1\bar{l}_2\bar{j}_1\bar{j}_2\bar{j}_3} \mu_{k i_1 i_2 i_4}^{\bar{j}_1\bar{j}_2\bar{j}_3}. \tag{19}$$

Next, let us explicitly match the holomorphic twist of type IIB supergravity with Kodaira–Spencer theory at the level of the kinetic term in the Lagrangian. In (13) we have expressed the kinetic term in the Kodaira–Spencer action as a sum of two terms. We first show how there is a similar kinetic term involving the metric $g$ and the anti-field to the gravitino $\lambda^*$ when we twist type IIB supergravity.

Recall that the holomorphic twist amounts to assigning a certain component of the superghost a nontrivial VEV. As an $SU(5)$ representation the superghost $Q$ can be written as a sum of three tensors $Q^{(0)}, Q^{\bar{j}_1 \bar{j}_2}, Q^{\bar{j}_1 \cdots \bar{j}_4}$, which are the components of the even exterior powers of the anti-fundamental representation of $SU(5)$. Here $Q^{(0)}$ denotes the $SU(5)$ invariant component of the superghost in the $\mathcal{N} = (1, 0)$ subalgebra; this is the component in which the holomorphic supercharge lives. A term in the BV action involving $\lambda^*$ and $Q$ arises from a supersymmetric variation of the gravitino $\lambda$.

Reverting back to $SO(10)$ notation, where $a, b = 1, \ldots, 10$ are vector indices and $\alpha, \beta, \ldots = 1, \ldots, 32$ are spinor indices, the supersymmetric variation of the gravitino is of the form

$$\delta \lambda_a^\alpha = \delta_{ab}(\partial_{x_b} \epsilon^\alpha + A_\beta^{\alpha b}(g) \epsilon^\beta). \tag{20}$$

Here $A$ is the spin Levi-Civita tensor in the spin representation of $Spin(10)$.[6] Taking a perturbative expansion of the flat metric of the form $\delta^{ab} + g^{ab}$ and working to low order in $g^{ab}$, we can write the ordinary Levi-Civita connection as

$$A_a^{bc} = \frac{1}{2} \delta_{ad}(\partial_{x_c} g^{bd} + \partial_b g^{cd} - \partial_d g^{bc}) + O(g^2). \tag{21}$$

In terms of this ordinary Levi-Civita connection, the spin Levi-Civita connection can be written, employing the usual $\Gamma$-matrices, as

$$A_\beta^{\alpha b} = \Gamma_c^{\alpha \gamma} \Gamma_{\beta \gamma}^a A_a^{bc}. \tag{22}$$

We are interested in the covariant derivative of the constant spinor $\epsilon^{(0)}$.

As before, a spinor decomposes, as an $SU(5)$ representation, into a sum of even exterior powers of the anti-fundamental representation. The index $(0)$ represents the $SU(5)$ invariant part of the spinor. A simple computation with $\Gamma$-matrices shows that the components of the spin Levi-Civita connection whose lower index is $(0)$ and upper index is $(\overline{ij})$ are

$$A_{(0)}^{(\overline{ij})k} = A_j^{\bar{i}k} \delta^{j\bar{j}},$$
$$A_{(0)}^{(\overline{ij})\bar{k}} = A_j^{\overline{ik}} \delta^{j\bar{j}},$$

where the ordinary Christoffel symbols appear on the right hand side (with $SU(5)$ indices).

Plugging in (22) we see that the desired variation of the gravitino is

$$\begin{aligned}
\delta \lambda_k^{\overline{ij}} &= \delta_{k\bar{k}} A_{(0)}^{(\overline{ij})\bar{k}} \epsilon^{(0)} \\
&= \delta_{k\bar{k}} \delta^{j\bar{j}} A_j^{\overline{ik}} \epsilon^{(0)} \\
&= \frac{1}{2} \delta_{k\bar{k}} \delta^{j\bar{j}} \delta_{j\bar{l}} \left( \delta_{\bar{z}_{\bar{k}}} g^{\overline{li}} + \partial_{\bar{z}_{\bar{i}}} g^{\overline{lk}} - \partial_{\bar{z}_{\bar{l}}} g^{\overline{ik}} \right) \epsilon^{(0)} \\
&= \frac{1}{2} \delta_{k\bar{k}} \left( \delta_{\bar{z}_{\bar{k}}} g^{\overline{ji}} + \partial_{\bar{z}_{\bar{i}}} g^{\overline{jk}} - \partial_{\bar{z}_{\bar{j}}} g^{\overline{ik}} \right) \epsilon^{(0)} \\
&= \epsilon^{\overline{ij}} \delta_{k\bar{k}} \partial_{\bar{z}_{\bar{i}}} g^{\overline{jk}} \epsilon^{(0)}.
\end{aligned}$$

In the last line we have used the fact that $\bar{i}, \bar{j}$ appear anti-symmetrically on the left hand side. It follows that once we assign a nonzero VEV to the superghost $Q^{(0)}$ in the BV action there is a term of the form

$$(\partial_{\bar{z}_{\bar{k}}} g^{\overline{ij}} \delta_{l\bar{i}}) \lambda_{\overline{kj}}^{*l}. \tag{23}$$

---

[6]We use $A$ instead of $\Gamma$ for the Levi-Civita connection to avoid confusion with $\Gamma$-matrices.

This matches precisely with the first term in the Kodaira–Spencer kinetic action.

The final fields we describe in terms of the holomorphic twist are the Ramond–Ramond fields in supergravity. These fields are sourced by $D(2k-1)$-branes and are forms of degree $8-2k$. In the original presentation of Kodaira–Spencer theory, certain components of the field strengths of such forms are present as polyvector fields. The field strength is a form of degree $9-2k$; in the holomorphic twist the component of this form which survives is of Hodge type $(5-k, 4-k)$ and corresponds to polyvector field of type $(k, 4-k)$ using the isomorphism

$$\mathrm{PV}^{k,4-k}(\mathbf{C}^5) \simeq_{\Omega} \Omega^{5-k,4-k}(\mathbf{C}^5) \subset \Omega^{9-2k}(\mathbf{R}^{10}) \otimes \mathbf{C}, \tag{24}$$

determined by the Calabi–Yau form.

A special Ramond–Ramond form is the four-form $C \in \Omega^4(\mathbf{R}^{10})$ sourced by a $D3$-brane. Such a field is required to be 'chiral' in the sense that its field strength $F = \mathrm{d}C$ is self-dual. The component of the field strength

$$F^{\bar{i}_1\bar{i}_2 j_1 j_2 j_3} \in \Omega^{3,2}(\mathbf{C}^5), \tag{25}$$

survives the holomorphic twist. Using the holomorphic volume form, these components are identified with the fields

$$F^{\bar{i}_1\bar{i}_2 j_1 j_2 j_3} \mapsto \epsilon^{j_1 j_2 j_3 j_4 j_5} \mu^{\bar{i}_1\bar{i}_2}_{j_4 j_5}, \tag{26}$$

which is a polyvector field of type $(2, 2)$. Self-duality becomes the constraint $\partial_j \mu^{\bar{i}_1\bar{i}_2}_{jk} = 0$ that this polyvector field be divergence-free. This constraint gives rise to the non-local kinetic term present in equation (13). For more on the relationship between constraints and non-local kinetic terms we refer to [16].

This concludes our general discussion of the twist of ten-dimensional type IIB supergravity in terms of Kodaira–Spencer theory. We now turn to compactifications as understood in the twist.

## 2.3 Compactification of Kodaira–Spencer theory

We will focus on the setting where we compactify Kodaira-Spencer theory on a complex surface. This section largely follows [3], which analyzed the compactification of Kodaira-Spencer theory on $T^4$ (but actually can be extended to any compact holomorphic symplectic surface with no difficulty), and the subsequent backreaction computation in the twisted D1-D5 system. Many of the computations easily generalize when the $T^4$ is replaced by $K3$.

Let $Y$ be a complex surface (which we will soon take to be compact) with a fixed holomorphic symplectic structure. A general field of Kodaira–Spencer theory on $\mathbf{C}^3 \times Y$ is a Dolbeault-valued polyvector field which is annihilated by the divergence operator with respect to the holomorphic volume form. We will use coordinates $z, w_1, w_2$ on $\mathbf{C}^3$ and we fix the standard Calabi–Yau form $\Omega = \mathrm{d}z\mathrm{d}w_1\mathrm{d}w_2$.

A Dolbeault-valued polyvector field $\alpha^{k,\bullet}$ on $\mathbf{C}^3 \times Y$ of type $(k, \bullet)$ can be written as a tensor product of one on $\mathbf{C}^3$ with one on $Y$

$$\alpha^{k,\bullet} = \sum_{i+j=k} \beta^{i,\bullet} \otimes \gamma^{j,\bullet}, \tag{27}$$

where $\beta^{i,\bullet}, \gamma^{j,\bullet}$ are polyvector fields of type $(i, \bullet), (j, \bullet)$ on $\mathbf{C}^3, Y$ respectively. Polyvector fields on $Y$ are the same as differential forms, because the holomorphic symplectic form on $Y$ identifies the tangent and cotangent bundles. In particular, the harmonic polyvector fields are given simply by the de Rham cohomology of $Y$. Furthermore, polyvector fields on $Y$ which are harmonic are automatically in the kernel of the divergence operator $\partial$, by standard Hodge theory arguments. To summarize, there is an equivalence of graded algebras

$$\mathrm{PV}(\mathbf{C}^3) \otimes \left( \ker \partial|_{\mathrm{PV}(Y)} \right) \simeq \mathrm{PV}(\mathbf{C}^3) \otimes H^\bullet(Y).$$

We will use this equivalence to describe the fields of the theory on $\mathbf{C}^3$ upon compactification along $Y$.

Let

$$R = H^{\bullet}(Y), \tag{28}$$

denote the cohomology ring of $Y$. We are interested in the case that $Y$ is a $K3$ surface, in which case this algebra is generated by even elements $\eta, \overline{\eta}, \eta_a$ for $a = 1, \ldots 20$ subject to the relations

$$\eta^2 = \bar{\eta}^2 = 0,$$
$$\eta_a \eta_b = h_{ab} \eta \bar{\eta}, \tag{29}$$

where $h_{ab}$ is a non-degenerate symmetric pairing on $\mathbf{C}^{20}$. Let $I$ denote the ideal generated by these equations so that $R = \mathbf{C}[\eta, \bar{\eta}, \eta_a]/I$.

As before, we write the polyvector fields on $\mathbf{C}^3$ in terms of a superspace by introducing odd variables $\theta^i$, $\overline{\theta}_{\bar{j}}$. Here, $\theta^i$ represents the coordinate vector field $\partial_{z_i}$ and $\overline{\theta}_{\bar{i}}$ represents the coordinate Dolbeault form $d\overline{z}_{\bar{i}}$. Then we can write the field content as a collection of superfields

$$\mu(z, \overline{z}, \theta^i, \overline{\theta}_{\bar{i}}, \eta) \in \oplus_{i,j} \mathrm{PV}^{i,j}(\mathbf{C}^3) \otimes R. \tag{30}$$

Here, we are using the shorthand $\eta$ to inform that there is a dependence on $\eta, \overline{\eta}$, and $\eta_a$, $a = 1, \ldots, 20$. As such, such a superfield decomposes in its dependencies on the generators of the cohomology of $Y$ as

$$\mu(z, \overline{z}, \theta^i, \overline{\theta}_{\bar{i}}) + \mu_{\eta}(z, \overline{z}, \theta^i, \overline{\theta}_{\bar{i}})\eta + \mu_{\overline{\eta}}(z, \overline{z}, \theta^i, \overline{\theta}_{\bar{i}})\overline{\eta} + \mu^a(z, \overline{z}, \theta^i, \overline{\theta}_{\bar{i}})\eta_a + \mu_{\eta\overline{\eta}}(z, \overline{z}, \theta^i, \overline{\theta}_{\bar{i}})\eta\bar{\eta}. \tag{31}$$

We emphasize that the $\eta$-variables represent harmonic polyvector fields on $Y$ and so are not acted on by any differential operators along $\mathbf{C}^3$.

The superfield satisfies the equation $\overline{\partial}\mu = 0$[7] where, in the superspace formulation,

$$\overline{\partial} = \overline{\theta}_{\bar{j}} \partial_{\overline{z}_{\bar{j}}}, \tag{32}$$

$$\partial = \partial_{\theta^i} \partial_{z_i}. \tag{33}$$

We denote by

$$\int_{\mathbf{C}^3}^{\Omega} (-)|_{\eta\overline{\eta}} \colon \mathrm{PV}^{3,3} \otimes R \to \eta\overline{\eta}\,\mathrm{PV}^{3,3} \to \mathbf{C}, \tag{34}$$

the projection onto the summand $\mathbf{C}\eta\overline{\eta} \subset R$ followed by integration as in (7). We emphasize that the notation $(-)|_{\eta\overline{\eta}}$ means we pick up only the $\eta\overline{\eta}$ component.

The Lagrangian is

$$\frac{1}{2}\int_{\mathbf{C}^3}^{\Omega} \mu \overline{\partial} \partial^{-1} \mu|_{\eta\overline{\eta}} + \frac{1}{6}\int_{\mathbf{C}^3} \mu^3|_{\eta\overline{\eta}}. \tag{35}$$

We can simplify the field content somewhat, following [2] which the authors in [1] refer to as minimal Kodaira–Spencer theory. We note that the coefficient of $\theta^1\theta^2\theta^3$ does not appear in the kinetic term in the action. This field does not propagate, so we can (and will) impose the additional constraint

$$\partial_{\theta^1}\partial_{\theta^2}\partial_{\theta^3}\mu(z, \overline{z}, \theta, \overline{\theta}, \eta) = 0. \tag{36}$$

This constraint only removes a single topological degree of freedom and hence will not significantly modify quantities like OPEs later on.

---

[7]For notational simplicity, we will no longer make manifest the dependence of the divergence operator on $\Omega$.

Next, let us expand the superfield $\mu$ only in the $\theta^i$ variables:

$$\mu = \mu(z, \overline{z}, \overline{\theta}, \eta) + \mu_i(z, \overline{z}, \overline{\theta}, \eta)\theta^i + \dots \tag{37}$$

We note that the constraint $\partial \mu_{ij} = 0$ implies that there is some super-field

$$\widehat{\mu}_{ijk}(z, \overline{z}, \overline{\theta}, \eta) = \alpha(z, \overline{z}, \overline{\theta}, \eta)\epsilon_{ijk}, \tag{38}$$

so that $\partial_{z_i}\widehat{\mu}_{ijk} = \mu_{jk}$. This is parallel to the maneuver that we made for Kodaira–Spencer theory on $\mathbf{C}^5$ as in (12) above.

It is convenient to rephrase the theory in terms of the field $\alpha(z, \overline{z}, \overline{\theta}, \eta)$, which has no holomorphic index. We will also change notation and let $\gamma(z, \overline{z}, \overline{\theta}, \eta)$ be the term with no $\theta^i$ dependence in the superfield $\mu(z, \overline{z}, \theta, \overline{\theta}, \eta)$.

In summary, we have the following fundamental superfields in the compactified theory on $\mathbf{C}^3$:

- $\mu_i(z, \overline{z}, \overline{\theta}, \eta)\theta^i$ which we identify with an element in the graded space

$$\mu \in \mathrm{PV}^{1,\bullet}(\mathbf{C}^3) \otimes R[1]. \tag{39}$$

- $\alpha(z, \overline{z}, \overline{\theta}, \eta)$ which we identify with an element of the graded space

$$\alpha \in \Omega^{0,\bullet}(\mathbf{C}^3) \otimes R. \tag{40}$$

- $\gamma(z, \overline{z}, \overline{\theta}, \eta)$ which we also identify with an element of the graded space

$$\gamma \in \Omega^{0,\bullet}(\mathbf{C}^3) \otimes R[2]. \tag{41}$$

We explain the cohomological shifts in the next paragraph. In terms of these fields, the Lagrangian is

$$\frac{1}{2}\int_{\mathbf{C}^3} \epsilon^{ijk}\overline{\partial}\mu_i(\partial^{-1}\mu)_{jk}\, \mathrm{d}^3z|_{\eta\overline{\eta}} + \int_{\mathbf{C}^3} \alpha\overline{\partial}\gamma\, \mathrm{d}^3z|_{\eta\overline{\eta}} + \frac{1}{6}\int_{\mathbf{C}^3} \epsilon_{ijk}\mu_i\mu_j\mu_k\, \mathrm{d}^3z|_{\eta\overline{\eta}} + \int_{\mathbf{C}^3} \alpha\mu_i\partial_{z_i}\gamma\, \mathrm{d}^3z|_{\eta\overline{\eta}}. \tag{42}$$

In this expression we project onto the component $\eta\overline{\eta}$ as before.

Just as when we twist a field theory, when we twist a supergravity theory the ghost number of the twisted theory is a mixture of the ghost number and a $U(1)_R$-charge of the original physical theory. To define a consistent ghost number, one can choose any $U(1)_R$ in the physical theory under which the supercharge has weight 1. In general, there are many ways to do this. It is convenient for us to make the following assignments of ghost number.

1. The fermionic variables $\eta_a$ have ghost number 0.

2. The anti-commuting variables $\overline{\theta}_i$ have ghost number 1.

3. The field $\alpha$ has ghost number zero.

4. The field $\mu$ has ghost number $-1$ (so that the $\overline{\theta}_i$ component has ghost number zero.

5. The field $\gamma$ has ghost number $-2$ (so that the $\overline{\theta}_i\overline{\theta}_j$ component of $\gamma$ has ghost number zero).

With these choices one can check that the action (42) is ghost number zero. Note that in the case $R = \mathbf{C}$ the choice of ghost numbers we take here is in agreement of the presentation of Kodaira–Spencer theory on $\mathbf{C}^3$ as in [2], who used this formulation to explore the chiral algebra subsector of 4d $\mathcal{N} = 4$ SYM and its twisted gravity dual.[8]

---

[8]See also [17] for the first exploration of the gravitational dual of the chiral algebra subsector of 4d $\mathcal{N} = 4$ SYM.

## 2.4 Compactification and twisted multiplets

In this section we comment on the content of twisted six-dimensional $\mathcal{N} = (2,0)$ supergravity in terms of standard six-dimensional $\mathcal{N} = (2,0)$ multiplets.

In six-dimensional $\mathcal{N} = (2,0)$ supersymmetry there are two multiplets which appear in compactifications from ten dimensions: (i) the graviton multiplet and (ii) the tensor (or chiral two-form [18]) multiplet (the latter being the same multiplet describing the twist of a single $M5$ brane in eleven-dimensional supergravity on $\mathbf{R}^{11}$). The holomorphic twists of these multiplets have been characterized in [15,16]. By virtue of their holomorphicity, each theory shares a linear gauge symmetry by the $\overline{\partial}$ operator, schematically of the form $\delta\Phi = \overline{\partial}\Phi$ and so in the free field descriptions below we will use Dolbeault complexes to label twists of the multiplets.

We recall the field content of each of the twisted six-dimensional multiplets, whose origin we will review in more detail below.

(i) The holomorphic twist of the the graviton multiplet has fundamental fields

$$(\mu, \rho, \widetilde{\alpha}) \in \left(\Pi\mathrm{PV}^{1,\bullet}(\mathbf{C}^3)[1] \cap \ker \partial\right)^{\oplus 3}, \tag{43}$$

as well as fields

$$(\widetilde{\gamma}^i, \widetilde{\beta}_j) \in \Omega^{0,\bullet}(\mathbf{C}^3)^{\oplus 2} \oplus \Omega^{0,\bullet}(\mathbf{C}^3)^{\oplus 2} \oplus \Omega^{0,\bullet}(\mathbf{C}^3)^{\oplus 2}, \tag{44}$$

where $i, j = 1, 2, 3$. In $\mathcal{N} = (1,0)$ language this is the holomorphic twist of a $\mathcal{N} = (1,0)$ graviton multiplet, three hypermultiplets, and a single $\mathcal{N} = (1,0)$ tensor multiplet.

(ii) The holomorphic twist of the $\mathcal{N} = (2,0)$ tensor multiplet has fields

$$\alpha \in \left(\Pi\Omega^{2,\bullet}(\mathbf{C}^3)[1]\right) \cap \ker \partial, \tag{45}$$

together with

$$(\gamma, \beta) \in \Omega^{0,\bullet}(\mathbf{C}^3)^{\oplus 2}. \tag{46}$$

In $\mathcal{N} = (1,0)$ language this is the holomorphic twist of a single hypermultiplet and a single $\mathcal{N} = (1,0)$ tensor multiplet.

We will see how these multiplets arise from compactification of our ansatz for the twist of type IIB supergravity on a $K3$ surface. Following the above presentation of Kodaira–Spencer theory we express the field content of the twist of type IIB supergravity on a Calabi–Yau fivefold $X$ as:

$$(\gamma_{IIB}, \beta_{IIB}) \in \mathrm{PV}^{0,\bullet}(X) \oplus \mathrm{PV}^{4,\bullet}(X) \cap \ker \partial,$$
$$(\mu_{IIB}, \rho_{IIB}) \in \Pi\mathrm{PV}^{1,\bullet}(X) \cap \ker \partial \oplus \Pi\mathrm{PV}^{3,\bullet}(X) \cap \ker \partial,$$
$$\alpha_{IIB} \in \mathrm{PV}^{2,\bullet}(X) \cap \ker \partial,$$

where $\Pi$ denotes parity shift.

On a fivefold of the form $X = \mathbf{C}^3 \times Y$ where $Y$ is a K3 surface, $\gamma_{IIB}$ decomposes as

$$\gamma_{IIB} = (\widetilde{\gamma}, \gamma_{0,2}) \in \Omega^{0,\bullet}(\mathbf{C}^3) \oplus \Omega^{0,\bullet}(\mathbf{C}^3) \otimes H^{0,2}(Y). \tag{47}$$

Up to topological degrees of freedom, $\beta_{IIB}$ decomposes also as

$$(\widetilde{\beta}, \beta_{2,0}) \in \Omega^{0,\bullet}(\mathbf{C}^3) \oplus \Omega^{0,\bullet}(\mathbf{C}^3) \otimes H^{2,0}(Y). \tag{48}$$

The field $\mu_{IIB}$ decomposes as

$$\mu_{IIB} = (\mu, \alpha_{0,2}; \Gamma) \in \left(\mathrm{PV}^{1,\bullet}(\mathbf{C}^3)[1] \oplus \mathrm{PV}^{1,\bullet}(\mathbf{C}^3)[1] \otimes H^{0,2}(Y)\right) \cap \ker \partial \oplus \Omega^{0,\bullet}(\mathbf{C}^3) \otimes H^{1,1}(Y), \tag{49}$$

where the divergence is with respect to the CY form on $\mathbf{C}^3$. We decompose $\Gamma$ further as $(\gamma_{1,1}^{a'}, \widetilde{\gamma}_{1,1}^{\omega})$ where $\widetilde{\gamma}_{1,1}^{\omega} \in \Omega^{0,\bullet}(\mathbf{C}^3)$ is associated to the Kähler form $\omega \in H^{1,1}(Y)$ and $a' = 1, \ldots, 19$ labels the remaining cohomology classes in $H^{1,1}(Y)$.

Similarly, if we neglect topological degrees of freedom, the field $\rho_{IIB}$ decomposes as

$$(\rho, \alpha_{2,0}, B) \in \left(\mathrm{PV}^{1,\bullet}(\mathbf{C}^3)[1] \otimes H^{2,2}(Y) \oplus \mathrm{PV}^{1,\bullet}(\mathbf{C}^3)[1] \otimes H^{2,0}(Y)\right) \cap \ker \partial \oplus \Omega^{0,\bullet}(\mathbf{C}^3) \otimes H^{1,1}(Y), \quad (50)$$

where we decompose $B$ as $(\beta_{1,1}^{a'}, \widetilde{\beta}_{1,1}^{\omega})$ where $\widetilde{\beta}_{1,1}^{\omega} \in \Omega^{0,\bullet}(\mathbf{C}^3)$ is associated to the Kähler form and $a' = 1, \ldots, 19$.

Finally, the field $\alpha_{IIB}$ decomposes, up to topological degrees of freedom, as

$$(\widetilde{\gamma}', \widetilde{\beta}', \gamma_{2,0}, \beta_{0,2}; \mathsf{A}) \in \Omega^{0,\bullet}(\mathbf{C}^3)^{\oplus 4} \oplus \left(\mathrm{PV}^{1,\bullet}(\mathbf{C}^3) \cap \ker \partial\right) \otimes H^{1,1}(\mathbf{C}^3), \quad (51)$$

where we further decompose $\mathsf{A}$ as $(\widetilde{\alpha}^{\omega}, \alpha_{1,1}^{a'})$ as we did above.

Now we can assemble these fields into twisted multiplets as follows.

- The fields

$$(\mu, \rho, \widetilde{\alpha}^{\omega}; \widetilde{\gamma}, \widetilde{\gamma}', \widetilde{\gamma}_{1,1}^{\omega}, \widetilde{\beta}, \widetilde{\beta}', \widetilde{\beta}_{1,1}^{\omega}), \quad (52)$$

    comprise the twist of the $\mathcal{N} = (2,0)$ graviton multiplet.

- The fields

$$(\alpha_{0,2}, \alpha_{2,0}, \alpha_{1,1}^{a'}; \gamma_{0,2}, \gamma_{2,0}, \gamma_{1,1}^{a'}, \beta_{0,2}, \beta_{2,0}, \beta_{1,1}^{a'}), \quad (53)$$

    comprise the twist of $1 + 1 + 19 = 21$ tensor multiplets with $\mathcal{N} = (2,0)$ supersymmetry.

To conclude, we see that in terms of multiplets the compactification of the twist of type IIB supergravity on a K3 surface decomposes as

$$\text{type IIB supergravity} \rightsquigarrow (i) + 21 (ii). \quad (54)$$

This combination of multiplets is known to be anomaly free and is compatible with the description of the K3 compactification of the physical type IIB supergravity (see, e.g., [19]) at the level of the holomorphic twist. It would be interesting to work out the anomaly cancellation mechanism in a purely holomorphic language, following similar work as in [20].

## 2.5 Backreaction as a deformation

From now on we fix the holomorphic coordinates $(z, w_1, w_2)$ on $\mathbf{C}^3$. We start with type IIB supergravity on $\mathbf{C}^3 \times Y$, with $Y$ a $K3$ surface, and consider a system of $D1$–$D5$ branes where some number of $D1$ branes wrap the complex line $\{w_i = 0\}$ in $\mathbf{C}^3$ and a point in $K3$:

$$\{w_i = 0\} \times \{x\} \subset \mathbf{C}^3 \times Y, \quad (55)$$

and some number of $D5$ branes wrap the same complex line $\{w_i = 0\}$ in $\mathbf{C}^3$ together with the entire $K3$ surface:

$$\{w_i = 0\} \times K3 \subset \mathbf{C}^3 \times Y. \quad (56)$$

The effective open string theory associated to this system of branes will be supported on the intersection of this system which is simply the complex line $\{w_i = 0\}$ in $\mathbf{C}^3$.

Using classic results [21], we can apply a duality to turn this into a D3 brane system which wraps

$$\mathbf{C} \times 0 \times \Sigma \subset \mathbf{C}^3 \times Y, \quad (57)$$

for a (special) Lagrangian two-cycle $\Sigma \subset Y$. This follows from the fact that any general D-brane (bound) state on $Y$ may be described by a Mukai vector $v$, which is a primitive vector such that

$F \in \Gamma^{4,20}, F^2 > 0$. Any two such vectors of equal length can be related to one another by $T$-duality transformations in $O(\Gamma^{4,20})$. Of course, matching the moduli between the two duality frames can be an involved task. For our purposes, we will only need a few basic features in this frame.[9] As in our setup, B-branes (which, again, are BPS with respect to some chosen $\mathcal{N} = (2,2)$ subalgebra of the $\mathcal{N} = (4,4)$ superconformal algebra) on K3 surfaces can wrap not only 2-cycles, but also curves of dimension 0 and 4 (i.e. points or the entire $K3$ surface).

In the last section, we argued that the compactification along a $K3$ surface becomes an extended version of Kodaira–Spencer theory where the extra fields are labeled by the cohomology of the surface. Upon compactification, the $D3$ system becomes a system of $B$-type branes in this extended version of Kodaira–Spencer theory.

The charge of these branes is labeled by a cohomology class

$$F \in H^2(Y) \subset R. \tag{58}$$

In particular, we take $F$ to be a primitive Mukai vector, as above. We denote

$$N \overset{\text{def}}{=} \langle F, F \rangle, \tag{59}$$

using the inner product on $H^2(Y)$. Explicitly, if $F = f\eta + \overline{f}\,\overline{\eta} + f^a \eta_a$ for $f, \overline{f}, f_a$ complex numbers, then $N = 2f\overline{f} + f^a f^b h_{ab}$ where $h_{ab}$ is the fixed non-degenerate symmetric pairing. Then the D-brane charge is related to the number of D1-D5 branes in the original duality frame $N \sim N_1 N_5$.[10] (To satisfy the primitivity condition, we assume $N_1, N_5$ are coprime. Since the supergravity theory is only sensitive to the product $N$, rather than the constituents $N_1, N_5$, it is often convenient to take $N_1 = N, N_5 = 1$.)

Notice that the *length* of the D-brane charge vector $F^2$ is of order $N$. We will always work in the supergravity limit in which any formal series in the inverse of these parameters is treated as an asymptotic series. More generally, let us explicate the parameters available to us in twisted supergravity. Exactly as in [3], the Kodaira-Spencer Lagrangian on flat space comes with an overall power of $\frac{1}{g_s^2}$ with $g_s$ the string coupling constant, which can be completely absorbed by rescaling of the fermionic variables $\eta_a \to g_s^{-1/2}\eta_a$. However, in the backreacted geometry, rescaling the fermionic variables rescales the D-brane charge vector $F$ by $\frac{1}{g_s}$ and $N$ by $\frac{1}{g_s^2}$ so that $g_s \sim \frac{1}{\sqrt{N}}$ as usual. We will always perform this rescaling. Notice that it is convenient for us to start with flat space and treat the backreaction *perturbatively*, i.e. as a small-$N$ expansion; as in [3], we find that the backreaction truncates to a finite series due to the presence of the fermionic coordinates, so one can work equally well in small-$N$ (which is convenient for the Koszul duality computations in the sequel), or in large-$N$ (as usual for holography).[11]

---

[9]A simpler application of these ideas, in which the dimensionality of the wrapped cycle does not change, is the following. The positive-definite 4-plane which specifies the hyperkähler structure on K3 can be decomposed into two orthogonal 2-planes which amounts to making a choice of complex structure and complexified (by the B-field) Kähler form. A quaternionic rotation of the 4-plane then exchanges the complex and Kähler structures, which is equivalent to a mirror symmetry transformation on the K3 surface. This will exchange the notion of B-branes and A-branes on K3, where B-branes wrap holomorphic curves (with respect to a chosen complex structure) and A-branes wrapping special Lagrangian 2-cycles. This point of view can also be reformulated as an application of the Strominger-Yau-Zaslow [22] picture of mirror symmetry as a composition of $T$-dualities acting on an elliptic fiber.

[10]We will always work in the supergravity approximation, and neglect the difference between the D-brane charges and numbers of D-branes in this work.

[11]By contrast, [2] works in the exact deformed geometry, rather than perturbatively around flat space, so that $N$ is fixed immediately as the period of the holomorphic volume form. It is a phenomenological observation in twisted holographic computations that observables (at the very least, observables involving operators with conformal weights that do not scale with $N$) either truncate to finite series in $N$ or can be resummed to quantities analytic in $N$, allowing us to match small-$N$ (Koszul duality) expansions with the large-$N$ holographic expansions; it would be desirable to have a more fundamental proof of these observations.

Generally, the backreaction deforms the geometry away from the locus of the brane. Before backreacting, we should say what geometry is actually being deformed. In the case of ordinary Kodaira–Spencer theory on $\mathbf{C}^3$, it was shown in [2] that the backreaction of B-branes along $\mathbf{C} \subset \mathbf{C}^3$ deformed the complex structure on $\mathbf{C}^3 \setminus \mathbf{C}$ to the deformed conifold, isomorphic to $SL_2(\mathbf{C})$. In the case of the compactification of the IIB string on $T^4$, the resulting backreacting geometry is a super enhancement of the conifold [3].

Our case is similar in that the branes are supported along the same locus as in [2,3]. The difference is that, compared to [2], we are working with a bigger space of fields, roughly extended by the cohomology of the $K3$ surface. Recall that $R = H^\bullet(Y)$ denoted the cohomology ring of the $K3$ surface. Notice that this algebra is canonically augmented by the map which sends all non-unit generators to zero (see [8] for a physical interpretation of the augmentation and its relationship to Koszul duality). A useful perspective on the extended version of Kodaira–Spencer theory we obtain by compactification along $K3$ is as a family of theories over the scheme $\operatorname{Spec} R$. This family has the property that over the augmentation ideal $\mathfrak{m}_R$ we obtain ordinary Kodaira–Spencer theory. We will see that in the case of type IIB compactified on a $K3$ surface that the backreaction determines an infinitesimal deformation of the complex manifold $\mathbf{C}^3 \setminus \mathbf{C}$ over $\operatorname{Spec} R$.

If $R$ is any local ring, an infinitesimal deformation of a complex manifold $M_0$ over $\operatorname{Spec} R$ is an element

$$\mu_{def} \in \operatorname{PV}^{1,1}(M_0) \otimes \mathfrak{m}_R, \tag{60}$$

satisfying the Maurer–Cartan equation. In our case, $M_0 = \mathbf{C}^3 \setminus \mathbf{C}$ and $\mu_{def}$ is a field sourced by the branes. The Maurer–Cartan equation is the equation of motion for $\mu_{def}$. The cohomology ring $R$ of a $K3$ surface is a local ring. Following [2,3], the backreaction of this system of branes introduces a twisted supergravity field

$$\mu_{BR} \in \operatorname{PV}^{1,1}(\mathbf{C}^3 \setminus \mathbf{C}) \otimes R, \tag{61}$$

which we can identify with an element of $\Omega^{2,1}(\mathbf{C}^3 \setminus \mathbf{C}) \otimes R$ using the Calabi–Yau form on $\mathbf{C}^3$. This field satisfies the following defining equations

$$\begin{aligned}
\overline{\partial} \mu_{BR} &= F \, \delta_{\mathbf{C} \subset \mathbf{C}^3}, \\
\partial \mu_{BR} &= 0.
\end{aligned} \tag{62}$$

For quantization we will also impose the standard gauge fixing condition that $\overline{\partial}^* \mu_{BR} = 0$ in terms of the usual codifferential $\overline{\partial}^*$. There is a unique solution to the above equations given by

$$\mu_{BR} = \frac{\epsilon^{ij} \overline{w}_i \mathrm{d}\overline{w}_j}{4\pi^2 |w|^4} \partial_z \otimes F. \tag{63}$$

Note that this field is of the form $\mu_{BR,0} \otimes F$ where $\mu_{BR,0} \in \operatorname{PV}^{1,1}$ is the Beltrami differential which gives rise to the deformed conifold [2]—all of the dependence on the parameters $\eta, \overline{\eta}, \eta_a$ is in the cohomology class $F$. Also we notice that $F \in \mathfrak{m}_R$.

Equations (62) imply that $\mu_{BR}$ determines an infinitesimal deformation of $\mathbf{C}^3 \setminus \mathbf{C}$ over $\operatorname{Spec} R$. The Kodaira–Spencer map associated to this infinitesimal deformation is of the form

$$KS \colon T_{\operatorname{Spec} R} \to H^1(\mathbf{C}^3 \setminus \mathbf{C}, T),$$

where $T$ denotes the tangent sheaf of the corresponding space, and simply maps a derivation $\delta$ of $A$ to the class

$$\delta(F) \left[ \frac{\epsilon^{ij} \overline{w}_i \mathrm{d}\overline{w}_j}{|w|^4} \partial_z \right] \in H^1(\mathbf{C}^3 \setminus \mathbf{C}, T).$$

We point out a more explicit characterization of this infinitesimal deformation of $\mathbf{C}^3 \setminus \mathbf{C}$ as a subvariety of $\mathbf{C}^4 \times \mathrm{Spec}\,R$ following similar manipulations as in [2,3]. Choose coordinates $(\eta, \overline{\eta}, \eta_a)$ so that $\mathrm{Spec}\,R$ is thought of as an algebraic subvariety of $\mathbf{C}^{22}$ (recall the dimension of the second cohomology of K3 is $b_2(K3) = 22$) cut out by the equations (29). From this point of view, the flux $F$ can be thought of as (the restriction of) a linear function on $\mathrm{Spec}\,R$. An arbitrary function

$$\Phi = \Phi(z, \overline{z}, w_i, \overline{w}_i, \eta, \overline{\eta}, \eta_a), \tag{64}$$

is holomorphic in the deformed complex structure determined by $\mu_{BR}$ if and only if

$$\mathrm{d}\overline{w}_i \frac{\partial \Phi}{\partial \overline{w}_i} + F \frac{\epsilon^{ij} \overline{w}_i \mathrm{d}\overline{w}_j}{4\pi^2 |w|^4} \frac{\partial \Phi}{\partial z} = 0. \tag{65}$$

The following functions are holomorphic for this deformed complex structure

$$\begin{aligned} u_1 &\overset{\mathrm{def}}{=} w_1 z - F \frac{\overline{w}_2}{4\pi^2 |w|^2}, \\ u_2 &\overset{\mathrm{def}}{=} w_2 z + F \frac{\overline{w}_1}{4\pi^2 |w|^2}. \end{aligned} \tag{66}$$

In addition to the relations satisfied by the variables $\eta, \overline{\eta}, \eta_a$, these functions satisfy

$$u_2 w_1 - u_1 w_2 = F. \tag{67}$$

We denote this geometry by $X$, which the above formulas have expressed as a quadratic cone inside $\mathbf{C}^4 \times \mathrm{Spec}(R)$, where $u_i, w_j$ are coordinates on the $\mathbf{C}^4$. The backreacted geometry is given by the locus where we further remove the locus where the coordinates $u_i, w_j$ are both zero; this is an open subset that we denote by $X^0 \subset X$.

We point out that there is a canonical projection

$$X^0 \to \mathrm{Spec}\,R, \tag{68}$$

thus exhibiting $X^0$ as an $R$-deformation of the conifold. In analogy with the backreaction in the ordinary $B$-model, we will refer to $X^0$ as the *K3 conifold*.

The holomorphic volume form $\Omega = \mathrm{d}z\,\mathrm{d}w_1\,\mathrm{d}w_2$ is unchanged upon making this deformation since $\mu_{BR}$ is divergence-free. We can write this volume form in the deformed coordinates above as

$$\Omega = w_1^{-1} \mathrm{d}u_1 \mathrm{d}w_1 \mathrm{d}w_2 \tag{69}$$

(or a similar expression involving $w_2^{-1}$) and note that this volume form is only well-defined on the fibers of the projection $X^0 \to \mathrm{Spec}(A)$.

## 2.6 A generalized Kodaira–Spencer theory

Before moving on, we point out that the above constructions make sense in the following generality. Fix a graded commutative ring $R$ equipped with a trace $\mathrm{tr}: R \to \mathbf{C}$. In the entirety of this section $R = H^\bullet(Y)$ and $\mathrm{tr}(a) = \int_Y a$, where $Y$ is a K3 surface (or $T^4$ as in [3]).

More generally we can consider a complex three-dimensional theory whose fields, in the BV formalism, are given by

$$\mu \in \mathrm{PV}^{1,\bullet}(\mathbf{C}^3) \otimes R[1], \tag{70}$$

and

$$\alpha \in \Omega^{0,\bullet}(\mathbf{C}^3) \otimes R, \qquad \gamma \in \Omega^{0,\bullet}(\mathbf{C}^3) \otimes R[2]. \tag{71}$$

The action functional is

$$\frac{1}{2}\int_{\mathbf{C}^3}\epsilon^{ijk}\operatorname{tr}\overline{\partial}\mu_i(\partial^{-1}\mu)_{jk}\,\mathrm{d}^3Z+\int_{\mathbf{C}^3}\operatorname{tr}\alpha\overline{\partial}\gamma\,\mathrm{d}^3Z+\frac{1}{6}\int_{\mathbf{C}^3}\epsilon_{ijk}\operatorname{tr}\mu_i\mu_j\mu_k\,\mathrm{d}^3Z+\int_{\mathbf{C}^3}\operatorname{tr}\alpha\mu_i\partial_{z_i}\gamma\,\mathrm{d}^3Z\,.\tag{72}$$

We refer to this as $R$-Kodaira–Spencer theory. For a general ring $R$, we lose the interpretation of type IIB supergravity compactified on some holomorphic symplectic surface. On the other hand, judicious choices of $R$ may allow one to consider 'compactifications' of supergravity on possibly singular surfaces.

## 3  Enumerating twisted supergravity states

We have derived our twisted supergravity theory in the backreacted geometry; we will refer to the latter henceforth as the $K3$ conifold, adapting the terminology of [3]. Our theory conjecturally captures a protected subsector of IIB supergravity on $\mathrm{AdS}_3 \times S^3 \times K3$ (which we will refer to as the untwisted theory), and we would like to perform some sanity checks of this conjecture. In particular, in this section we demonstrate that the partition function of twisted supergravity states reproduces the seminal count of $\frac{1}{4}$-BPS Kaluza–Klein modes in the untwisted theory [10]. The methods in this section are only slight modification of those in [2,3], so we refer to these original references for more details.

### 3.1  Inclusion of boundary divisors

In order to enumerate twisted supergravity states, we must understand the boundary divisors of the K3 conifold, which are the geometric support for the asymptotic scattering states that participate in (the holomorphic analogue of) Witten diagram computations.[12]

The idea is to compactify the $K3$ conifold $X^0$ to a super-projective variety $\overline{X^0}$ inside $\mathbf{CP}^4 \times \operatorname{Spec}(R)$.[13] We give the $\mathbf{CP}^4$ homogeneous coordinates $U_i, W_i, Z$, so that we can complete the K3 conifold defined by equation 67 to

$$\epsilon^{ij}U_iW_j = FZ^2\,.\tag{73}$$

The boundary is then at $Z = 0$, given by $\epsilon^{ij}U_iW_j = 0$, which is the variety $\mathbf{CP}^1 \times \mathbf{CP}^1 \times \operatorname{Spec}(R) \subset \mathbf{CP}^3 \times \operatorname{Spec}(R)$. As in [2], the two $\mathbf{CP}^1$'s may be understood, respectively, as the 2-sphere boundary of $\mathrm{AdS}_3$, and the $S^2$ base of the $S^3$ factor, viewed as a Hopf fibration. Each $\mathbf{CP}^1$ is naturally acted on by a copy of $\mathrm{SL}_2$.

To determine the complex structure in the neighborhood of the boundary, we must find coordinates which are holomorphic in the deformed geometry, as described in the previous section. To start, we can endow the two $\mathbf{CP}^1$'s with holomorphic coordinates $w, z$ and anti-holomorphic coordinates $\bar{w}, \bar{z}$ (in addition to the coordinates $\eta$ on $\operatorname{Spec}(R)$), and take the $\mathbf{CP}^1$ with coordinates $z, \bar{z}$ to be the boundary of $AdS_3$ on which the dual twisted SCFT will live. In addition, we can specify a coordinate normal to the two boundary spheres by $n$, which has a simple pole at $z = \infty$ and at $w = \infty$. We need to specify the behavior of Kodaira-Spencer fields at $n = 0$, where the complement of $n = 0$ is the uncompactified K3 conifold. In these

---

[12]While we will not study bulk scattering directly in this work, it would be interesting to explore methods to make such bulk computations more efficient, perhaps by generalizing the technology of [23,24] to curved backgrounds.

[13]Note that other compactifications are possible, depending on one's application. In [25], the deformed conifold $SL(2,\mathbb{C})$ was not compactified to a quadric, as here, but instead was compactified inside the blow up of a flag variety. That compactification was the one compatible with the symmetries inherent from viewing the deformed conifold as the twistor space of 4d Burns space, which has isometry group $SU(2) \times U(1)$. It would be interesting to extend the analysis of [25] to the conifolds of [3] and the present article, and view them as twistor spaces in turn.

coordinates, the holomorphic volume form is

$$\Omega = -\frac{dn\,dw\,dz}{n^3} + \frac{F}{n}\frac{dn\,dw\,d\bar{w}}{(1+|w|^2)^2}\,. \tag{74}$$

With these coordinates, one can straightforwardly define twisted supergravity states via the usual AdS/CFT extrapolate dictionary.

However, this naive coordinate system is not holomorphic. Rather, the complex structure is deformed by the Beltrami differential

$$Fn^2 d\bar{w}\frac{1}{(1+|w|^2)^2}\partial_z\,. \tag{75}$$

Holomorphic functions in the neighborhood of the boundary are given by

$$w_1 \stackrel{\text{def}}{=} \frac{1}{n}\,, \tag{76}$$

$$w_2 \stackrel{\text{def}}{=} \frac{w}{n}\,, \tag{77}$$

$$u_1 \stackrel{\text{def}}{=} \frac{z}{n} - Fn\frac{\bar{w}}{(1+|w|^2)^2}\,, \tag{78}$$

$$u_2 \stackrel{\text{def}}{=} \frac{wz}{n} + Fn\frac{1}{(1+|w|^2)^2}\,. \tag{79}$$

Notice that these coordinates have poles at $n = 0$ and satisfy $u_2 w_1 - u_1 w_2 = F$. Moreover, in these coordinates the holomorphic volume form again takes the canonical form

$$\Omega = \frac{du_1\,dw_1\,dw_2}{w_1}\,. \tag{80}$$

## 3.2 Enumerating states in Kodaira–Spencer theory

To describe boundary conditions on the fields in our theory, we can use the partial compactification of the $K3$ conifold described in §3.1. All that remains is, following the usual AdS/CFT prescription, to specify vacuum boundary conditions for our Kodaira-Spencer supergravity fields. Then, our twisted supergravity states are solutions to the equation of motion that satisfy these vacuum boundary conditions except at a point on the conformal boundary of the $\text{AdS}_3$ factor, say $z_*$. In other words, twisted supergravity states are, as usual, local modifications of the boundary conditions, which are equivalent to boundary operators placed along $\mathbf{CP}^1_w \times \{z_*\}$.

Recall that there are three fundamental fields for Kodaira–Spencer theory. Two fundamental fields $\alpha, \gamma$ are Dolbeault forms of type $(0, \bullet)$. The last fundamental field $\mu$ is a $(0, \bullet)$ form valued in in the holomorphic tangent bundle. We can use the Calabi–Yau form to view $\mu$ as a Dolbeault form of type $(2, \bullet)$.

- The vacuum boundary condition for the fields $\alpha, \gamma$ is that each are divisible by the coordinate $n$. That is, we require these fields to vanish on the boundary divisor.

- The vacuum boundary condition for the field $\mu$ is that, when viewing it as a Dolbeault form of type $(2, \bullet)$, it can be expressed as a sum of terms which are each wedge products of $d\log n, dw, dz, d\bar{n}, d\bar{w}, d\bar{z}$ with coefficients that are regular at $n = 0$. (Notice that we allow this field to have logarithmic poles on the boundary divisor, although one may also choose to impose the more restrictive condition that $\mu$ is a regular Dolbeault form).

We can now enumerate the supergravity states that satisfy these boundary conditions except for at a point-localized disturbance or source. Here, we consider ordinary Kodaira–Spencer theory on $\mathbf{C}^3$ with $B$-branes wrapping $\mathbf{C} \subset \mathbf{C}^3$. The result is a recapitulation of [2], to which we refer the reader for more details.

Denote by $\left(\frac{\mathbf{m}}{\mathbf{2}}\right)_S$ the short representation of $\mathfrak{psu}(1,1|2)$ whose highest weight vector has $(J_0^3, L_0)$ eigenvalues $(\frac{m}{2}, \frac{m}{2})$. Denote by $y$ the fugacity for the $U(1)$ symmetry $2J_0^3$ and $q$ the fugacity for the $U(1)$ symmetry $L_0$. Let

$$D = (1-q)(1-q^{1/2}y)(1-q^{1/2}y^{-1}). \tag{81}$$

This is the denominator that will appear in the single particle index computed below. The factor $(1-q)^{-1}$ arises from the tower of $\partial_z$-derivatives. The factors $(1-q^{1/2}y^{\pm 1})^{-1}$ arise from the towers of $\partial_{w_1}, \partial_{w_2}$ respectively.

- State $\mu \sim n^{-k}\mathrm{d}\log n\mathrm{d}w_1\delta_{z=0}$. For $k \geq 1$ these even states and their descendants contribute

$$\frac{yq^{1/2}}{D}, \tag{82}$$

  to the single particle index.

- Lowest lying state $\mu \sim n^{-k}\mathrm{d}\log n\mathrm{d}w_2\delta_{z=0}$. For $k \geq 1$ these even states and their descendants contribute

$$\frac{y^{-1}q^{1/2}}{D}, \tag{83}$$

  to the single particle index.

- Lowest lying state $\mu \sim n^{-k}\mathrm{d}\log n\mathrm{d}z\delta_{z=0}$. For $k \geq 2$ these even states and their descendants contribute

$$\frac{q^2}{D} - \frac{q}{D}, \tag{84}$$

  to the single particle index. The term $-q/D$ appears due to the constraint satisfied by the field $\mu$, $\partial_\Omega \mu = 0$.

- State $\alpha \sim n^{1-k}\delta_{z=0}$. For $k \geq 1$ these odd states and their descendants contribute

$$-\frac{q}{D}, \tag{85}$$

  to the single particle index.

- State $\gamma \sim n^{1-k}\delta_{z=0}$. For $k \geq 1$ these odd states and their descendants contribute

$$-\frac{q}{D}, \tag{86}$$

  to the single particle index.

In total we find that the single-particle gravitational index is

$$\frac{q^2 - 3q + q^{1/2}(y+y^{-1})}{(1-q)(1-q^{1/2}y)(1-q^{-1/2}y^{-1})} = \frac{yq^{1/2}}{1-yq^{1/2}} + \frac{y^{-1}q^{1/2}}{1-y^{-1}q^{1/2}} - \frac{q}{1-q}. \tag{87}$$

Alternatively, one can use an explicit expression for the character $\chi_m(q,y)$ of the $\mathfrak{psu}(1,1|2)$-representation $\left(\frac{\mathbf{m}}{\mathbf{2}}\right)_S$, see equation 4.1.16-17 of [3], and evaluate the single particle index

$$\chi\left(\oplus_{m\geq 1}\left(\frac{\mathbf{m}}{\mathbf{2}}\right)_S\right) = \sum_{m\geq 0}\chi_m(q,y). \tag{88}$$

The result is the same.

### 3.3 The twisted supergravity elliptic genus

The supergravity states were enumerated in [3] in the case that one compactifies type IIB supergravity along either $T^4$ or $K3$. We briefly recall the results here, with an emphasis on the case of a $K3$ surface.

The twisted supergravity states organize into a representation for the super Lie algebra $\mathfrak{psu}(1,1|2)$. The bosonic factor of this super Lie algebra is $\mathfrak{su}(2)_L \times \mathfrak{su}(2)_R$. The first copy is the global conformal transformations in the $z$-plane and the second copy is the $R$-symmetry algebra which rotates the $w$-coordinate. We take the Cartan of this Lie algebra to be generated by $(L_0, J_0^3)$.

Denote by $(\frac{\mathbf{m}}{\mathbf{2}})_S$ the short representation of $\mathfrak{psu}(1,1|2)$ whose highest weight vector has $(L_0, J_0^3)$ eigenvalue $(m/2, m/2)$ [11]. As an example, the short representation $(\mathbf{1})_S$ consists of a boson with weight $(L_0 = 1, J_0^3 = 1)$, which in our notation corresponds to

$$\mu \sim n^{-2} \mathrm{d}\log n \mathrm{d}z \delta_{z=0}. \tag{89}$$

There are also two fermions in $(\mathbf{1})_S$ with weights $(3/2, 1/2)$ corresponding to the states

$$\alpha \sim n^{-1}\delta_{z=0} + \cdots, \qquad \gamma \sim n^{-1}\delta_{z=0} + \cdots, \tag{90}$$

and another boson of weight $(2, 0)$ corresponding to

$$\mu \sim n^{-2} \mathrm{d}\log n \mathrm{d}w \delta_{z=0} + \cdots. \tag{91}$$

Here, the ellipses denote additional terms required to express the fields in the holomorphic coordinates of the deformed geometry (see [3] for the complete expressions in the $T^4$ case). In particular, only a finite number of terms are required to correct the holomorphicity of these expressions, due to the fact that the relations imposed on the coordinates of $\mathrm{Spec}(R)$ cause the expansions in the $\eta$'s to truncate.

We consider twisted type IIB supergravity on a Calabi–Yau surface $X$, where $X$ could be $T^4$ or a $K3$ surface.

The supergravity states for the D1-D5 brane system in twisted type IIB supergravity on a compact Calabi–Yau surface $X$ decompose as

$$\bigoplus_{m\geq 1}\left(\frac{\mathbf{m}}{\mathbf{2}}\right)_S \otimes H^\bullet(X) = \bigoplus_{m\geq 1}\bigoplus_{i,j}\left(\frac{\mathbf{m}}{\mathbf{2}}\right)_S \otimes H^{i,j}(X). \tag{92}$$

In particular, according to the previous section, when $X$ is a $K3$ surface the single particle twisted supergravity index is

$$f_{KS}(q,y) = 24\frac{q^2 - 3q + q^{1/2}(y + y^{-1})}{D}. \tag{93}$$

This result should be compared to [11], where the space of supergravity states upon supersymmetric localization (that is, the chiral half of the supergravity states) is found to be

$$\bigoplus_{m\geq 0}\bigoplus_{i,j}(\frac{\mathbf{m+i}}{\mathbf{2}})_S \otimes H^{i,j}(X). \tag{94}$$

The answers agree in the range where the highest weight of the short representation is at least two. The low weight discrepancies break up into two types:

- In [11] there is an extra factor of $(\mathbf{0})_S \otimes H^{0,i}(X)$. So, in the case that $X$ is a $K3$ surface there are two extra bosonic operators in the analysis of [11]. In [3] it was pointed out that these are topological operators, annihilated by $L_{-1}$, and have nonsingular OPE with

all remaining operators. Notice that these states are removed by hand from the infinite-$N$ $\text{Sym}^N(K3)$ elliptic genus in [11] (as we will review below), because their degeneracy scales with $N$. Though they naturally appear on the SCFT side, and in particular are well-defined for any finite $N$, the minimal Kodaira-Spencer theory does not contain them.

- In our analysis there is an extra factor of $(\frac{1}{2})_S \otimes H^{2,j}(X)$. In the case that $X$ is a $K3$ surface these two bosonic states can be removed by hand from the spectrum while preserving the $SO(21)$ symmetry. We will comment more on these modes in §5 when we examine their OPEs. Roughly speaking, they are the twist of the center of mass degrees of freedom, which are often removed in the near-horizon limit in holography. This limit is a bit subtle in twisted supergravity, and we see that these degrees of freedom most naturally remain in the Kodaira-Spencer theory. However, the states that we are interested in form a consistent subalgebra to which we restrict our attention (formally, the algebra generated by this additional twisted multiplet is a semidirect product with our subalgebra of interest. Note that it cannot be a trivial direct product and its algebra elements are, in particular, acted upon by the 2d $\mathcal{N}=4$ superconformal algebra).

Denote the single particle index of the supergravity states, described in equation (94), by $f_{sugra}(q, y)$. One of the main results of [11] is that the corresponding multiparticle index agrees with the large $N$ elliptic genus of the orbifold CFT of a $K3$ surface

$$\chi_{NS}(\text{Sym}^\infty X; q, y) = \text{PExp}\left[f_{sugra}(q, y)\right],\tag{95}$$

where PExp is the plethystic exponential defined by $\text{PExp}[f(x)] = \exp\left(\sum_{k=1}^\infty \frac{f(x^k)}{k}\right)$, which effects a "multi-particling" operation. For $X$ a $K3$ surface, the states $(\frac{1}{2})_S \otimes H^{2,\bullet}(X)$ contribute the single particle index

$$2f_1(q, y) = \frac{2}{1-q}\left(-2q + q^{1/2}(y + y^{-1})\right).\tag{96}$$

If we subtract this from the supergravity index we find an exact match with the supergravity index computed by [11]:

$$f_{sugra}(q, y) = f_{KS}(q, y) - 2f_1(q, y).\tag{97}$$

## 3.4 Global symmetry algebra

In this section we characterize the global symmetry algebra of the dual CFT at infinite $N$ from the point of view of the gravitational, or Kodaira–Spencer, theory following [2, 3]. The global symmetry algebra is, by definition, a subalgebra of the modes of the operators[14] of the CFT which preserve the vacuum at both 0 and $\infty$. Explicitly, if $\mathcal{O}$ is an operator of spin $\Delta$, then the modes

$$\oint z^m \mathcal{O}(z) \mathrm{d}z,\tag{98}$$

for $0 \leq m \leq 2\Delta - 2$ close as an algebra and preserve the vacua at $0, \infty$. Generally, the global symmetry algebra is a subalgebra of the mode algebra of the vertex algebra. For us, it can be expressed as the universal enveloping algebra of a particular Lie superalgebra.

---

[14]Again, we work with operators that survive in the planar limit; in the gauge theory context, these would be the single trace operators.

From the Kodaira-Spencer theory perspective, these are infinitesimal gauge symmetries which preserve the vacuum solutions to the equations of motion on the K3 conifold. Following a similar argument as in [3], one finds that the global symmetry algebra is the enveloping algebra of a Lie superalgebra of the form

$$\text{Vect}_0\left(X^0/\operatorname{Spec}R\right) \oplus \mathcal{O}(X^0) \otimes \Pi\mathbf{C}^2\,, \tag{99}$$

where:

- $X^0$ is the $R$-conifold defined as a family over $\operatorname{Spec}R$ where we have removed the singular locus; see section 2.5.

- $\mathcal{O}(X^0)$ denotes the algebra of holomorphic functions on $X^0$. By Hartog's theorem this is the algebra generated by the bosonic linear functions $u_i, w_j, \eta, \overline{\eta}, \eta_a$ where $i,j = 1,2$, $a = 1,\ldots,20$ subject to the relations

$$\eta^2 = \overline{\eta}^2 = \eta_a\eta_b - h_{ab}\eta\overline{\eta} = 0\,, \qquad \epsilon^{ij}u_iw_j = F\,.$$

- $\text{Vect}_0\left(X^0/\operatorname{Spec}(R)\right)$ is the Lie algebra of divergence-free holomorphic vector fields which point in the direction of the fibers of $X^0 \to \operatorname{Spec}(R)$ (those holomorphic vector fields preserving the holomorphic volume form on the fibers).

- $\Pi(-)$ denotes parity shift, so that this is a Lie superalgebra.

- The nontrivial Lie brackets (and anti-brackets) are:

$$\begin{aligned}
[V,V'] &= \text{commutator of vector fields,}\\
[V,f] &= V(f)\,,\\
[f_i,g_j] &= \epsilon_{ij}\Omega^{-1}\left(\partial f_i \wedge \partial g_j\right)\,,
\end{aligned} \tag{100}$$

where $V \in \left(X^0/\operatorname{Spec}(R)\right)$, $f_i, g_j \in \mathcal{O}(X^0) \otimes \Pi\mathbf{C}^2$.

A characterization of the global symmetry algebra will follow from the computation of OPEs of the boundary CFT (more precisely, its chiral algebra of holomorphic symmetries). As in the examples of [2,3], this global symmetry algebra is large enough to fix the *planar* 2 and 3-point functions.[15]

## 4 The twisted symmetric orbifold CFT

Supergravity on $AdS_3 \times S^3 \times Y$, where $Y$ is either $T^4$ or a $K3$ surface, is expected to be holographically dual to a particular two-dimensional superconformal field theory (SCFT). Though our primary interest in this note is $K3$, with the $T^4$ case studied in [3], we can be agnostic about $Y$ for many aspects of the analysis.

We will briefly review this system of interest, following [26] and references therein, with a focus towards applying the holomorphic twist to this system and isolating the $\frac{1}{4}$-BPS states. Of course, this SCFT is the IR limit of the field theory that arises from the zero modes of the open strings on the $D1-D5$ branes. The lowest-lying modes of open strings, which provide an effective field theory description of the $D1$ and $D5$-branes, naturally furnish a gauge theory whose IR limit we are primarily interested in. In principle, one could perform the twist, which is in principle insensitive to RG flow, of either the UV D1-D5 gauge theory or the symmetric orbifold CFT.

---

[15]In [3] it was shown that, for $N \to \infty$, all two-point functions of states with $SU(2)_R$ spin $\geq 1$ vanish. The same argument holds in this case, though of course at finite $N$ there will be nonvanishing 2-pt functions.

We recall that the $D5-D5$ strings give rise to a six-dimensional supersymmetric $U(N_5)$ gauge theory and the $D1-D1$ strings likewise produce a $U(N_1)$ gauge theory; $D1-D5$ strings will produce matter multiplets in the bifundamental of these gauge groups. When all the $D$-branes are coincident the gauge theory is in the Higgs phase and when some of the adjoint scalars in the field theory acquire a vev, corresponding to transverse separation of the branes, the theory is in the Coulomb phase. We will focus on the Higgs phase of the gauge theory throughout.[16]

On the Higgs branch, one must solve the vanishing of the bosonic potential (i.e. $D$-flatness equations) modulo the gauge symmetries $U(N_1) \times U(N_5)$ to obtain the moduli space. If one imagined that both sets of $D$-branes were supported on a noncompact six-dimensional space, these $D$-flatness equations can be rewritten to reproduce the ADHM equations for $N_1$ instantons of a six-dimensional $U(N_5)$ gauge theory a la [28]. So far, we have a description of the dual field theory in terms of an instanton moduli space, namely the moduli space of $N_1$ instantons of a $U(N_5)$ gauge theory on $Y$, for which a useful model is the Hilbert scheme of $N_1 N_5$ points on $Y$.[17] The (conformally invariant limit of the) gauge theory description is expected to only capture the regime of vanishing size instantons (i.e. when the hypermultiplets have small vevs). One can understand that the gauge theory description is approximate by noticing that the Yang-Mills couplings are given in terms of the $Y$ volume $V$ and string coupling as $g_1^2 = g_s(2\pi\alpha')$, $g_5^2 = g_s V/(\alpha'(2\pi)^3)$ so for energies much smaller than the inverse string length the gauge theories are strongly coupled [26]. To get the SCFT we take an IR limit, which would be dual to a near-horizon limit from the closed string point of view. In this limit, the gauge theory moduli space becomes the target space of the low-energy sigma-model. It has been argued that the correct instanton moduli space is a smooth deformation of the symmetric product theory $Sym^{N_1 N_5}(\tilde{X})/S_{N_1 N_5}$.[18] Indeed, there is a point in the SCFT moduli space (far from the supergravity point itself) where the theory takes precisely the symmetric orbifold form. The orbifold point is the analogue of free Yang-Mills theory in the perhaps more-familiar $AdS_5 \times S^5/$ 4d $\mathcal{N}=4$ SYM duality, and is dual to a stringy point in moduli space which has been explored extensively in recent years (see, e.g., [30–32]).

As usual, one can focus on moduli-independent quantities to provide preliminary matches between the supergravity and orbifold points, such as the signed count of $\frac{1}{4}$-BPS states at large-$N$, via the elliptic genus. The elliptic genus matches the corresponding count of BPS (or equivalently, twisted) supergravity states [11], which we reproduced in the previous section. We review the $N \to \infty$ elliptic genus computation and its matching to the twisted supergravity index below. This matching follows from the formal equivalence of the elliptic genus to the vacuum character of the chiral algebra in the holomorphic twist; this quantity is also sometimes referred to as the partition function of the half-twisted theory.

It would be preferable to "categorify" the standard elliptic genus computation, and reproduce it directly from the twisted CFT perspective using the holomorphic twist of the symmetric orbifold CFT.[19] As we mentioned, in two dimensions this is also known as the half-twist [5,6]. It is well-known that the half-twist of a sigma-model can be mathematically formulated as the chiral de Rham complex [6,34,35], and indeed this is precisely what our holomorphic twist captures.

---

[16]See [27] for a recent analysis of twisted holography in the Coulomb phase.

[17]For the purposes of this discussion, we will ignore the center of mass factor of the moduli space that produces a $\tilde{X}$ factor, for some $\tilde{X}$ not necessarily the same as the compactification $Y$. The relationship between the two manifolds in the $T^4$ case is clarified in [29].

[18]Here we are taking both $N_1, N_5$ large.

[19]Of course, whenever one wants to match more refined observables than the elliptic genus from the symmetric orbifold theory to the supergravity point (rather than the stringy dual of [32]), one must deal with moduli-dependence, e.g. [33].

Unfortunately, obtaining a global description of the half-twist on a curved, compact manifold is a nonperturbative computation subject to worldsheet instanton corrections, and so prohibitively difficult with current technology. We will instead review some aspects of the holomorphic twist from the perspective of the UV brane worldvolume gauge theory, and then discuss the connection to the half-twist/chiral de Rham complex of the symmetric orbifold SCFT, explaining their formal equivalence. When discussing the chiral de Rham complex, we must approximate K3 as $\mathbb{C}^2$.

## 4.1 Branes in twisted supergravity

We have already recollected the proposal of [1] that the twist of type IIB supergravity is equivalent to the topological $B$-model on a Calabi–Yau fivefold. At the level of branes, this proposal further asserts that $D(2k-1)$-branes in type IIB corresponds to topological $B$-branes. We use that perspective here to deduce the worldvolume CFT of the twist of the $D1/D5$ system in type IIB supergravity.

We consider the system of $D1/D5$ branes in the twist of type IIB on a Calabi–Yau fivefold $Z$. For simplicity, we assume that we have a collection of $N_1 = N$ $D1$ branes supported along a closed Riemann surface

$$\Sigma \subset Z,$$

together with a single $D5$ brane which is parallel to the $D1$ branes.

In topological string theory, one views branes as objects in some category. Morphisms between objects represent open strings stretching between two branes. In particular, a general feature of topological string theory is that the open string fields which start and end on the same brane can be described in terms of the algebra of derived endomorphisms of the object representing the brane. Indeed, following [36], one constructs a Chern–Simons theory based off of this derived algebra of endomorphisms where the gauge fields are degree one elements in the algebra of derived endomorphisms. In the $B$-model, the category is the category of coherent sheaves on the Calabi–Yau manifold. Fields of the corresponding open-string field theory (which start and on on the same brane) are given as holomorphic sections of the sheaf of derived endomorphisms. Following [1], we will use a Dolbeault model which resolves a sheaf of holomorphic sections to describe the space of fields as the cohomological shift by one of the Dolbeault resolutions of derived endomorphisms.

We consider $D1$ branes that are a sum of simple branes labeled by the structure sheaf $\mathcal{O}_\Sigma$. In particular, $N$ such $D1$ branes are represented by the object $\mathcal{O}_\Sigma^{\oplus N}$ in the category of quasi-coherent sheaves on the Calabi–Yau fivefold $Z$. A model for the sheaf of derived endomorphisms of $\mathcal{O}_\Sigma$ is the holomorphic sections of the exterior algebra of the normal bundle $\mathcal{N}_\Sigma$ of $\Sigma$ in $Z$. A model for the sheaf of derived endomorphisms of a stack of $N$ such branes is therefore

$$\operatorname{Ext}_{\mathcal{O}_Z}\left(\mathcal{O}_\Sigma^{\oplus N}\right) \simeq \mathfrak{gl}(N) \otimes \wedge^\bullet \mathcal{N}_\Sigma. \tag{101}$$

Thus, the Dolbeault model for the open string fields which stretch between two such $D1$ branes is given by

$$\Omega^{0,\bullet}\left(\Sigma, \mathfrak{gl}(N) \otimes \wedge^\bullet \mathcal{N}_\Sigma\right)[1]. \tag{102}$$

If we take $X$ to the be the total space of the bundle $\mathcal{N}_\Sigma$ then the Calabi–Yau condition requires $\wedge^4 N_\Sigma = K_\Sigma$. In the case $\Sigma = \mathbf{C}$ and $Z = \mathbf{C}^5$ we can write the open string fields (102) as

$$\Omega^{0,\bullet}\left(\mathbf{C}, \mathfrak{gl}(N)[\varepsilon_1, \dots, \varepsilon_4]\right)[1]. \tag{103}$$

Here the $\varepsilon_i$ are odd variables that carry spin 1/4, meaning they transform as constant sections of the bundle $K_{\mathbf{C}}^{1/4}$. This is precisely the field content of the holomorphic twist of two-dimensional $\mathcal{N} = (8,8)$ pure gauge theory which is the worldvolume theory living on a stack of $D1$ branes in twisted supergravity on flat space.

Next, we consider $D1 - D5$ strings. The open string fields are given by

$$\Omega^{0,\bullet}\left(\Sigma, \underline{\mathrm{Ext}}_{\mathcal{O}_X}\left(\mathcal{O}_Z, \mathcal{O}_\Sigma^{\oplus N}\right)\right). \tag{104}$$

Again, on flat space with $\Sigma = \mathbf{C}$ this can be written in a more explicit way as

$$\Omega^{0,\bullet}\left(\mathbf{C}, K_{\mathbf{C}}^{1/2}[\varepsilon_3, \varepsilon_4]\right) \otimes \mathrm{Hom}(\mathbf{C}, \mathbf{C}^N) = \Omega^{0,\bullet}\left(\mathbf{C}, K_{\mathbf{C}}^{1/2}[\varepsilon_3, \varepsilon_4]\right) \otimes \mathbf{C}^N. \tag{105}$$

Together with the $D5 - D1$ strings we get

$$\Omega^{0,\bullet}\left(\mathbf{C}, K_{\mathbf{C}}^{1/2}[\varepsilon_3, \varepsilon_4]\right) \otimes T^*\mathbf{C}^N. \tag{106}$$

In total, we see that the open-strings of the $D1/D5$ system along $\Sigma = \mathbf{C}$ are given by the Dolbeault complex valued in the following holomorphic vector bundle

$$\left(\mathfrak{gl}(N)[\varepsilon_1, \varepsilon_2][1] \oplus K_{\mathbf{C}}^{1/2} \otimes T^*\mathbf{C}^N\right) \otimes \mathbf{C}[\varepsilon_3, \varepsilon_4]. \tag{107}$$

If we choose twisting data so that the odd variables carry degree $\deg \varepsilon_1 = \deg \varepsilon_2 = +1$ then the bundle in parentheses can be written as

$$\mathfrak{gl}(N)[1] \oplus K_{\Sigma}^{1/2} \otimes T^*\left(\mathfrak{gl}(N) \oplus \mathbf{C}^N\right) \oplus \mathfrak{gl}(N)[-1]. \tag{108}$$

The first summand represents the ghosts of the holomorphic CFT and the last summand the anti-ghosts. The gauge symmetry in the middle term is induced from the standard action of $\mathfrak{gl}(N)$ on $T^*\left(\mathfrak{gl}(N) \oplus \mathbf{C}^N\right)$ by Hamiltonian vector fields (this is induced from the adjoint + fundamental action on the base of the cotangent bundle). Thus, we see that this model describes ($K_{\Sigma}^{1/2}$-twisted) holomorphic maps from $\Sigma$ into the well-known GIT description of the symmetric orbifold $\mathrm{Sym}^N \mathbf{C}^2$. That is, the worldvolume theory living on a stack of twisted $D1$ branes is the holomorphic $\sigma$-model of maps into the target $\mathrm{Sym}^N \mathbf{C}^2$.

This analysis happened entirely in flat space. The $D1$ branes wrapped

$$\mathbf{C} \times 0 \times 0 \times 0 \times 0 \subset \mathbf{C}^5, \tag{109}$$

while the $D5$ brane wrapped $\mathbf{C} \times \mathbf{C}^2 \times 0 \times 0 \subset \mathbf{C}^5$. At this stage, it is natural to replace this $\mathbf{C}^2$ by a general holomorphic symplectic surface $Y$ to arrive at the well-established expectation that the worldvolume theory, after twisting, is a holomorphic $\sigma$-model with target $\mathrm{Sym}^N Y$. A careful derivation of this would require one to work in the derived category of sheaves on $\mathbf{C}^3 \times Y$, which we have not done here.

## 4.2 The symmetric orbifold elliptic genus at large $N$

For completeness, we briefly recall the elliptic genus computation using the orbifold point in the string moduli space, which reproduces signed counts of $1/4$-BPS states in the SCFT. This is formally equal to the partition function of the chiral de Rham complex, or holomorphically twisted theory on the same underlying space.

We will take the effective 2d brane system to be supported on $\mathbf{R} \times S^1$ after compactification on $Y$, so that the CFT is defined on the cylinder. On the cylinder, the NS sector corresponds to anti-periodic boundary conditions on the fermions. The sigma model is then the $\mathcal{N} = (4, 4)$ theory whose bosonic fields are valued in maps from $S^1 \to Sym^N(Y)$.

The physical SCFT has R-symmetries $SO(4) \simeq SU(2)_L \times SU(2)_R$ dual to rotations of the $S^3$ and symmetries under a global $SO(4)_I \simeq SU(2)_a \times SU(2)_b$ of transverse rotations; this symmetry is broken by compactification on $Y$. Although broken by the background, $SO(4)_I$ is

still often used to organize the field content of the compactified theory, and acts as an outer automorphism on the $\mathcal{N} = (4,4)$ superconformal algebra. As is well known, the isometries of $AdS_3 \times S^3$ are $SL(2,\mathbf{R}) \times SL(2,\mathbf{R}) \times SO(4)$ which form the bosonic part of the supergroup $SU(1,1|2) \times SU(1,1|2)$. These symmetries form the global subalgebra of the $\mathcal{N} = (4,4)$ superconformal algebra.

Part of the underlying chiral algebra of the $\mathcal{N} = (4,4)$ SCFT OPEs is the usual holomorphic (small) $\mathcal{N} = 4$ superconformal algebra with $c = 6N$ (which can be explicitly constructed as a diagonal sum over the $N$ copies of the seed $c = 6$ sigma models). Part of the $\mathcal{N} = 4$ superconformal algebra involves $SU(2)$ spin 1 currents $\{J^a(z)\}$; the central charge determines the level of this current algebra as

$$J^a(z)J^b(w) \sim \frac{c}{12}\frac{\delta^{ab}}{(z-w)^2} + i\epsilon^{ab}_c \frac{J^c(w)}{z-w}. \tag{110}$$

Additionally there are odd Virasoro primaries $G^{\alpha A}(z)$ of spin 3/2 transforming in the fundamental of the $SU(2)$ current algebra which have self-OPE's:

$$G^{\alpha A}(z)G^{\beta B}(w) \sim -\epsilon^{AB}\epsilon^{\alpha\beta}\frac{T(w)}{z-w} - \frac{c}{3}\frac{\epsilon^{AB}\epsilon^{\alpha\beta}}{(z-w)^3} + \epsilon^{AB}\epsilon^{\beta\gamma}(\sigma^a)^\alpha_\gamma\left(\frac{2J^a(w)}{(z-w)^2} + \frac{\partial J^a(w)}{z-w}\right). \tag{111}$$

Above, we have written $SU(2)_a \times SU(2)_b$ doublet indices as $A, \dot{B}$ and $SU(2)_L \times SU(2)_R$ doublet indices as $\alpha, \dot{\beta}$.[20]

As mentioned earlier, it is difficult to perform explicit computations in the holomorphic twist beyond a local (flat space) model, even for a single copy of $Y$. Rather than try to work with the full chiral de Rham complex directly, we will outline the matching of (counts of) states between twisted supergravity and twisted CFT (via the elliptic genus). Then we will turn to the determination of the OPEs in the holomorphically twisted theory in the $N \to \infty$ limit by applying Koszul duality to our twisted supergravity theory; as a sanity check, we will easily recover the $\mathcal{N} = 4$ superconformal algebra and its $\mathfrak{psu}(1,1|2)$ global subalgebra[21]

Consider the chiral half of the $\mathcal{N} = (4,4)$ $\sigma$-model on the symmetric orbifold $\mathrm{Sym}^N Y$ where $Y$ is $T^4$ or a $K3$ surface. After performing the half-twist, this is all that remains of the supersymmetric $\sigma$-model. According to [37] we can regard the direct sum of the vacuum modules of the chiral algebras of $\mathrm{Sym}^N Y$, for each $N$, as being itself a Fock space. The generators of this Fock space are given by the single string states. These single string states are the analog of single trace operators in a gauge theory, and can be matched with single-particle states in the holographic dual. Let $c(n,m)$ be the super-dimension of the space of operators in supersymmetric $\sigma$-model into $Y$, which are of weight $n$ under $L_0$ and of weight $m$ under the action of the Cartan of $SU(2)_R$. Let $q, y$ be fugacities for $L_0$ and the Cartan of $SU(2)_R$, respectively—the elliptic genus $\chi(Y; q, y)$ is a series in these variables. Of course, for $Y = T^4$ the elliptic genus vanishes,[22] so we will now fix $Y = K3$.

Introducing another parameter $p$, which keeps track of the symmetric power, we can consider the generating series

$$\sum_{n \geq 0} p^n \chi(\mathrm{Sym}^n Y; q, y). \tag{112}$$

The main result of [11, 37] is an expression for this generating series

$$\sum_n p^n \chi(\mathrm{Sym}^n Y; q, y) = \prod_{l, m \geq 0, n > 0} \frac{1}{(1 - p^n q^m y^l)^{c(nm, l)}}, \tag{113}$$

---

[20]There is, of course, also a right-moving copy in the full SCFT, though only the chiral half above will be accessible in the holomorphic twist.

[21]More precisely, we will find $\mathfrak{psl}(1,1|2)$; for example, the $SU(2)$ Kac-Moody algebra using Koszul duality will naturally appear in the Cartan-Weyl basis.

[22]One could instead consider the modified elliptic genus for $T^4$, which is enriched with additional insertions of the fermion number operator to absorb the fermionic zero modes.

where $c(m, l)$ is a function of the quantity $4m - l^2$. In other words, we can interpret the direct sum of the vacuum modules of the $\mathrm{Sym}^n Y$ $\sigma$-models as being the Fock space generated by a trigraded super-vector space

$$V = \oplus_{n \geq 0, m, l} V_{n, m, l}, \tag{114}$$

where the super-dimension of $V_{n, m, l}$ is $c(nm, l)$.

The generating function of elliptic genera of $\mathrm{Sym}^N Y$ decomposes as

$$\sum_{N \geq 0} p^N \chi(\mathrm{Sym}^N Y; q, y) = \prod_{n > 0} \sum_{N \geq 0} p^{nN} \chi(\mathrm{Sym}^N \mathcal{H}_{(n)}^{\mathbf{Z}_n}; q, y), \tag{115}$$

with $\sum_{N \geq 0} p^{nN} \chi(\mathrm{Sym}^N \mathcal{H}_{(n)}^{\mathbf{Z}_n}; q, y) = \prod_{l, m \geq 0} \frac{1}{(1 - p q^m y^l)^{c(mn, l)}}$. Here, $\mathcal{H}_{(n)}$ is the Hilbert space of a single long string on $Y$ of length $n$ with winding number $1/n$.

We can extract the $N \to \infty$ limit of this expression, following the logic employed in [11, 38, 39], particularly [39]. First, in preparation for comparison to supergravity, we perform spectral flow[23] to the NS sector:

$$\begin{aligned}
\sum_{N \geq 0} p^N \chi_{NS}(\mathrm{Sym}^N Y; q, y) &= \sum_{N \geq 0} p^N \chi(\mathrm{Sym}^N Y; q, y\sqrt{q}) y^N q^{N/2} \\
&= \prod_{\substack{n \geq 0 \\ m \geq 0, m \in \mathbf{Z} \\ l \in \mathbf{Z}}} \frac{1}{(1 - p^n q^{m + l/2 + n/2} y^{l + n})^{c(nm, l)}} \\
&= \prod_{\substack{n \geq 0 \\ m' \geq |l'|/2, \, 2m' \in \mathbf{Z}_{\geq 0} \\ l' \in \mathbf{Z}, \, m' - l'/2 \in \mathbf{Z}_{\geq 0}}} \frac{1}{(1 - p^n q^{m'} y^{l'})^{c(nm' - nl'/2, n - l')}}.
\end{aligned}$$

At any power of $q$, there will be contributions from terms of the form $\frac{1}{(1 - p y^{l'})^{c(-l'/2, l' - 1)}}$. The only nonvanishing such term in our case when $m' = 0$ is $\frac{1}{(1 - p)^2}$. We wish to isolate the coefficients of all terms of the form $q^a y^b p^N$ for $a \ll N$. Taylor expanding $\frac{1}{(1 - p)^2}$ and extracting the desired coefficient gives $Nh(a, b) + \mathcal{O}(N^0)$ where $h(a, b)$ is the coefficient of $q^a y^b$ in

$$\prod_{\substack{m' \geq |l'|/2, \, 2m' \in \mathbf{Z}_{\geq 0} \\ l' \in \mathbf{Z}, \, m' - l'/2 \in \mathbf{Z}_{\geq 0}}} \frac{1}{(1 - q^{m'} y^{l'})^{f(m', l')}},$$

with $f(m', l') := \sum_{n > 0} c(n(m' - l'/2), l' - n)$. The coefficients $c(M, L)$ vanish for $4M - L^2 < -1$ so for $m' \geq 1$ the sum truncates to $f(m', l') = \sum_{n=1}^{4m'} c(n(m' - l'/2), l' - n)$.

Hence, we can get a finite contribution upon dividing by $N$.

We can also write out the non-vanishing $f(m', l')$ more explicitly, recalling that the coefficients are constrained to lie in the following range of the Jacobi variable: $-2m' \leq l' \leq 2m', l' \equiv 2m' \mod 2$. Reproducing the elementary manipulations in Appendix A of [39] (in particular, using the fact that $c(N, L)$ depends only on $4N - L^2$ and $L \mod 2$) allows us to rewrite the sum as

$$f(m', l') = \left( \sum_{\tilde{n} \in \mathbf{Z}} c(m'^2 - l'^2/4, \tilde{n}) \right) - c(0, l'), \tag{116}$$

where $n' := n - 2m$ in the first term. The first term is non-vanishing only when $l' = \pm 2m'$ and then it reduces to the Witten index of K3, i.e. $f(m', \pm 2m') = 24$ for general $m'$. Otherwise,

---

[23]We shift the overall power of $q$ by $q^{c/24}$ so that the vacuum occurs at $q^0$.

we have $f(m', l') = -c(0, l')$. When $m' \in \mathbb{Z}$ the nonvanishing such term is $-c(0,0) = -20$, and when $m' \in \mathbf{Z} + 1'/2$ we have $-c(0, 1) = -2$ and $-c(0, -1) = -2$.

In sum, we obtain

$$\lim_{N \to \infty} \frac{\chi_{NS}(\text{Sym}^N Y; q, y)}{N} = \prod_{k \geq 1} \frac{(1 - q^k)^{20}(1 - q^{k-1/2}y^{-1})^2(1 - q^{k-1/2}y)^2}{(1 - q^{k/2}y^k)^{24}(1 - q^{k/2}y^{-k})^{24}} \tag{117}$$

$$= 1 + \left(\frac{22}{y} + 22y\right)q^{1/2} + \left(\frac{277}{y^2} + 464 + 277y^2\right)q + \text{O}(q^{3/2}). \tag{118}$$

We will denote this large $N$ limit by $\chi_{NS}(\text{Sym}^\infty Y; q, y)$. In particular, for there are two bosonic towers corresponding to (anti)chiral primary states and three fermionic towers corresponding to (derivatives of) the states capturing the cohomology of a single copy of K3.

We observe that this expression for the large $N$ limit of the elliptic genus agrees exactly with the plethystic exponential of the single particle twisted supergravity index we computed in (97). One can easily see this by using the definition of the plethystic exponential

$$\text{PE}[f](q, y) = \exp\left(\sum_{k=1}^\infty \frac{f(q^k, y^k)}{k}\right), \tag{119}$$

and rewriting the infinite-N elliptic genus as $\text{PE}[f_{CFT}](q, y)$ in terms of the function

$$f_{CFT}(q, y) = \sum_{m=1}^\infty 24(q^{1/2}y)^m + 24(q^{1/2}y^{-1})^m - 20q^m - 2q^{m-1/2}y - 2q^{m-1/2}y^{-1}, \tag{120}$$

which can be immediately matched with $\text{PE}[f_{sugra}](q, y)$, as expected.

For a finite number of branes we have given a microscopic description of the twisted $D1/D5$ system in flat space as an explicit BRST theory and matched with the description in [26]. In the large $N$ limit, the states of a general BRST model can be described in terms of the Loday–Quillen–Tsygan theorem; see the recent work [2, 24, 40, 41]. It would be interesting to apply this theorem to understand the states of this model in the large $N$ limit and to reproduce the elliptic genus. It is easier to perform LQT for the $T^4$ case and enumerate the non-vanishing states, and it would be interesting to match this explicitly to the results of [3]. In the case of a $K3$ surface it is not yet clear how to apply this theorem to understand the large $N$ limit of the CFT.

## 5 Tree-level OPEs

In this section we initiate our computation of planar OPEs of the chiral algebra, using the same Koszul duality techniques as in [3] (to which we refer for a more complete discussion), by first considering contributions from tree diagrams. Tree diagrams, as we will see, correspond to the twisted open-closed string theory in flat space (i.e. before considering the backreaction of the D-branes). We will first recall the Koszul duality approach to twisted holography pioneered in [3, 7] (see [8] for a physical review of Koszul duality).[24]

Koszul duality enables us to *derive* the planar chiral algebra from our knowledge of the twisted supergravity dual. In this way, Koszul duality provides a way to extract twisted CFT data, encoded in the technically challenging chiral de Rham complex, using more tractable

---

[24]See also [42–44] for more on Koszul duality in twisted holography, and [45–48] for additional, closely related twisted holographic explorations.

supergravity computations.[25] The method is to write down the most general possible bulk-brane coupling and compute the BRST variation of all possible bulk-boundary (or Witten-like) diagrams order by order in perturbation theory. In this work, we will focus only on the diagrams that contribute in the $N \to \infty$ limit. Demanding that the sum of the BRST variations of all contributing diagrams at a given order vanish results in constraints on the operator product of the local operators on the brane worldvolume; these operators generate the chiral algebra of the twisted SCFT, and so Koszul duality directly extracts their OPEs.

We begin on flat space. In the next section, we will incorporate planar diagrams encoding the backreaction of the D1-D5 system. These are the diagrams responsible for deforming the initial flat space geometry to the K3 conifold. It was explained in [3] that, strikingly, only a finite number of such backreaction diagrams contribute at each order in the $\frac{1}{\sqrt{N}}$ expansion. Typically, one would have to re-sum an infinite number of such diagrams to obtain the deformed geometry. This simplification allows us to derive the chiral algebra as a deformation around flat space, using the perturbative, Feynman diagrammatic approach of Koszul duality. In particular, the complete, backreacted planar chiral algebra we will compute in the next two sections has the global subalgebra we derived from a different point of view in §3.4.

On flat space, we use holomorphic coordinates $Z = (z, w_1, w_2)$ on $\mathbf{C}^3$ where the system of branes wraps the locus $\{w_i = 0\}$. We will call the brane locus the support of the "defect chiral algebra", following the perspective of the Koszul dual chiral algebra as the universal defect algebra to which Kodaira-Spencer theory can couple in a gauge-anomaly-free manner [8]. (In other words, any other defect chiral algebra one might wish to couple to Kodaira-Spencer theory, such as an appropriate number of free chiral fermions, must furnish a representation for the Koszul dual/universal defect algebra.).

The Beltrami field $\mu$ has three holomorphic vector components that we denote by

$$\mu = \mu_z \partial_z + \mu_1 \partial_{w_1} + \mu_2 \partial_{w_2}, \tag{121}$$

where $\mu_z, \mu_i \in \Omega^{0,\bullet}(\mathbf{C}^3)$ are Dolbeault forms (recall that the ghost number zero fields arise from forms of degree $(0,1)$—so, actual Beltrami differentials). With this notation, the full classical interaction of our compactified Kodaira–Spencer theory is

$$\int_{\mathbf{C}^3} \mu_1 \mu_2 \mu_z|_{\eta\overline{\eta}} \, \mathrm{d}^3 Z + \int_{\mathbf{C}^3} \alpha \mu_i \partial_{w_i} \gamma|_{\eta\overline{\eta}} \, \mathrm{d}^3 Z + \int_{\mathbf{C}^3} \alpha \mu_z \partial_z \gamma|_{\eta\overline{\eta}} \, \mathrm{d}^3 Z. \tag{122}$$

Recall that for the kinetic part of the action for Kodaira–Spencer theory to be well-defined we must impose the following constraint on the field $\mu$:

$$\partial_z \mu_z + \partial_{w_i} \mu_i = 0. \tag{123}$$

Before moving into computations, we describe the operators present in the defect chiral algebra. In what follows we use the notation $D_{r,s}$ to denote the holomorphic differential operator

$$D_{r,s} = \frac{1}{r!} \frac{1}{s!} \partial_{w_1}^r \partial_{w_2}^s,$$

where the holomorphic derivatives point transversely to the brane. To simplify formulas we will use the notations

$$\int_{z,\eta} \omega = \int_{z \in \mathbf{C}_z} \omega|_{\eta\overline{\eta}} \, \mathrm{d}z, \tag{124}$$

---

[25]A complementary approach, compatible with a topological (as opposed to holomorphic) twist, is to study the rings of chiral primaries in symmetric orbifold theories [49–51] Chiral primaries are 1/2-BPS states, comprised of short multiplets with respect to both the holomorphic and anti-holomorphic $\mathcal{N} = 4$ algebras, and have nonsingular OPEs with one another. Koszul duality is sensitive to 1/4-BPS states but only captures the (purely holomorphic) singular terms of the chiral algebra OPEs. It would be interesting to reproduce (the holomorphic halves of) the chiral ring structure coefficients from our Kodaira-Spencer theory.

for integrals along the defect, and

$$\int_{Z,\eta} \omega = \int_{Z \in \mathbf{C}^3} \omega|_{\eta\overline{\eta}} \, \mathrm{d}^3 Z \,, \tag{125}$$

for integrals in the bulk.

As with the fields of our extended version of Kodaira–Spencer theory, the defect operators of the chiral algebra will all be polynomials in the variables $\eta$ parameterizing the cohomology of the $K3$ surface. The variables $\eta$ do not carry spin, parity, or ghost degree (this is one difference with the case of the complex torus $T^4$). For simplicity of notation we will not explicitly include this $\eta$-dependence until it is convenient.

Defect operators sourced by bulk fields *before* imposing the constraint (123) can be described as follows:

1. Bosonic Virasoro primaries $\widetilde{T}[r,s]$ of holomorphic conformal weight (i.e. "spin") $2 + r/2 + s/2$ which couple to the field $\mu_z$ by

$$\int_{z,\eta} \widetilde{T}[r,s] D_{r,s}\mu_z|_{w=0} \,. \tag{126}$$

2. Bosonic Virasoro primaries $\widetilde{J}^i[r,s]$, $i = 1,2$ of weight $1/2 + r/2 + s/2$ which couple to the fields $\mu_i$ by

$$\int_{z,\eta} \widetilde{J}^i[r,s] D_{r,s}\mu_i|_{w=0} \,. \tag{127}$$

3. Fermionic Virasoro primaries $G_\alpha[r,s]$, $G_\gamma[r,s]$ of weight $1 + r/2 + s/2$ which couple to the fields $\alpha, \gamma$ by

$$\int_{z,\eta} G_\alpha[r,s] D_{r,s}\alpha|_{w=0} \,, \qquad \int_{z,\eta} G_\gamma[r,s] D_{r,s}\gamma|_{w=0} \,. \tag{128}$$

The fermionic operators $G_\alpha, G_\gamma$ couple to unconstrained fields of the theory on $\mathbf{C}^3$. On the other hand, $\widetilde{T}, \widetilde{J}^i$ couple to the fields $\mu_z, \mu_i$ satisfying the divergence-free constraint (123). Only some combination of these operators will couple to the on-shell fields of the theory on $\mathbf{C}^3$. Explicitly, the constrained fields source the following defect operators

$$T[r,s] \stackrel{\text{def}}{=} \widetilde{T}[r,s] - \frac{1}{2(r+1)}\partial_z \widetilde{J}^1[r+1,s] - \frac{1}{2(s+1)}\partial_z \widetilde{J}^2[r,s+1], \quad r+s \geq 0,$$
$$J[k,l] \stackrel{\text{def}}{=} k\widetilde{J}^2[k-1,l] - s\widetilde{J}^1[k,l-1], \qquad\qquad\qquad k+l \geq 1. \tag{129}$$

We see that $T[r,s]$ has weight $2 + (r+s)/2$ and $J[k,l]$ has weight $(k+l)/2$ and live in the $SU(2)_R$ spin representation $(k+l)/2$.

As stated above, all operators are valued in the ring $R$ which in the case of compactification of a K3 surface is $R = H^\bullet(K3)$. It is convenient to expand operators in the fermionic-Fourier-dual variables $\widehat{\eta}$. If $\mathcal{O} = \mathcal{O}(\eta)$ is any of the operators defined above, then the Fourier-dual expansion is defined formally as

$$\mathcal{O}(\widehat{\eta}) = e^{\eta\widehat{\eta}} \mathcal{O}(\eta)|_{\eta\overline{\eta}} \,, \tag{130}$$

with a similar formula valid in the case of an arbitrary ring $R$ with trace. We will expand the OPEs that follow in this Fourier dual coordinate. Explicitly, if

$$\mathcal{O}(\eta) = \mathcal{O} + \mathcal{O}_\eta \eta + \mathcal{O}_{\widehat{\eta}}\widehat{\eta} + \mathcal{O}_{\eta_a}\eta_a + \mathcal{O}_{\eta\overline{\eta}}\eta\overline{\eta} \,, \tag{131}$$

then

$$\mathcal{O}(\widehat{\eta}) = \mathcal{O}_{\eta\overline{\eta}} + \widehat{\eta}\mathcal{O}_{\overline{\eta}} + \widehat{\overline{\eta}}\mathcal{O}_\eta + h_{ab}\mathcal{O}_{\eta_a}\widehat{\eta}_b + \mathcal{O}\widehat{\overline{\eta}}\widehat{\eta} \,. \tag{132}$$

## 5.1 $\widetilde{J}\widetilde{J}$ OPE

We first compute the OPE of the off-shell operators $\widetilde{J}^i[r,s]$ and then impose constraints to determine the OPE of the on-shell operators $J[r,s]$.

### 5.1.1

The coefficient of $\widetilde{J}^1[k,l]$ in the OPE will be determined by the terms in the BRST variation of $\mu_1$ which involve $\mathfrak{c}_1$ and $\mu_1$, $\mathfrak{c}_1$ and $\mu_2$, or $\mathfrak{c}_2$ and $\mu_1$.

Consider the gauge variation of

$$\int_{z,\eta} \widetilde{J}^1[r,s](z)D_{r,s}\mu_1 \,. \tag{133}$$

The gauge variation of $\mu_1$ is

$$Q\mu_1 = \overline{\partial}\mathfrak{c}_1 + \mu_i\partial_{w_i}\mathfrak{c}_1 + \mu_z\partial_z\mathfrak{c}_1 - \mathfrak{c}_i\partial_{w_i}\mu_1 - \mathfrak{c}_z\partial_z\mu_1$$
$$+ \partial_{w_2}\mathfrak{c}_\gamma\partial_z\alpha - \partial_z\mathfrak{c}_\gamma\partial_{w_2}\alpha + \partial_{w_2}\mathfrak{c}_\alpha\partial_z\gamma - \partial_z\mathfrak{c}_\alpha\partial_{w_2}\gamma \,.$$

For now, we can disregard the terms involving $\mathfrak{c}_\gamma$ and $\alpha$ or $\mathfrak{c}_\alpha$ and $\gamma$. These will play a role later on when we constrain the OPEs involving the operators $G_\alpha, G_\gamma$.

Inserting this gauge variation into the coupling to $\widetilde{J}^i[r,s]$, we see that the first term, $\overline{\partial}\mathfrak{c}_1$, vanishes by integration by parts. Cancellation of the remaining terms will give us constraints on the OPE coefficients. The remaining terms are

$$\int_z \widetilde{J}^1[r,s](z)D_{r,s}\left(\mu_i\partial_{w_i}\mathfrak{c}_1 + \mu_z\partial_z\mathfrak{c}_1 - \mathfrak{c}_i\partial_{w_i}\mu_1 - \mathfrak{c}_z\partial_z\mu_1\right)(z,w_i=0,\eta_a) \,.$$

Let us focus on the term in this expression which involves the fields $\mu_1$ and $\mathfrak{c}_1$. This is

$$\int_{z,\eta} \widetilde{J}^1[r,s](z)D_{r,s}\left(\mu_1\partial_{w_1}\mathfrak{c}_1 - \mathfrak{c}_1\partial_{w_1}\mu_1\right) \,.$$

Because this expression involves both $\mathfrak{c}_1$ and $\mu_1$, which are fields (and a corresponding ghost) that couple to $\widetilde{J}^1$, we find that it can only be cancelled by a gauge variation of an integral involving two copies of the operators $\widetilde{J}^1$, at separate points $z,z'$:

$$\tfrac{1}{2}\int_{z,z',\eta,\eta'} \widetilde{J}^1[k,l](z,\eta)D_{k,l}\mu_1(z,w=0,\eta)\widetilde{J}^1[r,s](z',\eta')D_{r,s}\mu_1(z',w'=0,\eta') \,.$$

Applying the gauge variation of $\mu_1$ to this expression, and retaining only the terms involving $\overline{\partial}\mathfrak{c}_1$, gives us

$$\int_{z,z',\eta,\eta'} \widetilde{J}^1[k,l](z,\eta)D_{k,l}\mu_1(z,w=0,\eta)\widetilde{J}^1[r,s](z',\eta')D_{r,s}\overline{\partial}\mathfrak{c}_1(z',w'=0,\eta') \,.$$

Here the $\overline{\partial}$ operator only involves the $z$-component because restricting to $w_i=0$ sets any $d\overline{w}_i$ to zero. We can integrate by parts to move the location of the $\overline{\partial}$ operator. Every field $\mu_i$ contains a $d\overline{z}$, as otherwise it would restrict to zero at $w=0$, so that $\partial_{\overline{z}}\mu_i = 0$.

This analysis shows that in order for the anomaly to cancel we must require

$$\int_{z,z',\eta,\eta'} \overline{\partial}_{\overline{z}}\left(\widetilde{J}^1[k,l](z,\eta)\widetilde{J}^1[r,s](z',\eta')\right)D_{m,n}\mu_1(z,w=0,\eta)D_{r,s}\mathfrak{c}_1(z',w'=0,\eta')$$

$$= \int_{z'',\eta''} \widetilde{J}^1[m,n](z'',\eta'')D_{m,n}\left(\mu_1\partial_{w_1}\mathfrak{c}_1 - \mathfrak{c}_1\partial_{w_1}\mu_1\right)(z'',w=0,\eta'') \,. \tag{134}$$

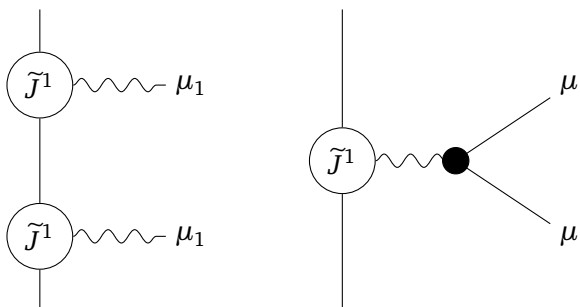

Figure 1: Cancellation of the gauge anomaly of these two diagrams leads to the equation for the self OPE of the currents $\widetilde{J}^1[k,l]$.

In these expressions, we sum over the indices $r, s, k, l, m, n$. This equation must hold for all values of the field $\mu_1, \mathfrak{c}_1$. To constrain the OPEs, we substitute the test fields

$$\mu_1 = G(z, \overline{z}, \eta) \mathrm{d}\overline{z} w_1^k w_2^l,$$
$$\mathfrak{c}_1 = H(z, \overline{z}, \eta) w_1^r w_2^s,$$

for $G, H$ arbitrary smooth functions of the variables $z, \overline{z}, \eta_a$.

Inserting these values for the fields into the anomaly-cancellation condition gives

$$\int_{z,z',\eta,\eta'} \overline{\partial}_{\overline{z}} \left( \widetilde{J}^1[k,l](z,\eta) \widetilde{J}^1[r,s](z',\eta') \right) G(z,\overline{z},\eta) H(z',\overline{z}',\eta')$$
$$= \int_{z'',\eta''} (r-k) \widetilde{J}^1[k+r-1,l+s](z'',\eta'') G(z'',\overline{z}'',\eta'') H(z'',\overline{z}'',\eta''). \quad (135)$$

Since this must hold for all values of the functions $G, H$ we get an identity of the integrands:

$$\overline{\partial}_{\overline{z}} \left( \widetilde{J}^1[k,l](z,\eta) \widetilde{J}^1[r,s](z',\eta') \right) = \delta_{z=z'} \delta_{\eta=\eta'} (r-m) \widetilde{J}^1[k+r-1,l+s].$$

The formal $\delta$-function $\delta_{\eta=\eta'}$, in the case $R = H^\bullet(K3)$, has the simple expression

$$\delta_{\eta=\eta'} = 1 \otimes \eta' \overline{\eta}' + \eta \otimes \overline{\eta}' + \overline{\eta} \otimes \eta' + h^{ab} \eta_a \otimes \eta'_b + (\boldsymbol{\eta} \leftrightarrow \boldsymbol{\eta}') + \eta \overline{\eta} \otimes 1'. \quad (136)$$

Anomaly cancellation leads to the OPE:

$$\widetilde{J}^1[k,l](0,\eta) \widetilde{J}^1[r,s](z,\eta') \simeq \frac{1}{z}(r-k) \widetilde{J}^1[k+r-1,l+s](0,\eta) \delta_{\eta=\eta'}. \quad (137)$$

We apply the formal Fourier transform to write this expression in terms of the operators $\widetilde{J}^1[k,l](0,\widehat{\eta})$. We find

$$\widetilde{J}^1[k,l](0,\widehat{\eta}) \widetilde{J}^1[r,s](z,\widehat{\eta}') \simeq \frac{1}{z}(r-k) \widetilde{J}^1[k+r-1,l+s](0,\widehat{\eta}+\widehat{\eta}'). \quad (138)$$

To simplify notation we will write this OPE in a way that does not explicitly refer to the $\eta$-variables as in:

$$\widetilde{J}^1[k,l](0) \widetilde{J}^1[r,s](z) \simeq \frac{1}{z}(r-k) \widetilde{J}^1[k+r-1,l+s]. \quad (139)$$

Diagrammatically, the OPE we have just deduced follows from the cancellation of the gauge anomaly in Figure 1.

### 5.1.2

Similar computations lead to the following tree-level OPEs. We have the $\widetilde{J}^2\widetilde{J}^2$ OPE:

$$\widetilde{J}^2[r,s](0)\widetilde{J}^2[k,l](z) \simeq \frac{1}{z}(l-s)\widetilde{J}^2[r+k,s+l-1](0).$$

The $\widetilde{J}^1\widetilde{J}^2$ OPE:

$$\widetilde{J}^1[r,s](0)\widetilde{J}^2[k,l](z) \simeq -\frac{1}{z}s\widetilde{J}^1[r+k,l+s-1](0) + \frac{1}{z}k\widetilde{J}^2[k+r-1,l+s](0).$$

And finally, the $\widetilde{J}^2\widetilde{J}^1$ OPE:

$$\widetilde{J}^2[r,s](0)\widetilde{J}^1[k,l](z) \simeq -\frac{1}{z}r\widetilde{J}^2[r+k-1,l+s](0) + \frac{1}{z}l\widetilde{J}^1[k+r,l+s-1](0).$$

### 5.1.3

The calculations so far have involved the OPEs of the "off-shell" operators $\widetilde{J}^i[r,s]$. To obtain the on-shell OPEs we apply the constraints in (129), which for the $J$-type operators takes the form

$$J[r,s] = r\widetilde{J}^2[r-1,s] - s\widetilde{J}^1[r,s-1]. \tag{140}$$

We find

$$\begin{aligned} J[r,s](0)J[k,l](z) = &\frac{1}{z}(l-s)kr\widetilde{J}^2[k+r-2,l+s-1] + \frac{1}{z}ls(k-r)\widetilde{J}^1[k+r-1,l+s-2] \\ &+ \frac{1}{z}r(r-1)l\widetilde{J}^2[r+k-2,l+s-1] - \frac{1}{z}l(l-1)r\widetilde{J}^1[k+r-1,l+s-2] \\ &+ \frac{1}{z}ks(s-1)\widetilde{J}^1[r+k-1,l+s-2] - \frac{1}{z}ks(k-1)\widetilde{J}^2[k+r-2,l+s-1]. \end{aligned} \tag{141}$$

Collecting the terms, we find the right hand side is

$$\begin{aligned} &\frac{1}{z}\big((l-s)kr + r(r-1)l - ks(k-1)\big)\widetilde{J}^2[k+r-2,l+s-1] \\ &\quad + \frac{1}{z}\big(ls(k-r) - l(l-1)r + ks(s-1)\big)\widetilde{J}^1[k+r-1,l+s-2]. \end{aligned}$$

Finally, using (140) we find that the OPE involving the on-shell operators $J[r,s]$ takes the form

$$J[r,s](0)J[k,l](z) = \frac{1}{z}(rl-ks)J[r+k-1,l+s-1](0).$$

As above, on the right hand side all operators are evaluated at $z=0$ and with the fermionic variables $\widehat{\eta} + \widehat{\eta}'$. Note that the operators $J[r,s]$ with $r+s=2$ which are independent of $\widehat{\eta}$ satisfy the OPE of the $\mathfrak{su}(2)$ Kac-Moody algebra at level zero. We will get a nontrivial level once we include the contribution from the backreaction, which we do in §6.

The OPEs described above lead to a mode algebra that is easy to describe and interpret. Let the $n$th mode of $J[r,s]$ be

$$J[r,s]_n \overset{\text{def}}{=} \oint dz\, z^{-n-1+(r+s)/2}J[r,s](z). \tag{142}$$

The OPEs above lead to the relation

$$[J[r,s]_n, J[r',s']_{n'}] = (sr'-rs')J[r+r'-1,s+s'-1]_{n+n'}, \tag{143}$$

which we can interpret geometrically as follows.

In the case that the hyperKähler surface on which we compactify type IIB supergravity is $T^4$ it is shown in [3] that the mode algebra corresponding to this full collection of OPEs of the $J$-operators can be expressed as the super loop space of the Lie algebra $\mathfrak{w}_\infty$ of Hamiltonian vector fields on $\mathbf{C}^2$.[26] This is the Lie algebra $\mathcal{L}^{1|4}\mathfrak{w}_\infty$ whose elements have the form

$$z^n f(w_1, w_2; \eta_a), \tag{144}$$

for $n \in \mathbf{Z}$, where $f(w_1, w_2; \eta_a) \in \mathbf{C}[w_1, w_2]/\mathbf{C} \otimes \mathbf{C}[\eta_a]$. (Here, the $\eta_a, a = 1, 2, 3, 4$ variables generate the cohomology of $T^4$, and are therefore fermionic.) The super bracket is

$$[z^n f, z^m g] = z^{n+m} \epsilon^{ij} \partial_{w_i} f \, \partial_{w_j} g \,. \tag{145}$$

More generally if $H^\bullet(T^4)$ is replaced by an arbitrary super ring $R$, the mode algebra of the $J[r, s]$-operators gives rise to a similar infinite-dimensional Lie superalgebra that we denote $L^R\mathfrak{w}_\infty$. Elements in this Lie algebra have the form

$$z^n f(w_1, w_2; \eta), \tag{146}$$

where $n \in \mathbf{Z}$ and $f \in \mathbf{C}[w_1, w_2]/\mathbf{C} \otimes R$. The bracket (before taking into account the backreaction) is identical to (145) and simply utilizes the commutative product on $R$. In the case of compactifying twisted IIB supergravity along a K3 surface we simply take $R = H^\bullet(K3)$.

If $R = \mathbf{C}$, then $L^{\mathbf{C}}\mathfrak{w}_\infty = L\mathfrak{w}_\infty$ is the Lie algebra of symmetries of $\mathbf{C}^2 \times \mathbf{C}^\times$ viewed as a bundle over $\mathbf{C}^\times$ with fibers the holomorphic symplectic manifold $\mathbf{C}^2$. More generally, $L^R\mathfrak{w}_\infty$ is the Lie algebra of symmetries of $\mathbf{C}^2 \times \mathbf{C}^\times \times \operatorname{Spec} R$ thought of as a bundle over $\mathbf{C}^\times \times \operatorname{Spec} R$.

In the next section we will see how the backreaction introduces additional terms (such as a central extension) in the bracket (145).

## 5.2 $TJ$ OPE

We turn to the tree-level OPE between the on-shell operators $T$ and $J$. First, we compute the tree-level OPE between the off-shell operators $\widetilde{J}$ and $\widetilde{T}$.

The coefficient of $\widetilde{J}^1$, for instance, in this OPE will be determined by the terms in the BRST variation of $\mu_1$ which involve $\mathfrak{c}_1$ and $\mu_z$ or $\mathfrak{c}_z$ and $\mu_1$. We collect such terms in the gauge variation of (133) and

$$\int_{(z, \eta_a) \in \mathbf{C}^{1|4}} \widetilde{T}[m, n](z, \eta_a) D_{m,n} \mu_z(z, w_i = 0, \eta_a) \,. \tag{147}$$

Recall that the gauge variation of $\mu_z$ is

$$Q\mu_z = \overline{\partial}\mathfrak{c}_z + \mu_i \partial_{w_i}\mathfrak{c}_z + \mu_z \partial \mathfrak{c}_z - \mathfrak{c}_i \partial_{w_i}\mu_z - \mathfrak{c}_z \partial_z \mu_z - \epsilon_{ij}\partial_i \mathfrak{c}_\gamma \partial_j \alpha - \epsilon_{ij}\partial_i \mathfrak{c}_\alpha \partial_j \gamma \,.$$

For now, we can disregard the terms involving $\alpha$ and $\mathfrak{c}_\gamma$ or $\mathfrak{c}_\alpha$ and $\gamma$.

The terms in the variations of (133) and (147) involving $\mathfrak{c}_1$ and $\mu_z$ or $\mathfrak{c}_z$ and $\mu_1$ is

$$\int_{z, \eta} \widetilde{J}^1[m, n](z, \eta_a) D_{m,n}(\mu_z \partial_z \mathfrak{c}_1 - \mathfrak{c}_z \partial_z \mu_1)(z, w_i = 0, \eta_a)$$

$$+ \int_{z, \eta} \widetilde{T}[m, n](z, \eta_a) D_{m,n}(\mu_1 \partial_{w_1}\mathfrak{c}_z - \mathfrak{c}_1 \partial_{w_1}\mu_z)(z, w_i = 0, \eta_a) \,.$$

---

[26] This is the quotient of the Lie algebra of functions on $\mathbf{C}^2$, which equipped with the standard Poisson bracket, by its center consisting of the constant functions.

The coefficient of $\mathfrak{c}_z$ can only be cancelled by a gauge variation of

$$\int_{z,z',\eta_a,\eta_a'} \widetilde{J}^1[r,s](z,\eta_a)D_{r,s}\mu_1(z,w_i=0,\eta_a)\widetilde{T}[k,l](z',\eta_a')D_{k,l}\mu_z(z',w_i'=0,\eta_a').$$

By similar manipulation as above, we find that the gauge variation of this expression is

$$\int_{z,z',\eta_a,\eta_a'} \overline{\partial}_z\left(\widetilde{J}^1[r,s](z,\eta_a)\widetilde{T}[k,l](z',\eta_a')\right)D_{r,s}\mathfrak{c}_1(z,w_i=0,\eta_a)D_{k,l}\mu_z(z',w_i'=0,\eta_a')$$

$$+ \int_{z,z',\eta_a,\eta_a'} \overline{\partial}_{z'}\left(\widetilde{J}^1[r,s](z,\eta_a)\widetilde{T}[k,l](z',\eta_a')\right)D_{r,s}\mu_1(z,w_i=0,\eta_a)D_{k,l}\mathfrak{c}_z(z',w_i'=0,\eta_a').$$

To constrain the OPEs, we use the test functions $\mu_z = 0$, $\mathfrak{c}_1 = 0$, $\mu_1 = G(z,\overline{z},\eta_a)\mathrm{d}\overline{z}w_1^k w_2^l$, $\mathfrak{c}_z = H(z,\overline{z},\eta_a)w_1^r w_2^s$ for $G,H$ arbitrary smooth functions of the variables $z,\overline{z},\eta_a$. This yields the anomaly cancellation condition

$$\int_{z,z',\eta_a,\eta_a'} \overline{\partial}_{z'}\left(\widetilde{J}^1[r,s](z,\eta_a)\widetilde{T}[k,l](z',\eta_a')\right)G(z,\overline{z},\eta_a)H(z',\overline{z}',\eta_a')$$

$$= -\int_{z'',\eta_a''} \widetilde{J}^1[r+k,s+l](z'',\eta_a'')H(z'',\overline{z}'',\eta_a'')\partial_{z''}G(z'',\overline{z}'',\eta_a'')$$

$$+ r\int_{z'',\eta_a''} \widetilde{T}[r+k-1,s+l](z'',\eta_a'')G(z'',\overline{z}'',\eta_a'')H(z'',\overline{z}'',\eta_a''). \tag{148}$$

Integrating the right hand side by parts gives us

$$\int_{z'',\eta_a''} \partial_{z''}\widetilde{J}^1[r+k,s+l](z'',\eta_a'')H(z'',\overline{z}'',\eta_a'')G(z'',\overline{z}'',\eta_a'')$$

$$+ \int_{z'',\eta_a''} \widetilde{J}^1[r+k,s+l](z'',\eta_a'')\partial_{z''}H(z'',\overline{z}'',\eta_a'')G(z'',\overline{z}'',\eta_a'')$$

$$+ r\int_{z'',\eta_a''} \widetilde{T}[r+k-1,s+l](z'',\eta_a'')G(z'',\overline{z}'',\eta_a'')H(z'',\overline{z}'',\eta_a''). \tag{149}$$

Because $G,H$ are arbitrary functions, we arrive at the OPE

$$\widetilde{T}[r,s](0,\eta_a)\widetilde{J}^1[k,l](z,\eta_a') \simeq \delta_{\eta_a=\eta_a'}\frac{1}{z}\partial_z\widetilde{J}^1[r+k,s+l](0,\eta_a)$$

$$+ \delta_{\eta_a=\eta_a'}\frac{1}{z^2}\widetilde{J}^1[r+k,s+l](0,\eta_a)$$

$$+ r\delta_{\eta_a=\eta_a'}\widetilde{T}[r+k-1,s+l](0,\eta_a). \tag{150}$$

Switching the $\eta_a$ variables to $\widehat{\eta}^a$ variables by applying the odd Fourier transform we can write this OPE as

$$\widetilde{T}[r,s](0,\widehat{\eta}^a)\widetilde{J}^1[k,l](z,\widehat{\eta}'^a) \simeq \frac{1}{z}\partial_z\widetilde{J}^1[r+k,s+l](0,\widehat{\eta}^a+\widehat{\eta}'^a) + \frac{1}{z^2}\widetilde{J}^1[r+k,s+l](0,\widehat{\eta}^a+\widehat{\eta}'^a)$$

$$+ r\widetilde{T}[r+k-1,s+l](0,\widehat{\eta}^a+\widehat{\eta}'^a). \tag{151}$$

### 5.1.1

In a completely similar way one can deduce the $\widetilde{T}\widetilde{J}^2$ OPE

$$\widetilde{T}[r,s](0,\widehat{\eta}^a)\widetilde{J}^2[k,l](z,\widehat{\eta}'^a) \simeq \frac{1}{z}\partial_z\widetilde{J}^2[r+k,s+l](0,\widehat{\eta}^a+\widehat{\eta}'^a) + \frac{1}{z^2}\widetilde{J}^2[r+k,s+l](0,\widehat{\eta_a}+\widehat{\eta_a}')$$

$$+ s\widetilde{T}[r+k,s+l-1](0,\widehat{\eta}^a+\widehat{\eta}'^a). \tag{152}$$

### 5.1.2

Using the $\widetilde{T}\widetilde{J}^i$ and $\widetilde{J}^i\widetilde{J}^2$ OPEs that we have computed, we deduce the OPEs between the on-shell operators $T$ and $J^i$ using (129). After some algebraic manipulation, we find

$$
J[m,n](0)T[r,s](z) \simeq (nr-ms)\frac{1}{z}T[m+r-1,n+s-1](0)
$$
$$
+ \frac{1}{z^2}\left(\frac{m}{2(r+1)}+\frac{n}{2(s+1)}\right)J[m+r,n+s](0)
$$
$$
+ \frac{1}{2z}\left(\frac{m}{m+r}+\frac{n}{n+s}\right)\partial_z J[m+r,n+s](0). \tag{153}
$$

On the right hand side, all operators are evaluated at the variables $\widehat{\eta}+\widehat{\eta}'$. We have dropped this dependence for clarity.

### 5.3 $TT$ OPE

Following the same logic we constrain the $\widetilde{T}\widetilde{T}$ OPE. These OPEs are determined by terms in the BRST variation of $\mu_z$ which involve $c_z$ and $\mu_z$.

Proceeding as above we set

$$
\mu_z = G(z,\overline{z},\eta_a)\mathrm{d}\overline{z}w_1^k w_2^l,
$$
$$
\mathfrak{c}_1 = H(z,\overline{z},\eta_a)w_1^r w_2^s,
$$

to arrive at the anomaly constraint

$$
\int_{z,z',\eta_a,\eta_a'} \overline{\partial}_{z'}\left(\widetilde{T}[r,s](z,\eta_a)\widetilde{T}[k,l](z',\eta_a')\right)G(z,\overline{z},\eta_a)H(z',\overline{z}',\eta_a') \tag{154}
$$
$$
= \int_{z'',\eta_a''} \widetilde{T}[r+k,s+l](z'',\eta_a'')\left(G(z'',\overline{z}'',\eta_a'')\partial_{z''}H(z'',\overline{z}'',\eta_a'')-H(z'',\overline{z}'',\eta_a'')\partial_{z''}G(z'',\overline{z}'',\eta_a'')\right).
$$

Integrating by parts and switching to the Fourier dual odd coordinates, we find the OPE

$$
\widetilde{T}[r,s](0,\widehat{\eta}^a)\widetilde{T}[k,l](z,\widehat{\eta}'^a) \simeq \frac{1}{z}\partial_z\widetilde{T}[r+k,s+l](0,\widehat{\eta}^a+\widehat{\eta}'^a)+2\frac{1}{z^2}\widetilde{T}[r+k,s+l](0,\widehat{\eta}^a+\widehat{\eta}'^a). \tag{155}
$$

Using the $\widetilde{T}\widetilde{T}$ and $\widetilde{J}^i\widetilde{J}^j$ OPEs that we have computed, we deduce the OPEs between the on-shell operator $T$ and itself using (129). After some algebraic manipulation, we find

$$
T[m,n](0)T[r,s](z) \sim \frac{1}{z}\left(1+\frac{r}{2(m+1)}+\frac{s}{2(n+1)}\partial_z\right)T[m+r,n+s](0)
$$
$$
+ \frac{1}{z^2}\left(2+\frac{r}{2(m+1)}+\frac{s}{2(n+1)}+\frac{m}{2(r+1)}+\frac{n}{2(s+1)}\right)T[m+r,n+s](0)
$$
$$
+ \frac{1}{4z}\left(\frac{1}{(m+1)(n+s+1)}-\frac{1}{(n+1)(m+r+1)}\right)\partial_z^2 J[m+r+1,n+s+1](0)
$$
$$
+ \frac{1}{4z^2}\left(\frac{1}{(m+1)(s+1)}-\frac{1}{(n+1)(r+s)}\right)\partial_z J[m+r+1,n+s+1](0)
$$
$$
+ \frac{1}{4z^2}\left(\frac{1}{n+s+1}(\frac{2+m+r}{(1+m)(1+r)})-\frac{1}{m+r+1}(\frac{2+n+s}{(1+n)(1+s)})\right)
$$
$$
\times \partial_z J[m+r+1,n+s+1](0)
$$
$$
+ \frac{1}{2z^3}\left(\frac{1}{(m+1)(s+1)}-\frac{1}{(n+1)(r+s)}\right)J[m+r+1,n+s+1](0).
$$

On the right hand side, all operators are evaluated at the variables $\widehat{\eta}+\widehat{\eta}'$. We have dropped this dependence for clarity.

### 5.4 *GG* OPE

To constrain the $G_\alpha$, $G_\gamma$ OPE we consider terms in the gauge variations of the classical couplings involving $\alpha$ and $\mathfrak{c}_\gamma$ or $\gamma$ and $\mathfrak{c}_\alpha$ (we have disregarded those terms in the analysis above as they played no role in the previous OPE calculations).

The term in the gauge variation of $\mu_i$ involving the fields $\alpha$ and $\mathfrak{c}_\gamma$ is $\epsilon_{ij}\partial_j\mathfrak{c}_\gamma\partial_{\bar z}\alpha - \epsilon_{ij}\partial_{\bar z}\mathfrak{c}_\gamma\partial_j\alpha$. Therefore, the gauge variation of $\int \widetilde{J}^i[m,n]D_{m,n}\mu_i$ involving such terms is

$$\int \widetilde{J}^i[m,n]D_{m,n}\left(\epsilon_{ij}\partial_{w_j}\mathfrak{c}_\gamma\partial_{\bar z}\alpha - \epsilon_{ij}\partial_{\bar z}\mathfrak{c}_\gamma\partial_{w_j}\alpha\right).$$

The term in the gauge variation of $\mu_{\bar z}$ involving $\alpha$ and $\mathfrak{c}_\gamma$ is $-\epsilon_{ij}\partial_{w_i}\mathfrak{c}_\gamma\partial_{w_j}\alpha$. Therefore, the gauge variation of $\int \widetilde{T}[m,n]D_{m,n}\mu_{\bar z}$ involving such terms is

$$\int \widetilde{T}[m,n]D_{m,n}(-\epsilon_{ij}\partial_{w_i}\mathfrak{c}_\gamma\partial_{w_j}\alpha).$$

The sum of these anomalies can only be cancelled by a gauge variation of a term of the form

$$\int_{z,z',\eta_a,\eta'_a} G_\alpha[r,s](z,\eta_a)D_{r,s}\alpha(z,w_i=0,\eta_a)G_\gamma[k,l](z',\eta'_a)D_{k,l}\gamma(z',w'_i=0,\eta'_a).$$

The gauge variation of this expression involving the terms $\mathfrak{c}_\gamma$ and $\alpha$ is

$$\int_{z,z',\eta_a,\eta'_a} \overline{\partial}_{z'}\left(G_\alpha[r,s](z,\eta_a)G_\gamma[k,l](z',\eta'_a)\right)D_{r,s}\alpha(z,w_i=0,\eta_a)D_{k,l}\mathfrak{c}_\gamma(z',w'_i=0,\eta_a).$$

Let us plug in test fields $\alpha = d\bar{z}w_1^r w_2^s G(z,\bar z,\eta_a)$ and $\mathfrak{c}_\gamma = w_1^k w_2^l H(z,\bar z,\eta_a)$ where $G,H$ are arbitrary functions. Cancellation of these gauge anomalies requires

$$\int_{z,z',\eta_a,\eta'_a} \overline{\partial}_{z'}\left(G_\alpha[r,s](z,\eta_a)G_\gamma[k,l](z',\eta'_a)\right)G(z,\bar z,\eta_a)H(z',\bar z',\eta'_a)$$

$$= l\int_{z'',\eta''_a} \widetilde{J}^1[r+k,s+l-1](z'',\eta''_a)H(z'',\bar z'',\eta''_a)\partial_{z''}G(z'',\bar z'',\eta''_a)$$

$$- k\int_{z'',\eta''_a} \widetilde{J}^2[r+k-1,s+l](z'',\eta''_a)H(z'',\bar z'',\eta''_a)\partial_{z''}G(z'',\bar z'',\eta''_a)$$

$$- s\int_{z'',\eta''_a} \widetilde{J}^1[r+k,s+l-1](z'',\eta''_a)\partial_{z''}H(z'',\bar z'',\eta''_a)G(z'',\bar z'',\eta''_a)$$

$$+ r\int_{z'',\eta''_a} \widetilde{J}^2[r+k-1,s+l](z'',\eta''_a)\partial_{z''}H(z'',\bar z'',\eta''_a)G(z'',\bar z'',\eta''_a)$$

$$- \int_{z'',\eta''_a} \widetilde{T}[r+k-1,s+l-1](z'',\eta''_a)H(z'',\bar z'',\eta''_a)G(z'',\bar z'',\eta''_a). \tag{156}$$

We integrate by parts to rewrite the right hand side as

$$\int_{z'',\eta_a''}\left(-l\partial_{z''}\widetilde{J}^1[r+k,s+l-1]+k\partial_{z''}J[r+k-1,s+l]\right.$$

$$\left.-\widetilde{T}[r+k-1,s+l-1]\right)(z'',\eta_a'')H(z'',\bar{z}'',\eta_a'')G(z'',\bar{z}'',\eta_a'')$$

$$-(s+l)\int_{z'',\eta_a''}\widetilde{J}^1[r+k,s+l-1](z'',\eta_a'')\partial_{z''}H(z'',\bar{z}'',\eta_a'')G(z'',\bar{z}'',\eta_a'')$$

$$+(r+k)\int_{z'',\eta_a''}\widetilde{J}^2[r+k-1,s+l](z'',\eta_a'')\partial_{z''}H(z'',\bar{z}'',\eta_a'')G(z'',\bar{z}'',\eta_a''). \qquad (157)$$

From these expressions we can read off the OPEs just as above. We obtain

$$G_\alpha[r,s](0,\widehat{\eta}_a)G_\gamma[k,l](z,\widehat{\eta}_a')\simeq-(s+l)\frac{1}{z^2}\widetilde{J}^1[r+k,s+k-1]+(r+k)\frac{1}{z^2}\widetilde{J}^2[r+k-1,s+l]$$

$$-l\frac{1}{z}\partial_z\widetilde{J}^1[r+k,s+l-1]+k\frac{1}{z}\partial_z\widetilde{J}^2[r+k-1,s+l]$$

$$+(rl-sk)\frac{1}{z}\widetilde{T}[r+k-1,s+l-1]. \qquad (158)$$

Using (129) we obtain the on-shell $G^\alpha-G^\gamma$ OPEs

$$G^\alpha[m,n](0)G^\gamma[r,s](z)\sim\frac{(nr-ms)}{z}T[m+r-1,n+s-1](0)+\frac{1}{z^2}J[m+r,n+s](0)$$

$$+\frac{1}{z}\left(\frac{m}{2(m+r)}+\frac{n}{2(n+s)}\right)\partial_zJ[m+r,n+s](0).$$

On the right hand side, all operators are evaluated at the variables $\widehat{\eta}+\widehat{\eta}'$.

## 5.5 $TG$ OPE

The $TG$ OPE can be computed similarly. For brevity, we will simply record the result in the next section.

## 5.6 Tree-level on-shell OPEs

The OPEs we have just computed completely characterize the tree-level defect chiral algebra. In the final part of this section we summarize all tree-level OPEs that we have deduced above. In the next section we will characterize planar backreaction effects effects (which are certain planar diagrams of loop topology) which deform and centrally extend this tree-level chiral algebra.

If $nr-ms>0$ the OPEs are

$$J[m,n](0)J[r,s](z)\sim\frac{(nr-ms)}{z}J[m+r-1,n+s-1](0), \qquad (159)$$

$$J[m,n](0)T[r,s](z)\sim\frac{(nr-ms)}{z}T[m+r-1,n+s-1](0)$$

$$+\frac{1}{z^2}\left(\frac{m}{2(r+1)}+\frac{n}{2(s+1)}\right)J[m+r,n+s](0)$$

$$+\frac{1}{2z}\left(\frac{m}{m+r}+\frac{n}{n+s}\right)\partial_zJ[m+r,n+s](0), \qquad (160)$$

$$G[m,n](0)J[r,s](z)\sim\frac{(ms-rn)}{z}G[m+r-1,n+s-1](z), \qquad (161)$$

$$G[m,n](0)T[r,s](z) \sim \left(\frac{1}{z}\partial_z + \frac{1}{z^2}\right)G[m+r,n+s](0)$$
$$+\left(\frac{m}{2(r+1)} + \frac{n}{2(s+1)}\right)\frac{1}{z^2}G[m+r,n+s](0), \qquad (162)$$

$$T[m,n](0)T[r,s](z) \sim \frac{1}{z}\left(1 + \frac{r}{2(m+1)} + \frac{s}{2(n+1)}\partial_z\right)T[m+r,n+s](0)$$
$$+\frac{1}{z^2}\left(2 + \frac{r}{2(m+1)} + \frac{s}{2(n+1)} + \frac{m}{2(r+1)} + \frac{n}{2(s+1)}\right)T[m+r,n+s](0)$$
$$+\frac{1}{4z}\left(\frac{1}{(m+1)(n+s+1)} - \frac{1}{(n+1)(m+r+1)}\right)\partial_z^2 J[m+r+1,n+s+1](0)$$
$$+\frac{1}{4z^2}\left(\frac{1}{(m+1)(s+1)} - \frac{1}{(n+1)(r+s)}\right)\partial_z J[m+r+1,n+s+1](0)$$
$$+\frac{1}{4z^2}\left(\frac{1}{n+s+1}\frac{2+m+r}{(1+m)(1+r)} - \frac{1}{m+r+1}\frac{2+n+s}{(1+n)(1+s)}\right)$$
$$\times \partial_z J[m+r+1,n+s+1](0)$$
$$+\frac{1}{2z^3}\left(\frac{1}{(m+1)(s+1)} - \frac{1}{(n+1)(r+s)}\right)J[m+r+1,n+s+1](0), \qquad (163)$$

$$G^\alpha[m,n](0)G^\gamma[r,s](z) \sim \frac{(nr-ms)}{z}T[m+r-1,n+s-1](0) + \frac{1}{z^2}J[m+r,n+s](0)$$
$$+\frac{1}{z}\left(\frac{m}{2(m+r)} + \frac{n}{2(n+s)}\right)\partial_z J[m+r,n+s](0).$$

The coefficients in the OPEs between $J - T$ and $G - G$ have to be treated with slightly more care for special choices of $n, r, m, s$, though the basic structure of the OPEs is the same. For the $J - T$ OPE, the above expression also holds when $nr - ms = 0$ and $nr = ms > 0$. For the $G - G$ OPE, if $nr - ms = 0$ and $nr = ms > 0$ we have

$$G^\alpha[m,n](0)G^\gamma[r,s](z) \sim \frac{1}{z^2}J[m+r,n+s](0) + \frac{1}{z}\partial_z J[m+r,n+s](0). \qquad (164)$$

The remaining cases are as follows. If $nr - ms = 0$ and $nr = ms = 0$ the TJ and GG OPE coefficients are instead as follows.
If $r = m = 0, s \neq 0$:

$$J[0,n](0)T[0,s](z) \sim \frac{1}{z^2}\left(\frac{n}{2(s+1)}\right)J[0,n+s](0) + \frac{1}{2z}\left(\frac{n}{n+s}\right)\partial_z J[0,n+s](0), \qquad (165)$$

$$G^\alpha[0,n](0)G^\gamma[0,s](z) \sim \frac{1}{z^2}J[0,n+s](0) + \frac{1}{z}\left(\frac{n}{(n+s)}\right)\partial_z J[0,n+s](0).$$

If $n = s = 0, r \neq 0$:

$$J[m,0](0)T[r,0](z) \sim \frac{1}{z^2}\left(\frac{m}{2(r+1)}\right)J[m+r,0](0) + \frac{1}{2z}\left(\frac{m}{m+r}\right)\partial_z J[m+r,0](0), \qquad (166)$$

$$G^\alpha[m,0](0)G^\gamma[r,0](z) \sim \frac{1}{z^2}J[m+r,0](0) + \frac{1}{z}\left(\frac{m}{m+r}\right)\partial_z J[m+r,0](0). \qquad (167)$$

If $r = s = 0$ (note that there are no $G$ operators for these values):

$$J[m,n](0)T[0,0](z) \sim \frac{1}{2z^2}(m+n)J[m,n](0) + \frac{1}{z}\partial_z J[m,n](0). \qquad (168)$$

We have so far discussed OPEs that come from cancelling the BRST variation of bulk/defect Feynman diagrams that have the topology of tree diagrams. However, they do not yet constitute the complete planar, i.e. $N \to \infty$, chiral algebra. In particular, we have not accounted for

the effects of backreaction, which will serve to deform and centrally extend the planar algebra. For example, observe that tree-level OPEs of the lowest $\eta$-component of the operators

$$J[r,s], T[0,0], \qquad G_\alpha[k,\ell], \qquad G_\gamma[k,\ell], \tag{169}$$

with $r + s = 2$ and $k + \ell = 1$, comprise the (small) $\mathcal{N} = 4$ superconformal vertex algebra of central charge zero.

If we perform the rescaling of the Kodaira-Spencer Lagrangian in the backreacted geometry, as discussed in section 2.5, then the diagrammatics have the following dependence on $N$:

1. The Kodaira–Spencer Lagrangian scales like $\sim N$.

2. The term in the Lagrangian implementing the backreaction, i.e. the cubic vertex coupling Kodaira–Spencer theory to the defect, scales like $\sim N$.

3. The propagator (either in the form of bulk-bulk or bulk-defect propagators) scales like $\sim N^{-1}$.

Putting this together, we find that the same class of diagrams as in [3] survive in the planar limit. We reproduce these below in Figure 2. As expected, this leads to central terms scaling like the CFT central charge $c \sim N$, as well as a new class of diagrams arising from the backreaction that do not scale with $N$ and deform the algebra. In the next section, we turn now to computing these diagrams and completing our characterization of the planar chiral algebra from Koszul duality.

To compute non-planar corrections, one would need to repeat this procedure for a larger class of bulk diagrams including: 1) diagrams with loops in the bulk, and 2) diagrams with $\leq 2$ backreaction legs attached to the defect plus an arbitrary number of bulk-defect propagators. The integrals quickly get difficult when working beyond the box topology, but we remark that an impressive class of non-planar contributions to the OPE of two open-string bulk operators (that is, considering additional space-filling D-branes coupled to Kodaira–Spencer theory), has been computed in [52] by incorporating a refined model for Kaluza–Klein reduction via homotopy transfer. These corrections, valid for a chiral algebra dual to Kodaira-Spencer type theories plus space-filling D-branes, can be appended immediately to our chiral algebra, but does not yet include any dependence on the fermionic variables of the internal compactification manifold. One can view the non-planar contributions of [52] as incorporating diagrams of the second type (i.e. those without bulk loops). It would be very interesting to understand if other techniques from homological algebra can be leveraged to more directly obtain other non-planar contributions. We leave the incorporation of non-planar corrections to future work.

## 5.7 Matching states in the global symmetry algebra

In §3.4 we have given a geometric characterization of the global symmetry algebra. In this short section we match explicitly with operators in the defect CFT.

Recall that this global symmetry algebra is of the form

$$\mathrm{Vect}_0\left(X^0/\mathrm{Spec}\,R\right) \oplus \mathcal{O}(X^0) \otimes \Pi\mathbf{C}^2. \tag{170}$$

As $SU(2)_R \times SL(2,\mathbf{C})$ representations we have the decompositions

$$\mathcal{O}(X^0) = R \otimes \oplus_{m \geq 0}\left(\frac{m}{2}, \frac{m}{2}\right),$$

$$\mathrm{Vect}_0\left(X^0/\mathrm{Spec}\,R\right) = R \otimes (1,0) \oplus R \otimes \left(\frac{3}{2}, \frac{1}{2}\right) \oplus R \otimes \oplus_{m \geq 2}\left(\frac{m-2}{2}, \frac{m}{2}\right) \oplus \left(\frac{m+2}{2}, \frac{m}{2}\right).$$

At the level of vector spaces, it is immediate to see the match between the global symmetry algebra and certain modes of the gravitational chiral algebra that we have computed. We describe the modes which make up the global symmetry algebra.

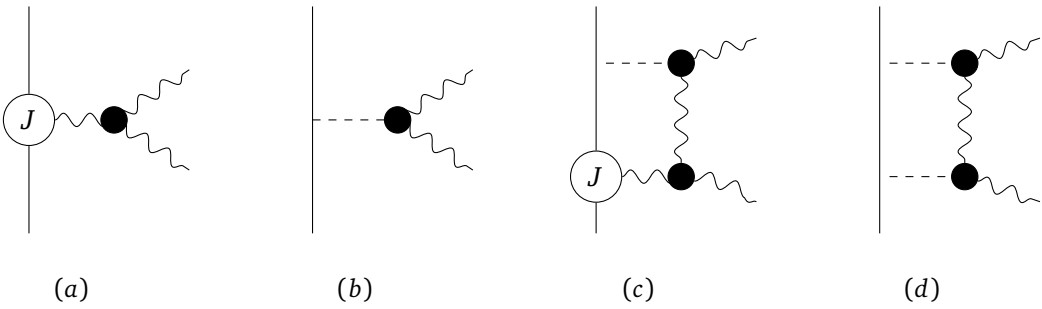

$(a)$ $(b)$ $(c)$ $(d)$

Figure 2: All diagrams that contribute in the planar limit. The solid vertical line represents the stack of $N$ branes. Wiggly lines represent Kodaira-Spencer propagators; dashed lines represent backreaction legs; circles anchored on the brane represent local operators in the chiral algebra. Diagrams $(a)$ and $(c)$ scale like $\sim \mathcal{O}(1)$ in the large-$N$ limit, and comprise 3-pt functions. We have computed the chiral algebra OPEs arising from Diagrams $(a)$ in this section. Diagrams $(b)$ and $(d)$ scale like $\sim \mathcal{O}(N)$ in the large-$N$ limit and contribute to the 2-pt function or central extension of the algebra (terms in the OPE proportional to the identity operator).

- The bosonic part of the global symmetry algebra is generated by two classes of modes. The first class is

$$\{T[r,s]_n\}, \tag{171}$$

where $0 \le n \le r+s+2$. The modes with $r = s = 0$ comprise the representation $R \otimes (1,0) = R \otimes \mathfrak{sl}(2)$. The modes with $r+s = 1$ comprise the representation $R \otimes \left(\frac{3}{2}, \frac{1}{2}\right)$. The modes with $r+s = m \ge 2$ comprise the representation $R \otimes \left(\frac{m+2}{2}, \frac{m}{2}\right)$.

- The remaining bosonic part of the global symmetry algebra is generated by the modes

$$\{J[r,s]_n\}, \tag{172}$$

where $0 \le n \le r+s-2$. Such modes satisfying $r+s = m \ge 2$ comprise the representation $\left(\frac{m-2}{2}, \frac{m}{2}\right)$. Notice that the modes of the low lying operators $J[1,0]$ and $J[0,1]$ do not appear in the global symmetry algebra. (In particular, the central term in $\widehat{L^R \mathfrak{w}_\infty}$ does not appear in the global symmetry algebra).

- The fermionic part of the global symmetry algebra is generated by the modes

$$\{G_\alpha[r,s]_n, G_\gamma[r,s]_\ell\}, \tag{173}$$

where $0 \le n, \ell \le r+s$. Such modes satisfying $r+s = m \ge 0$ comprise the representation $R \otimes \left(\frac{m}{2}, \frac{m}{2}\right) \otimes \Pi \mathbf{C}^2$.

The modes $L_{n-1} = T[0,0]_n$, $n = 0, 1, 2$, $J_0^1 = J[2,0]_0, J_0^2 = J[0,2]_0, J_0^3 = J[1,1]_0$ comprise the bosonic part of the global superconformal mode algebra. The modes $G_\alpha[1,0]_n, G_\alpha[0,1]_n, G_\gamma[1,0]_n, G_\gamma[0,1]_n$ with $n = 0, 1$ comprise the fermionic part of the global superconformal mode algebra. We can perform the usual mode integrals to convert the tree-level OPEs we have just described to obtain the familiar commutators of the $\mathfrak{psl}(1,1|2)$ global subalgebra.

# 6 OPEs from backreaction

The correspondence between the theory on a stack of branes and the gravitational theory defined on the locus away from the brane is not an exact one, even at the twisted level: to obtain a

match one must include effects from the backreaction. Geometrically, the backreaction defines the sort of geometry which is dual to the theory on a large stack of branes. This perspective persists for twisted holography. Algebraically, and importantly for us, the backreaction has the effect of deforming the dual gravitational chiral algebra defined on the boundary of (twisted) *AdS* space.

In this section, we proceed to compute planar corrections to the OPE which involve the backreaction. This will complete the determination of the planar limit of the holographically dual chiral algebra.

Since the integrals arising from diagrams this section are slightly more involved, we set up the following notations. The holomorphic coordinate on $\mathbf{C}^3$ will be $Z = (z, w)$ where $w = (w^1, w^2)$ is a holomorphic coordinate on $\mathbf{C}^2$. The defect will be located along $w = 0$. In the formulas below, our convention is that $Z^0 = z$ and $Z^i = w^i$ for $i = 1, 2$.

Before getting into the main computation of the section, we turn our attention to a simpler example.

## 6.1 Warmup: Holomorphic Chern–Simons theory

In this section, we warm up by computing the effect of backreaction on the open string sector only of a "bulk" theory. That is, we study how holomorphic Chern-Simons theory, which may be interpreted as the open string field theory for some space-filling branes in the bulk, deforms in the presence of a certain Kodaira-Spencer field (or Beltrami differential). More precisely, we consider holomorphic Chern–Simons in the presence of a Kodaira–Spencer field which is sourced by $N$ $D1$ branes wrapping $\mathbf{C} \subset \mathbf{C}^3$. The backreaction field is

$$\mu_{BR} = \frac{\epsilon_{ij}\overline{w}^i \mathrm{d}\overline{w}^j}{2\pi\|w\|^4} \partial_z \in \mathrm{PV}^{1,1}(\mathbf{C}^3 \setminus \mathbf{C}).$$

This field satisfies the equation

$$\overline{\partial}\mu_{BR} \wedge \Omega_{w_i=0} = N\delta_{w_i=0}\partial_z, \tag{174}$$

where $\delta_{w_i=0}$ is the $\delta$-function supported at $w_i = 0$. This couples to the holomorphic Chern–Simons field by

$$S_{BR} = \frac{1}{2}\int_{\mathbf{C}^3} \mu_{BR} \vee \mathrm{tr}(A\partial A) = \frac{1}{2}N\int_{\mathbf{C}^3} A^a \frac{\epsilon_{ij}\overline{w}^i \mathrm{d}\overline{w}^j}{2\pi\|w\|^4}\partial_z A^a.$$

We will denote $\omega = \frac{\epsilon_{ij}\overline{w}^i \mathrm{d}\overline{w}^j}{2\pi\|w\|^4}$ so that the coupling can be written $S_{BR} = \frac{N}{2}\int_{\mathbf{C}^3} A\omega\partial_z A$.

The backreaction coupling has a gauge anomaly even at tree-level. Indeed, the tree-level gauge variation of $S_{BR}$ is

$$\int_{\mathbf{C}^3} A^a(\overline{\partial}\mu_{BR})\mathfrak{c}^a = \int_{\mathbf{C}_z} A^a_{\overline{z}}\partial_z\mathfrak{c}^a.$$

In order to cancel this gauge anomaly one must introduce an $N$-dependent term in the OPE of the currents $J_a[k, l]$. In fact, at tree level only the OPE between currents with $k = l = 0$ is affected by the tree-level backreaction. In the presence of the backreaction the currents $J_a[0, 0]$ form a Kac–Moody algebra of level $N$

$$J_a[0,0](0)J_b[0,0](z) \simeq f^c_{ab}\frac{1}{z}J_c[0,0] + \delta_{ab}N\frac{1}{z^2}\mathrm{Id}.$$

The second term in the OPE is present due the the existence of a tree-level anomaly which involves the back reaction. The diagram which represents this anomaly is presented in figure 3.

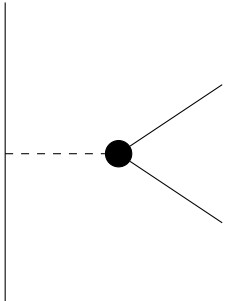

Figure 3: Tree-level diagram involving the backreaction which contributes an anomaly.

What about higher loop anomalies involving the backreaction? For scaling dimension reasons, there are no further corrections to the $J[0,0]-J[0,0]$ OPE. Let's consider the possibility of quantum corrections to the OPE between the fields $J_a[1,0]$ and $J_b[0,1]$. Before accounting for the back reaction, the tree and one-loop level OPE is

$$J_a[1,0](z)J_b[0,1] \simeq \frac{1}{z}f_{ab}^c J[1,1] + \hbar\frac{1}{z}K^{fe}f_{ae}^c f_{bf}^d J_c[0,0]J_d[0,0] \tag{175}$$

(see e.g. section 6 of [3]). By conformal invariance, the possible $N$-dependent terms in the OPE $J_a[1,0](0)J_b[0,1](z)$ must be of the form

$$\alpha f_{ae}^c K^{be}\left(\frac{1}{z^2}J_c[0,0] + \frac{1}{z}\partial_z J_c[0,0]\right) + \beta K^{ab}\frac{1}{z^3}\mathrm{Id},$$

for some (possibly zero) constants $\alpha, \beta$ which depend on $N$ (notice that the form of the central term in the last term is consistent with the fact that $J[1,0], J[0,1]$ are of spin $3/2$). The diagrams which give rise to the anomalies necessitating these terms in the OPE are presented in figure 4. In these diagrams, the dotted lines represent coupling to the backreaction and the wiggle lines represent bulk propagators. The straight lines label bulk field inputs, as before.

To evaluate the integrals associated to these diagrams we use point splitting on the defect so that operators are placed at $z_1, z_2 \in \mathbf{C}$ with $|z_1-z_2| \geq \epsilon$. The edges of the diagram correspond to the propagator for the free part of holomorphic Chern–Simons theory, which is determined by the parametrix for the $\overline{\partial}$-operator on $\mathbf{C}^3$:

$$(\overline{\partial}P) \wedge \mathrm{d}^3 Z = \delta_{Z=0}. \tag{176}$$

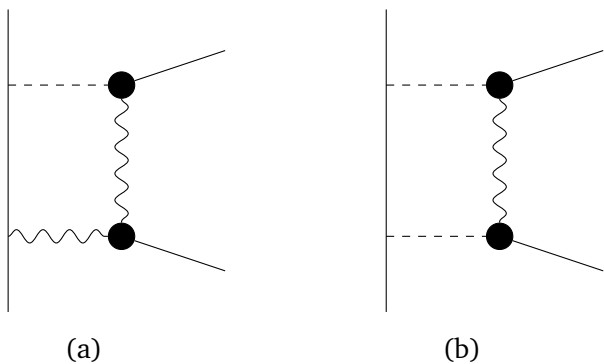

(a)             (b)

Figure 4: One-loop diagrams involving the backreaction which contribute an anomaly.

Explicitly, this is the $(0,2)$-form

$$P(Z) = \frac{1}{4\pi^2 r^6} \varepsilon_{ijk} \overline{Z}^i \, \mathrm{d}\overline{Z}^j \, \mathrm{d}\overline{Z}^k \,. \tag{177}$$

We first focus on diagram 4 (b). The weight is represented by the integral

$$\int_{(X,Y)} A_1(X)\,\omega(x)\,\partial_z\partial_w P(X,Y)\,\omega(y)A_2(Y)\,, \tag{178}$$

where we use coordinates $X = (x_1, x_2, z), Y = (y_1, y_2, w)$ and impose a cutoff $|z - w| \geq \epsilon$. In appendix A.1 we evaluate this integral to obtain

$$\frac{N^2}{2} K_{ab} \varepsilon_{ij} \int_{|z-w| \geq \epsilon} \frac{1}{(z-w)^3} \partial_{w_i} A_1^a \partial_{w_j} A_2^b |_{w_i=0} \,, \tag{179}$$

where $A_1, A_2$ are the input gauge fields. The linear BRST variation $A \mapsto A + \overline{\partial} c$ of this diagram thus gives rise to the anomaly

$$\frac{N^2}{2} K_{ab} \varepsilon_{ij} \int_{|z-w| \geq \epsilon} \frac{1}{(z-w)^3} \partial_{w_i} A^a \partial_{w_j} \overline{\partial} c^b |_{w_i=0} \,. \tag{180}$$

Integrating by parts and taking $\epsilon \to 0$ this becomes

$$\frac{N^2}{2} K_{ab} \varepsilon_{ij} \partial^3 \delta_{z_1=z_2} \partial_{w_i} A^a \partial_{w_j} \overline{\partial} c^b |_{w_i=0} \,. \tag{181}$$

In this form it is clear that this anomaly is canceled by introducing the term in the OPE in (175) with

$$\beta = \frac{N^2}{2} \,. \tag{182}$$

## 6.2 Tree-level backreaction in Kodaira–Spencer theory

We now turn to the effects of backreaction in our version of Kodaira–Spencer theory obtained by compactifying the twist of type IIB supergravity on a $K3$ surface.

The first nontrivial contribution from the backreaction actually occurs at tree-level, and is represented by Diagram b) in figure 2. We will determine this diagram first. Part of this contribution was computed in [3]. The backreaction field $\mu_{BR} = \mu_{BR}(\eta)$ takes a similar form as in the previous section. It is a distributional section

$$\mu_{BR} \in \mathrm{PV}^{1,1}(\mathbf{C}^3) \otimes R \,, \tag{183}$$

which satisfies the defining distributional equation

$$\overline{\partial} \mu_{BR} = \delta_{w_i=0} F \partial_z \,, \tag{184}$$

where $F \in H^2(K3) \subset A$ is the flux labeling the brane configuration.

The field $\mu_{BR}$ couples to the fields $\mu_i$ via

$$\int_{Z,\eta} \mu_{BR} \mu_1 \mu_2 \,. \tag{185}$$

It couples to the fields $\alpha, \gamma$ through

$$\int_{Z,\eta} \mu_{BR} \alpha \partial_z \gamma \,. \tag{186}$$

Notice that by type reasons the backreaction field does not couple to the Beltrami field $\mu_z$ in the direction parallel to the brane.

We first consider the gauge anomaly involving the coupling (185). The tree-level gauge variation of the backreaction coupling (185) is

$$\int_{Z,\eta} \mu_{BR}\overline{\partial}\mathfrak{c}_1\mu_2 + \int_{Z,\eta} \mu_{BR}\mu_1\overline{\partial}\mathfrak{c}_2 = \int_{z,\eta} (\mathfrak{c}_1\mu_2 + \mu_1\mathfrak{c}_2)|_{w=0}. \tag{187}$$

Similarly, the tree-level gauge variation of the coupling (186) is

$$\int_{Z,\eta} \mu_{BR}\overline{\partial}\mathfrak{c}_\alpha\partial_z\gamma + \int_{Z,\eta} \mu_{BR}\alpha\overline{\partial}\partial_z\mathfrak{c}_\gamma = \int_{z,\eta} \left(\mathfrak{c}_\alpha\partial_z\gamma + \alpha\partial_z\mathfrak{c}_\gamma\right)|_{w=0}. \tag{188}$$

Notice that neither of these expression involve $w_i$-derivatives. Since $\widetilde{J}^i[0,0]$ couples to $\mu_i$, the anomaly in (187) can be cancelled by the gauge variation of

$$\int_{z,\eta,z',\eta'} \widetilde{J}^1[0,0](z)\mu_1(z)\widetilde{J}^2[0,0](z')\mu_2(z'), \tag{189}$$

provided that the $\widetilde{J}^i[0,0]$ operators satisfy an appropriate OPE. Similarly, the anomaly in (188) can be cancelled by the gauge variation of a coupling of the form

$$\int_{z,\eta,z',\eta'} G_\alpha[0,0](z)\alpha(z)G_\gamma[0,0](z')\gamma(z'). \tag{190}$$

Proceeding as above by working in the Fourier dual odd coordinates and then transforming to the basis of on-shell fields, we see that to cancel the first of these anomalies there must be a term in the off-shell $\widetilde{J}\widetilde{J}$ OPE of the form

$$\widetilde{J}^i[0,0](0,\widehat{\boldsymbol{\eta}})\widetilde{J}^j[0,0](z,\widehat{\boldsymbol{\eta}}') \simeq \varepsilon^{ij}\frac{1}{z}\widehat{F}(\widehat{\boldsymbol{\eta}} + \widehat{\boldsymbol{\eta}}'). \tag{191}$$

Using the constraints (129) we can write this OPE in terms of on-shell fields as

$$J[1,0](0,\widehat{\boldsymbol{\eta}})J[0,1](z,\widehat{\boldsymbol{\eta}}') \simeq \frac{1}{z}\widehat{F}(\widehat{\boldsymbol{\eta}} + \widehat{\boldsymbol{\eta}}'). \tag{192}$$

To cancel the second anomaly (188) there must be a term in the $GG$ OPE of the form

$$G_\alpha[0,0](0,\widehat{\boldsymbol{\eta}})G_\gamma[0,0](z,\widehat{\boldsymbol{\eta}}') \simeq \frac{1}{z^2}\widehat{F}(\widehat{\boldsymbol{\eta}} + \widehat{\boldsymbol{\eta}}'). \tag{193}$$

Recall that in section 3.3 we pointed out a discrepancy in our supergravity elliptic genus and the one computed in [11], which in the notation of that section arose from the two representations $(\frac{1}{2})_S \otimes H^{2,0}(K3)$ and $(\frac{1}{2})_S \otimes H^{2,2}(K3)$. We observe that these representations form a sub-chiral algebra. Indeed, if we expand $J[1,0]$ in the Fourier dual coefficients as

$$J[1,0](\widehat{\boldsymbol{\eta}}) = J_0[1,0] + \widehat{\eta}J_{\widehat{\eta}}[1,0] + \cdots, \tag{194}$$

and similarly for $J[0,1]$, then these representations correspond to the fields

$$J_0[1,0], J_{\widehat{\eta}}[1,0], J_0[0,1], J_{\widehat{\eta}}[0,1]. \tag{195}$$

The only OPEs between these fields involves the flux $F$. They are given by

$$J_0[1,0](0)J_{\widehat{\eta}}[0,1](z) \simeq \frac{\overline{f}}{z},$$

$$J_0[0,1](0)J_{\widehat{\eta}}[1,0](z) \simeq -\frac{\overline{f}}{z},$$

where $\overline{f}$ is the component of $\overline{\boldsymbol{\eta}}$ in the original flux $F \in H^2(K3)$.

Consider next the operators

$$J[1,0](\widehat{\boldsymbol{\eta}}), J[0,1](\widehat{\boldsymbol{\eta}}), G_\alpha[0,0](\widehat{\boldsymbol{\eta}}), G_\gamma[0,0](\widehat{\boldsymbol{\eta}}).$$

These operators form a subalgebra of the full gravitational chiral algebra, even after taking into account the effect of the backreaction. We can relate this to a familiar system of free fields by a simple modification. Recall that the spin of the operator $G_\alpha[0,0]$ is one. If we choose a spin zero operator $\widetilde{G}_\alpha[0,0]$ such that $\partial \widetilde{G}_\alpha[0,0]$ then we can obtain the same OPE as above if we declare that

$$\widetilde{G}_\alpha[0,0](0,\widehat{\boldsymbol{\eta}})G_\gamma[0,0](z,\widehat{\boldsymbol{\eta}}') \simeq \frac{1}{z}\widehat{F}(\widehat{\boldsymbol{\eta}}^a + \widehat{\boldsymbol{\eta}}'^a). \tag{196}$$

The operators $J[1,0](\widehat{\boldsymbol{\eta}}), J[0,1](\widehat{\boldsymbol{\eta}}), \widetilde{G}_\alpha[0,0](\widehat{\boldsymbol{\eta}}), G_\gamma[0,0](\widehat{\boldsymbol{\eta}})$ form a familiar chiral algebra of free fields. The zero mode of $\widetilde{G}$ is topological and can be ignored; the fact that we take the derivative arises in Kodaira-Spencer theory from the fact that we chose a potential for the corresponding polyvector field in §2.

Explicitly, this is the $\beta\gamma bc$ system defined over the ring $R$. This is the chiral algebra whose fields (of spins $0,1,1/2,1/2$ respectively)

$$c = \widetilde{G}_\alpha[0,0], \qquad b = G_\gamma[0,0], \qquad \beta = J[1,0], \qquad \gamma = J[0,1], \tag{197}$$

are each valued in $R$.

From the point of view of the UV gauge theory, this comes from the twist of the fields in the $U(1)$ supermultiplet that corresponds to the collective motion of the $D1-D5$ system in the transverse directions. We emphasize that while these center of mass operators do have nontrivial OPEs with the remaining part of the chiral algebra, the operators which do not include the center of mass operators form a subalgebra of our holographically dual chiral algebra; recall that the contribution of these center of mass operators was subtracted by hand in §3 to match the elliptic genus of §4.

## 6.3 The propagator for Kodaira–Spencer theory

In a moment we will proceed with the characterization of how higher loop effects involving the backreaction in the $K3$ version of Kodaira–Spencer theory deforms the boundary chiral algebra. To set up the computations we recall the form of the propagator in Kodaira–Spencer theory. In this section we follow [40] which introduced this propagator.

The propagator for Kodaira–Spencer theory on $\mathbf{C}^3$ is the kernel for the operator $\partial\overline{\partial}^*\triangle^{-1}$. We obtain this by applying the divergence operator to the kernel for the operator $\overline{\partial}^*\triangle^{-1}$ (the analytic part of this kernel is the same as the analytic part of the propagator used in holomorphic Chern–Simons theory).

As usual, we use $Z = (Z_1 = w_1, Z_2 = w_2, Z_3 = z)$ for the holomorphic coordinate on $\mathbf{C}^3$. Using the Calabi–Yau form one can express the integral kernel for the operator $\overline{\partial}^*\triangle^{-1}$ in terms of the distributional Kodaira–Spencer field

$$P(Z) = \frac{1}{4\pi^2 r^6}\varepsilon_{ijk}\overline{Z}^i d\overline{Z}^j d\overline{Z}^k \partial^3, \tag{198}$$

where $\partial^3 = \partial_{Z_1}\partial_{Z_2}\partial_{Z_3}$. The kernel is obtained by pulling back this section along the difference map

$$\mathbf{C}^3 \times \mathbf{C}^3 \to \mathbf{C}^3, \qquad (Z,Z') \mapsto Z - Z'.$$

We denote the pulled back section by

$$P(Z,Z') \in \overline{\mathrm{PV}}^{3,2}(\mathbf{C}^3 \times \mathbf{C}^3).$$

Here $\overline{\mathrm{PV}}^{3,2}$ stands for distributional Dolbeault valued polyvector fields of type $(3,2)$. Notice that this section is smooth away from the diagonal in $\mathbf{C}^3 \times \mathbf{C}^3$.

We are interested in the Kodaira–Spencer propagator which we will denote by $\mathbf{P}$; this is the kernel of the operator $\partial \overline{\partial}^* \Delta^{-1}$. To obtain this, we first apply the divergence operator to $P$

$$\mathbf{P} = \partial P \in \overline{\mathrm{PV}}^{2,2}(\mathbf{C}^3).$$

Explicitly this is

$$\mathbf{P}(Z) = \frac{3}{4\pi^2 r^8} \varepsilon_{ijk} \varepsilon_{lmn} \overline{Z}^i \overline{Z}^l \, \mathrm{d}\overline{Z}^j \, \mathrm{d}\overline{Z}^k \partial_{Z_m} \partial_{Z_n}. \tag{199}$$

We can expand this in terms of the coordinates $Z = (z, w_1, w_2)$ where $z$ is the holomorphic coordinate along the defect. Then,

$$\begin{aligned}
\mathbf{P}(z, w_i) = {} & \frac{3 \mathrm{d}\overline{w}_1 \mathrm{d}\overline{w}_2}{4\pi^2 r^8} \left( \overline{z}^2 \partial_{w_1} \partial_{w_2} - \overline{z}\overline{w}_1 \partial_z \partial_{w_2} + \overline{z}\overline{w}_2 \partial_z \partial_{w_1} \right) \\
& + \frac{3 \mathrm{d}\overline{w}_2 \mathrm{d}\overline{z}}{4\pi^2 r^8} \left( \overline{z}\overline{w}_1 \partial_{w_1} \partial_{w_2} - \overline{w}_1^2 \partial_z \partial_{w_2} + \overline{w}_1 \overline{w}_2 \partial_z \partial_{w_1} \right) \\
& + \frac{3 \mathrm{d}\overline{z} \mathrm{d}\overline{w}_1}{4\pi^2 r^8} \left( \overline{z}\overline{w}_2 \partial_{w_1} \partial_{w_2} - \overline{w}_1 \overline{w}_2 \partial_z \partial_{w_2} + \overline{w}_2^2 \partial_z \partial_{w_1} \right).
\end{aligned}$$

Pulling back along the difference map $\mathbf{C}^3 \times \mathbf{C}^3 \to \mathbf{C}^3$ we obtain the Kodaira–Spencer theory propagator

$$\mathbf{P}(Z, Z') \in \overline{\mathrm{PV}}^{2,2}(\mathbf{C}^3 \times \mathbf{C}^3).$$

This distribution is the integral kernel for the operator $\partial \overline{\partial}^* \Delta^{-1}$ acting on polyvector fields. As in the case of the propagator for holomorphic Chern–Simons theory, it is smooth away from the diagonal. We interpret this propagator as a symmetric element of the (completed) tensor square of the fields of Kodaira–Spencer theory on $\mathbf{C}^3$.

The propagator for Kodaira–Spencer theory on $K3 \times \mathbf{C}^3$ (after compactification) is the kernel for the operator $\partial \overline{\partial}^* \Delta^{-1}$ acting on the full space of fields which acts on the odd $\boldsymbol{\eta}$-coordinates by the identity:

$$\mathbf{P}(Z, \boldsymbol{\eta}; Z, \boldsymbol{\eta}') = \mathbf{P}(Z, Z') \delta_{\eta = \eta'}. \tag{200}$$

## 6.4 The central term

We have classified the planar bulk-boundary Feynman diagrams which involve the backreaction; there were three types. The first type occurs at tree-level, involving only a single backreaction vertex, and we have characterized the effect on the boundary chiral algebra in section 6.2. There are two planar one-loop diagrams involving the backreaction: one involves a single backreaction vertex, see figure 6, and the other involves two backreaction vertices as in figure 5. In this section we focus on the latter one-loop diagram, involving two backreaction vertices, which has the special feature (like the tree-level backreaction effect) that it only couples to the identity operator in the chiral algebra along the brane. This means that the gauge anomaly resulting from this diagram introduces a central term in the OPE.

We proceed with the description of the anomaly associated to the diagram in figure 5 which involves two backreaction vertices and a single propagator. We first consider the terms in the weight of the diagram involving the bulk fields $\mu_1 - \mu_2$ (there are also terms involving input fields $\mu - \mu$ and $\alpha - \gamma$). The weight of this diagram involving these fields is represented by the integral

$$\int_{X, \boldsymbol{\eta}_X, Y, \boldsymbol{\eta}_Y} \mu_1(X) \mu_{BR}(x) \mathbf{P}(X, Y) \mu_{BR}(y) \mu_2(Y), \tag{201}$$

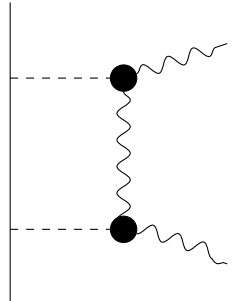

Figure 5: The diagram which encodes the one-loop central term in the OPE.

where we use coordinates $X = (x_1, x_2, z), Y = (y_1, y_2, w)$ for $\mathbf{C}^3 \times \mathbf{C}^3$ and impose a point splitting cutoff $|z - w| \geq \epsilon$.

We first observe the $\eta$-dependence of the integral above. The backreaction $\mu_{BR}$ is proportional to $F$ and the $\eta$-dependence on the propagator is through $\delta_{\eta_X = \eta_Y}$. Thus, in total, the $\eta$-dependence on the integrand is

$$\mu_{BR}(\eta_X)\mu_{BR}(\eta_Y)F(\eta_X)F(\eta_Y)\delta_{\eta_X = \eta_Y}. \tag{202}$$

From this we see that the anomaly associated to this diagram will only involve the unit component of the field $\mu_{BR}(\eta) = \mu_{BR,0} + \mathcal{O}(\eta)$ and the resulting OPE will be proportional to $N = F^2|_{\eta\bar{\eta}}$.

In appendix A.2 we evaluate this integral to obtain

$$-\frac{N}{4}\varepsilon_{ij}\int_{|z-w|\geq\epsilon}\frac{1}{(z-w)^2}\partial_{w_i}\mu_1\partial_{w_j}\mu_2|_{w_i=0,\eta=0}. \tag{203}$$

From this expression, we see that there is a gauge anomaly which can be canceled upon introducing the following term OPE

$$\widetilde{J}_0^i[1,0](0)\widetilde{J}_0^j[0,1](z,\widehat{\boldsymbol{\eta}}') \simeq \cdots \boxed{-\varepsilon^{ij}\frac{1}{4z^2}N.} \tag{204}$$

The $\cdots$ indicates terms in the OPE which do not depend on the backreaction that we characterized in the previous section (and possibly terms that arise from anomalies associated to other diagrams involving the backreaction, but in this case there are none).

One can use this expression to solve for the OPE involving the on-shell fields. This central term in the OPE will involve the operators $J[r,s]$ with $r + s = 2$, which implies that the lowest $\boldsymbol{\eta}$-components of such operators comprise an $\mathfrak{sl}(2)$-current algebra of level $N/2$. For example

$$J_0[1,1](0)J_0[1,1](z) \simeq \frac{1}{z^2}\frac{N}{2}, \tag{205}$$

where $J_0[1,1]$ is the lowest $\boldsymbol{\eta}$-component of the operator $J[1,1]$.

There is also a central term in the OPE involving the operators $G_\alpha[1,0], G_\alpha[0,1], G_\gamma[1,0], G_\gamma[0,1]$ resulting from the BRST variation of the weight represented by figure 5 where the input fields are $\alpha, \gamma$ respectively. This weight is represented by the following integral

$$\int_{X,\eta_X,Y,\eta_Y}\alpha(X)\mu_{BR}(x)\partial_z\partial_w P(X,Y)\mu_{BR}(y)\gamma(Y), \tag{206}$$

where $X = (z, x_1, x_2)$, $Y = (w, y_1, y_2)$, and $P(X, Y)$ is the propagator for $\overline{\partial}$. This is identical to the integral which is computed in appendix A.1; the result is

$$G_{\alpha,0}[1,0]G_{\gamma,0}[0,1] \simeq \cdots - 2N\frac{1}{(z-w)^3} + \cdots,$$

$$G_{\alpha,0}[0,1]G_{\gamma,0}[1,0] \simeq \cdots + 2N\frac{1}{(z-w)^3} + \cdots,$$

where the $\cdots$ denote non-central terms.

In the previous section, we observed that the tree-level OPE's between the bosonic operators

$$T_0[0,0], J_0[2,0], J_0[1,1], J_0[0,2], \tag{207}$$

together with the fermionic operators

$$G_{\alpha,0}[1,0], G_{\alpha,0}[0,1], G_{\gamma,0}[1,0], G_{\gamma,0}[0,1], \tag{208}$$

comprise the (small) $\mathcal{N} = 4$ superconformal vertex algebra at central charge zero. We have just seen that the backreaction introduces a level $k = \frac{N}{2}$ of the $\mathfrak{sl}(2)$ current algebra generated by the fields $J_0[2,0], J_0[1,1], J_0[0,2]$. This level completely determines the central charge of the superconformal algebra generated by these operators, $c = 12k = 6N$. One can alternatively directly compute the corresponding integrals corresponding to the $TT$ (after putting them on-shell) and $GG$ OPEs and find precisely the remaining central extension terms.

More generally, the diagram analyzed above gives central terms in OPE's of the form $J[k,l]J[r,s] \sim \frac{1}{z^2}$ where the total spin of the generators is 2. We have presented the calculation when $k + l = 2, r + s = 2$. This is the only combination of spins that impacts the superconformal algebra. At the level of unconstrained fields we only considered operators $\widetilde{J}^i[k,l]$ with $k + l = 1$. Therefore, the only other possibility we have not yet considered is the OPE between the unconstrained fields $\widetilde{J}^i[0,0]$ and $\widetilde{J}^j[1,1]$. By a completely similar computation, one finds (in the equations below we suppress $\mathcal{O}(1)$ constants, although they can easily be reinstated)

$$\widetilde{J}_0^i[0,0](0)\widetilde{J}_0^j[1,1](z) \simeq \cdots + \epsilon^{ij}\frac{1}{z^2}N. \tag{209}$$

At the level of the constrained (on-shell) operators, this becomes (up to dropped constants)

$$J_0[1,0](0)J_0[1,2](z) \sim \cdots + \frac{N}{z^2}, \tag{210}$$

$$J_0[0,1](0)J_0[2,1](z) \sim \cdots - \frac{N}{z^2}. \tag{211}$$

## 6.5 Non-central effects from backreaction

We move onto the anomaly arising from the one-loop diagram involving a single backreaction vertex as depicted in figure 6. In addition to the backreaction, this diagram involves two propagators and a single bulk vertex. We will mostly focus on the corrections of the OPEs involving the generators that have no dependence on the cohomology ring of K3 or $T^4$, although one can generalize our computations to include this case. Thus, the results in this section give corrections to the gravitational OPE for $B$-branes in the topological string on $\mathbf{C}^3$.

The description of the weight of this diagram is more complicated than the central backreaction terms we have considered so far. One reason is that this diagram will affect the OPE between an infinite tower of operators in the holographically dual chiral algebra (even in the planar limit). Secondly, there are more choices of possible labelings of the external edges of this diagram by fields in Kodaira–Spencer theory.

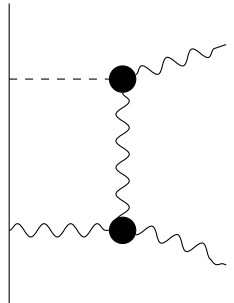

Figure 6: This diagram describes the non-central effect of the backreaction.

Consider the case where the input fields are $\mu_j$, $j = 1, 2$ so that the weight of the diagram is represented by the integral

$$-\int_{w,\boldsymbol{\eta}_w} \widetilde{J}^k[a_1, a_2](w) \int_{X,\boldsymbol{\eta}_X,Y,\boldsymbol{\eta}_Y} \mu_{BR}(x)\mu_i(z,x)\mathbf{P}(X,Y)\mu_j(Y)D_{a_1,a_2}\mathbf{P}(Y,W). \tag{212}$$

Here $w, \boldsymbol{\eta}_w$ are coordinates at the defect vertex and $X, \boldsymbol{\eta}_X, Y, \boldsymbol{\eta}_Y$ are coordinates at the bulk vertices which we integrate over. For notational symmetry, we have used the notation $W = (w, 0)$ for viewing the defect coordinate as a bulk coordinate.

There are similar contributions correcting the other OPEs, but we will focus on the $JJ$ OPE because (1) it is the most technically difficult to compute; all the other integrals can be performed with simpler versions of the computations we present in appendix B and (2) the $J$-fields include the highest weight states in each superconformal multiplet, so that the other OPEs can be alternatively obtained by leveraging the superconformal symmetry.

To get some intuition first, let us note that the gauge variation of this anomaly is of the schematic form

$$c(i, j, k, l)\int_{w,\boldsymbol{\eta}_w} (D_1\mathfrak{c}_i)\,\partial_w^l\left(D_2\mu_j\right)\widetilde{J}^k[a_1, a_2]|_{w^t=0} \wedge F(\boldsymbol{\eta}_w), \tag{213}$$

where $D_i$ are constant coefficient differential operators, in the $w_1, w_2$-coordinates whose orders sum to $2l + a_1 + a_2 + 1$, and the $c(i, j, k, l)$ are some coefficients. The order of the differential operators is determined from form of the diagram, which involves a single backreaction. This anomaly will introduce additional linear terms in the OPE between the (off-shell) operators $\widetilde{J}^i[k_1, k_2]$ and $\widetilde{J}^j[m_1, m_2]$ of the following heuristic form

$$\widetilde{J}^i[k_1, k_2](z,\boldsymbol{\eta})\widetilde{J}^j[m_1, m_2](0,\boldsymbol{\eta}') \simeq \cdots + c'(i,j,k,l)\frac{1}{z^{l+1}}\widetilde{J}^k[a_1, a_2](0)\widehat{F}(\widehat{\boldsymbol{\eta}} + \widehat{\boldsymbol{\eta}}') + \cdots, \tag{214}$$

where $k_1 + k_2 + m_1 + m_2 = 2l + a_1 + a_2 + 1$. The first $\cdots$ refer to tree-level terms which we computed in the previous section. The second $\cdots$ refer to terms with more derivatives acting on $J^k[a_1, a_2]$. In appendix B, we will find by explicit computation that $l = 1$ (and moreover, $a_1, a_2$ are fixed in terms of $k_1, k_2, m_1, m_2$ using the fact that the integrals we are evaluating are $U(1)$-equivariant with respect to $x^i, Y$) so that the leading pole goes like $\frac{1}{z^2}$. The subleading single pole $\frac{1}{z}\partial J$ is fixed by symmetry as usual to have half the coefficient of the double pole (but can also be obtained from direct integration, although we will not present that here).

Let us now explicitly see how the $JJ$ OPE will get deformed for some particular low-lying modes. (In particular, there should be no non-vanishing diagrams deforming the $\mathcal{N} = 4$ superconformal algebra, and indeed that is the case).

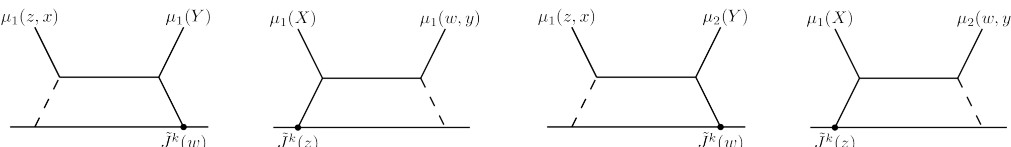

Figure 7: The diagrams contributing to the non-central deformation of the planar OPE. We will need to consider all possible defects $\tilde{J}^k[r,s], k = 1, 2$ to obtain the full on-shell OPE.

All the contributing diagrams of the given topology, including labelings of external lines, are displayed in figure 7. Let us evaluate these diagrams for a few illustrative examples. We first consider the OPE $J[2,1]J[0,2]$. In terms of off-shell OPEs, we have a rather simple expression (see also equation (B.18))

$$J[2,1]J[0,2] = 2\tilde{J}^1[2,0]\tilde{J}^1[0,1] - 4\tilde{J}^1[0,1]\tilde{J}^2[1,1].\tag{215}$$

This OPE receives contributions from the diagrams in figure 7. We will show how to explicitly calculate the contribution from the first diagram, and only state the results for the other three.

A more general treatment of these calculations for all $\tilde{J}^i[m,n]\tilde{J}^j[k,l]$ OPEs (with arbitrary $m,n,k,l$) is presented in appendix B.

The weight of the first diagram is

$$\mathcal{W}_{11}(0) = -\int_{z,w} \tilde{J}^k[0](w) \int_{\mathbb{C}^2 \times \mathbb{C}^3} \mu_{BR}(x)\mu_1(z,x)P(X,Y)\mu_1(Y)P(Y,W).\tag{216}$$

We specialize the external legs to be

$$\mu_1(z,x) = z(x^1)^2 d\bar{z}\partial_{x^1}, \qquad \mu_1(Y) = (y^2)d\bar{y}^0\partial_{y^1}.\tag{217}$$

Then, $k = 1$ and the only non-trivial contributions come from the $\partial_{(z-y^0)}\partial_{(x^2-y^2)}$ component of $P(X,Y)$, and the $\partial_{y^1}\partial_{y^2}$ component of $P(Y,W)$.

$$\mathcal{W}_{11}(0) = -\left(\frac{1}{2\pi}\right)^5 3^2 4^2 \int_{z,w} z\tilde{J}^1[0](w) \int_{\mathbb{C}^2 \times \mathbb{C}^3} \frac{[\bar{x},\bar{y}](\bar{z}-\bar{y}^0)(\bar{y}^0-\bar{w})(\bar{x}^1-\bar{y}^1)(x^1)^2(y^2)}{(||x||^2)^2(||X-Y||^2)^4(||Y-W||^2)^4}.\tag{218}$$

We first integrate over $d^3Y$. For cleanliness, we will write only the part of the diagram that participates nontrivially in the $d^3Y$ integral as $\tau_y$ (and similarly for $d^4x$ shortly), and combine all contributions at the end. Using Feynman's trick,

$$\tau_y = \int_Y \frac{[\bar{x},\bar{y}](\bar{z}-\bar{y}^0)(\bar{y}^0-\bar{w})(\bar{x}^1-\bar{y}^1)(y^2)}{(||X-Y||^2)^4(||Y-W||^2)^4}\tag{219}$$

$$= \left(\frac{\Gamma(8)}{\Gamma(4)^2}\right)\int_0^1 dt\, t^3(1-t)^3 \int_Y \frac{[\bar{x},\bar{y}](\bar{z}-\bar{y}^0)(\bar{y}^0-\bar{w})(\bar{x}^1-\bar{y}^1)(y^2)}{(t||X-Y||^2 + (1-t)||Y-W||^2)^8}.\tag{220}$$

We shift the integration variable $Y \to Y + tX + (1-t)W$ and impose $U(1)_Y$ equivariance

$$\tau_y = \left(\frac{\Gamma(8)}{\Gamma(4)^2}\right)(\bar{z}-\bar{w})^2(\bar{x}^1)^2 \int_0^1 dt\, t^4(1-t)^5 \int_Y \frac{(|y^2|^2)}{(||Y||^2 + t(1-t)||X-W||^2)^8}.\tag{221}$$

We introduce radial coordinates $r^i = \frac{|y^i|^2}{t(1-t)||X-W||^2}$ and perform the angular integration,

$$\tau_y = \left(\frac{\Gamma(8)}{\Gamma(4)^2}\right)(\bar{z}-\bar{w})^2(-2\pi i)^3\frac{(\bar{x}^1)^2}{(||X-W||^2)^4}\int_0^1 dt(1-t)\int_0^\infty \frac{r^2}{(r^0+r^1+r^2+1)^8}. \quad (222)$$

Integrating over the radial coordinates and t, we find that the $Y$ integral gives us the expression

$$\tau_y = \left(\frac{(-2\pi i)^3}{2\Gamma(4)}\right)(\bar{z}-\bar{w})^2\frac{(\bar{x}^1)^2}{(||X-W||^2)^4}. \quad (223)$$

We can now integrate over $d^4x$.

$$\tau_x = \int_x \frac{(|\bar{x}^1|)^2}{(||x||^2)^2(||X-W||^2)^4} \quad (224)$$

$$= \left(\frac{\Gamma(6)}{\Gamma(4)}\right)\int_0^1 ds\, s(1-s)^3 \int_x \frac{(|\bar{x}^1|)^2}{(||x||^2+(1-s)|z-w|)^6}. \quad (225)$$

We introduce radial coordinates $r^i = \frac{|x^i|^2}{(1-s)|z-w|^2}$ and perform the angular integration,

$$\tau_x = \left(\frac{\Gamma(6)}{\Gamma(4)}\right)(2\pi i)^2\left(\frac{1}{|z-w|^2}\right)^2\int_0^1 ds\, s(1-s)\int_0^\infty \frac{(r^1)^2}{(r^1+r^2+1)^6}. \quad (226)$$

Integrating over the radial coordinates and t,

$$\tau_x = \left(\frac{(-2\pi i)^2}{3\Gamma(4)}\right)\left(\frac{1}{|z-w|^2}\right)^2. \quad (227)$$

Putting it all together, we find

$$\mathcal{W}_{11}(0) = \frac{2i}{3}\int_{z,w} z\tilde{J}^1[0](w)\left(\frac{1}{z-w}\right)^2. \quad (228)$$

Performing similar calculations for the other three diagrams, we find the following off-shell OPEs

$$\tilde{J}^1[2,0](z,\eta)\tilde{J}^1[0,1](w,\eta') \simeq \left(\frac{-2i}{9(z-w)^2}\right)\tilde{J}^1[0,0](w)\widehat{F}(\widehat{\eta}+\widehat{\eta}'),$$

$$\tilde{J}^1[0,1](z)\tilde{J}^2[1,1](w) \simeq \left(\frac{7i}{9(z-w)^2}\right)\tilde{J}^1[0,0](w)\widehat{F}(\widehat{\eta}+\widehat{\eta}'). \quad (229)$$

Inserting this into eq.(215), we find that the on-shell OPE is

$$J[2,1](z,\eta)J[0,2](w,\eta') \simeq \left(\frac{32i}{9(z-w)^2}\right)J[0,1](w)\widehat{F}(\widehat{\eta}+\widehat{\eta}'). \quad (230)$$

One can verify that this is consistent with the more general integrals computed in appendix B. Let us take another example. Consider the OPE $J[3,0]J[0,3]$. Using equation (B.18) we have

$$J[3,0](z,\eta)J[0,3](w,\eta') \cong \left(\frac{36i}{z^2}\right)\left(\gamma_1^{(0,1)}(0,2;2,0)-\beta_1^{(0,1)}(0,2;2,0)+\beta_1^{(0,1)}(2,0;0,2)\right)\tilde{J}^2[0,1](w)\widehat{F}(\widehat{\eta}+\widehat{\eta}')$$

$$+\left(\frac{36i}{z^2}\right)\left(\gamma_2^{(1,0)}(2,0;0,2)-\beta_2^{(1,0)}(2,0;0,2)+\beta_2^{(1,0)}(0,2;2,0)\right)\tilde{J}^1[1,0](w)\widehat{F}(\widehat{\eta}+\widehat{\eta}'). \quad (231)$$

Plugging into our expressions for $\gamma$ and $\beta$ (see equations B.6, B.7), we find:

$$\gamma_1^{(0,1)}(0,2;2,0) = \frac{1}{18} = -\gamma_2^{(1,0)}(2,0;0,2),\tag{232}$$

$$-\beta_1^{(0,1)}(0,2;2,0) = \frac{5}{9} = \beta_2^{(1,0)}(2,0;0,2),\tag{233}$$

$$\beta_1^{(0,1)}(2,0;0,2) = \frac{1}{3} = -\beta_2^{(1,0)}(0,2;2,0).\tag{234}$$

We thus find that the on-shell OPE is

$$J[3,0](z,\boldsymbol{\eta})J[0,3](w,\boldsymbol{\eta}') \simeq \left(\frac{34i}{z^2}\right)J[1,1](w)\widehat{F}(\widehat{\boldsymbol{\eta}} + \widehat{\boldsymbol{\eta}}').\tag{235}$$

Finally, we remark that the planar chiral algebra should contain the information of the $c = 6N$ small $\mathcal{N} = 4$ superconformal algebra (which we have reproduced in the OPE of the low-lying generators) as well as OPEs among the superconformal descendants. It would therefore be enlightening to match the Koszul duality approach with more standard bootstrap analyses. This may be slightly tedious, since Koszul duality expresses the chiral algebra in a rather different basis than the one which is natural from the perspective of these symmetries. For example, we can use the results of the $\mathcal{N} = 4$ long-multiplet bootstrap of [53], take the $h \to (m+n)/2$ limit in which the multiplets become short, and remove the null states, to characterize the nonvanishing 2-pt functions. This is simple to check using the Mathematica code provided in [53] for the lowest-lying modes, but those come from nothing but the center of mass multiplet and the $\mathcal{N} = 4$ superconformal algebra itself, which we knew from other methods already. It could be fruitful to apply these checks, and carefully match the results, for the higher modes.

## Acknowledgments

We are grateful to Kevin Costello and Nathan Benjamin for discussions and collaboration on related works, to Surya Raghavendran for many fruitful conversations about twisted holography in general, and to Jihwan Oh for explaining the Mathematica code of [53]. NP also thanks Harvard CMSA and the Perimeter Institute's Visiting Fellow program for additional support and hospitality while this work was underway. Research at Perimeter Institute is supported by the Government of Canada through Industry Canada and by the Province of Ontario through the Ministry of Research and Innovation.

**Funding information** NP and VF are supported by funds from the Department of Physics and the College of Arts & Sciences at the University of Washington, the DOE Early Career Research Program under award DE-SC0022924, and the Simons Foundation as part of the Simons Collaboration on Celestial Holography.

## A Loop computations involving backreaction

### A.1 Backreaction in holomorphic Chern–Simons

Let $X = (z,x) = (z,x_1,x_2), Y = (w,y) = (w,y_1,y_2)$. We compute the integral

$$\int_{(X,Y)\in\mathbf{C}_1^3\times\mathbf{C}_2^3} A_1(X)\,\omega(x)\,\partial_z\partial_w P(X,Y)\,\omega(y)A_2(Y),\tag{A.1}$$

where $A_i$ are $(0, 1)$-forms on $\mathbf{C}^3$, and $P(X, Y) = P(X - Y)$ is as in equation (177). Plugging in $A = x_1 d\overline{z}$ and $B = y_2 d\overline{w}$ this integral becomes $\int_{z,w} dz \, dw \, \partial_{\overline{z}} \partial_{\overline{w}} I(z, w)$ where

$$I(z, w) \stackrel{\text{def}}{=} (\overline{z} - \overline{w}) \int_{\mathbf{C}^2 \times \mathbf{C}^2} d^4 x d^4 y \, \frac{[\overline{xy}] x_1 y_2}{\|x\|^4 (|z - w|^2 + \|x - y\|^2)^3 \|y\|^4} . \tag{A.2}$$

We compute $I(z, w)$ as a function of the difference $z - w$. Note that there is an additional factor over $\frac{1}{(2\pi)^4}$ arising from the propagator and $\omega$ which we have suppressed, and will restore at the end.

First, we perform the integration along $y \in \mathbf{C}^2$. Using Feynman's trick we have

$$\int_{\mathbf{C}^2} d^4 y \, \frac{[\overline{xy}] y_2}{(|z - w|^2 + \|x - y\|^2)^3 \|y\|^4}$$
$$= \frac{4!}{2!} \int_0^1 dt \, t^2 (1 - t) \int_{\mathbf{C}^2} d^4 y \, \frac{[\overline{xy}] y_2}{(t|z - w|^2 + t\|x - y\|^2 + (1 - t)\|y\|^2)^5} . \tag{A.3}$$

Introduce the new variable $\widetilde{y} = y - tx$. The the right hand side becomes

$$12 \int_0^1 dt \, t^2 (1 - t) \int_{\mathbf{C}^2} d^4 \widetilde{y} \, \frac{[\overline{x}(\widetilde{y} + t\overline{x})](y_2 + tx_2)}{(\|\widetilde{y}\|^2 + t(1 - t)\|x\|^2 + t|z - w|^2)^5} . \tag{A.4}$$

Changing to polar coordinates and first computing the residue we see that only terms invariant under $U(1) \times U(1)$ rotations of $\mathbf{C}^2$ will contribute to this integral. The $U(1) \times U(1)$ invariant part of the numerator is $\overline{x}_1 |\widetilde{y}_2|^2$. After computing the residue along both the $\widetilde{y}_1$ and $\widetilde{y}_2$ directions the integral then becomes

$$12(-2\pi i)^2 \overline{x}_1 \int_0^1 dt \, t^2 (1 - t) \int_{(0,\infty) \times (0,\infty)} d^2 \rho \, \frac{\rho_2}{(\rho_1 + \rho_2 + t(1 - t)\|x\|^2 + t|z - w|^2)^5} . \tag{A.5}$$

Performing the integration over $(0, \infty) \times (0, \infty)$ we obtain

$$\frac{(-2\pi i)^2}{2} \overline{x}_1 \int_0^1 \frac{1 - t}{(|z - w|^2 + (1 - t)\|x\|^2)^2} . \tag{A.6}$$

Returning to the original integral we must now compute

$$\int_0^1 dt \, (1 - t) \int_{\mathbf{C}^2} d^4 x \, \frac{|x_1|^2}{\|x\|^4 (|z - w|^2 + (1 - t)\|x\|^2)^2} . \tag{A.7}$$

We compute the integral over $x$.

Using the Feynman trick again we have

$$\int_{\mathbf{C}^2} d^4 x \, \frac{|x_1|^2}{\|x\|^4 (|z - w|^2 + (1 - t)\|x\|^2)^2} = \int_0^1 ds \, s(1 - s) \int_{\mathbf{C}^2} d^4 x \, \frac{|x_1|^2}{(s|z - w|^2 + (1 - ts)\|x\|^2)^4} . \tag{A.8}$$

After computing the angular integrations this becomes

$$(-2\pi i)^2 \int_{(0,\infty) \times (0,\infty)} d^2 \rho \, \frac{\rho_1}{(s|z - w|^2 + (1 - ts)(\rho_1 + \rho_2))^4} = \frac{1}{s(1 - ts)^3 |z - w|^2} . \tag{A.9}$$

Finally, plugging back into the original expression we have

$$I(z, w) = \frac{(-2\pi i)^2}{z - w} \int_0^1 dt \int_0^1 ds \, \frac{(1 - t)(1 - s)}{(1 - ts)^3} . \tag{A.10}$$

The integral over $t, s$ gives $\frac{1}{2}$. Combining all the resulting factors from the preceding computations, and reinstating the propagator normalization, we therefore have

$$I(z, w) = \frac{(-2\pi i)^4}{2} \frac{1}{(2\pi)^4} \frac{1}{2(z-w)} = \frac{1}{4(z-w)} . \tag{A.11}$$

## A.2 The central term in Kodaira–Spencer theory

Let the notation for the coordinates $X, Y$ be as in the last section. We will compute the integral

$$\int_{X,Y} \mu_1(X) \mu_{BR}(x) \mathbf{P}(X, Y) \mu_{BR}(y) \mu_2(Y) . \tag{A.12}$$

Without loss of generality, we plug in the test functions

$$\mu_1(X) = x_1 \partial_{x_1} d\overline{z}, \qquad \mu_2(Y) = y_2 \partial_{y_2} d\overline{w} . \tag{A.13}$$

The vector field type is determined by the symmetry of the graph while the powers of the holomorphic coordinates $x, y$ which appear are determined by the scaling properties of the propagator and backreaction.

Notice that $\mu_{BR}(x)$ is proportional to the differential form $\varepsilon_{ij} \overline{x}_i d\overline{x}_j$ and similarly for $\mu_{BR}(y)$. Thus, for these test functions only the $\partial_{x_1 - y_1} \partial_{x_2 - y_2}$ part of the BCOV propagator $\mathbf{P}(X, Y)$ will contribute to this integral. Furthermore, the terms in the BCOV propagator proportional to $d\overline{z} - d\overline{w}$ will not contribute by type reasons. Simplifying, we see that for this choice of test functions this integral becomes $\int_{z,w} dz\, dw\, I(z, w)$ where

$$I(z, w) \overset{\mathrm{def}}{=} (\overline{z} - \overline{w})^2 \int_{\mathbf{C}_x^2 \times \mathbf{C}_y^2} d^4x\, d^4y\, \frac{[\overline{xy}] x_1 y_2}{\|x\|^4 (|z-w|^2 + \|x-y\|^2)^4 \|y\|^4} , \tag{A.14}$$

where we have again suppressed the constant factors from the propagator and $\omega$, to be restored at the end. We remark that the factor $(\overline{z} - \overline{w})^2$ comes from the BCOV propagator. We compute $I(z, w)$ as a function of the difference $z - w$.

First, we perform the integration along $y \in \mathbf{C}^2$. Using Feynman's trick we have

$$\int_{\mathbf{C}^2} d^4y\, \frac{[\overline{xy}] y_2}{(|z-w|^2 + \|x-y\|^2)^4 \|y\|^4}$$
$$= \frac{5!}{3!} \int_0^1 dt\, t^3 (1-t) \int_{\mathbf{C}^2} d^4y\, \frac{[\overline{xy}] y_2}{(t|z-w|^2 + t\|x-y\|^2 + (1-t)\|y\|^2)^6} . \tag{A.15}$$

Introduce the new variable $\widetilde{y} = y - tx$. The the right hand side becomes

$$20 \int_0^1 dt\, t^3 (1-t) \int_{\mathbf{C}^2} d^4\widetilde{y}\, \frac{[\overline{x}(\widetilde{y} + t\overline{x})](y_2 + tx_2)}{(\|\widetilde{y}\|^2 + t(1-t)\|x\|^2 + t|z-w|^2)^6} . \tag{A.16}$$

The $U(1) \times U(1)$ invariant part of the numerator is $\overline{x}_1 |\widetilde{y}_2|^2$. After computing the residue along both the $\widetilde{y}_1$ and $\widetilde{y}_2$ directions the integral becomes

$$20(-2\pi i)^2 \overline{x}_1 \int_0^1 dt\, t^3 (1-t) \int_{(0,\infty) \times (0,\infty)} d^2\rho\, \frac{\rho_2}{(\rho_1 + \rho_2 + t(1-t)\|x\|^2 + t|z-w|^2)^6} . \tag{A.17}$$

Performing the integration over $(0, \infty) \times (0, \infty)$ we obtain

$$(-2\pi i)^2 \frac{5!}{3!} \frac{2!}{5!} \overline{x}_1 \int_0^1 \frac{1-t}{(|z-w|^2 + (1-t)\|x\|^2)^3} . \tag{A.18}$$

Returning to the original integral we must now compute (suppressing the overall constant factors for the moment)

$$\int_0^1 dt\,(1-t)\int_{\mathbf{C}^2} d^4 x\,\frac{|x_1|^2}{\|x\|^4(|z-w|^2+(1-t)\|x\|^2)^3}\,. \tag{A.19}$$

We compute the integral over $x$ as above to obtain

$$(-2\pi i)^4\frac{4}{4!}\frac{1}{|z-w|^4}\int_0^1 dt\int_0^1 ds\,\frac{(1-t)(1-s)}{(1-ts)^3}\,, \tag{A.20}$$

and hence, putting all the pieces together,

$$I(z,w)=\frac{3}{(2\pi)^4}\frac{(-2\pi i)^4}{6}\frac{1}{2(z-w)^2}=\frac{1}{4(z-w)^2}\,. \tag{A.21}$$

Upon changing to the basis of on-shell generators (i.e. currents sourcing the properly constrained Kodaira-Spencer fields), we will recover precisely the canonical Kac-Moody algebra at the expected level $\frac{N}{2}$.

## A.3  Evaluating a general holomorphic integral over $d^4 x\, d^4 y$

In the previous two appendices, we computed some holomorphic integrals which can deform a Koszul dual chiral algebra on a case-by-case basis. However, these integrals admit more general closed forms, and it is convenient to calculate them once and for all. In this appendix we will evaluate a general form of a holomorphic integral which is common to many 1-loop Koszul duality computations in holomorphic theories. Throughout this appendix, we employ the same notation as in §6.

We would like to obtain an expression of the form $\int dz\,dw\,I(z,w)$, where $I(z,w)$ is itself an integral over the four transverse directions $d^4 x\, d^4 y$. For notational expedience, let us strip off some overall factors which do not partake in the $d^4 x\, d^4 y$ integral, in particular: any functions of $\bar{z},\bar{w}$ which come from expanding the propagators, and any overall multiplicative constants which come from the normalizations of the propagators and the backreaction fields. We call this stripped-down integral $\mathcal{I}^1(z,w)$, and turn to its evaluation. (Of course, one must reinstate these factors at the end, and then perform the final integral over $dz\,dw$ to complete the determination of the OPE).

We begin with an integral of the form:

$$\mathcal{I}^1(\vec{j};\vec{k},\vec{l};\vec{m},\vec{n})=\int_{\mathbf{C}^2}\frac{(x^1)^{k_1}(x^2)^{k_2}(\overline{x}^1)^{l_1}(\overline{x}^2)^{l_2}}{(\|x\|^2)^{j_1}}\mathcal{I}_y(\vec{j};\vec{m},\vec{n})d^4 x\,, \tag{A.22}$$

where $\vec{k},\vec{l},\vec{m},\vec{n}\in(\mathbf{Z}_{\geq 0})^2,\vec{j}\in(\mathbf{Z}_{>0})^3, X=(z,x^{\dot{\alpha}}), Y=(w,y^{\dot{\alpha}})$ and:

$$\mathcal{I}_y(\vec{j};\vec{m},\vec{n})=\int_{\mathbf{C}^2}\frac{[\overline{x},\overline{y}](y^1)^{m_1}(y^2)^{m_2}(\overline{y}^1)^{n_1}(\overline{y}^2)^{n_2}}{(\|X-Y\|^2)^{j_2}(\|y\|^2)^{j_3}}d^4 y\,. \tag{A.23}$$

We have also made the following definition:

$$[\overline{x},\overline{y}]=\overline{x}^1\overline{y}^2-\overline{x}^2\overline{y}^1\,. \tag{A.24}$$

We first integrate over $d^4 y$. Using Feynman's trick,

$$\mathcal{I}_y(\vec{j};\vec{m},\vec{n})=\left(\frac{\Gamma(j_2+j_3)}{\Gamma(j_2)\Gamma(j_3)}\right)\int_0^1 dt\,t^{j_2-1}(1-t)^{j_3-1}\int_{\mathbf{C}^2}\frac{[\overline{x},\overline{y}](y^1)^{m_1}(y^2)^{m_2}(\overline{y}^1)^{n_1}(\overline{y}^2)^{n_2}}{(t\|X-Y\|^2+(1-t)\|y\|^2)^{j_2+j_3}}d^4 y\,.$$

Next, we shift the integration variable $y$, $y \to y + tX$, and use the binomial theorem:

$$\mathcal{I}_y(\vec{j}; \vec{m}, \vec{n}) = \left(\frac{\Gamma(j_2 + j_3)}{\Gamma(j_2)\Gamma(j_3)}\right) \int_0^1 dt\, t^{j_2-1}(1-t)^{j_3-1} \sum_{i=1}^2 \sum_{a_i=0}^{m_i} \sum_{b_i=0}^{n_i} \binom{m_i}{a_i}\binom{n_i}{b_i} \tag{A.25}$$

$$\times (tx^1)^{m_1-a_1}(tx^2)^{m_2-a_2}(t\overline{x}^1)^{n_1-b_1}(t\overline{x}^2)^{n_2-b_2} \int_{\mathbf{C}^3} \frac{[\overline{x},\overline{y}](y^1)^{a_1}(y^2)^{a_2}(\overline{y}^1)^{b_1}(\overline{y}^2)^{b_2}}{(t|z-w| + ||y||^2 + t(1-t)||x||^2)^{j_2+j_3}} d^4 y\,.$$

The integral over $y$ only receives contributions from those terms that are invariant under phase rotations of $y^{\dot\alpha}$. Let us make the following convenient definition for the summations:

$$\sum_{(a_1,a_2)}^{(\vec{m},\vec{n})} \equiv \left(\sum_{a_1=0}^{\mathrm{Min}[m_1,n_1]} \sum_{a_2=1}^{\mathrm{Min}[m_2,n_2+1]} \binom{n_1}{a_1}\binom{n_2}{a_2-1} - \sum_{a_1=1}^{\mathrm{Min}[m_1,n_1+1]} \sum_{a_2=0}^{\mathrm{Min}[m_2,n_2]} \binom{n_1}{a_1-1}\binom{n_2}{a_2}\right)\binom{m_1}{a_1}\binom{m_2}{a_2},$$

using which, eq.(A.25) reduces to

$$\mathcal{I}_y(\vec{j}; \vec{m}, \vec{n}) = \left(\frac{\Gamma(j_2 + j_3)}{\Gamma(j_2)\Gamma(j_3)}\right) \sum_{(a_1,a_2)}^{(\vec{m},\vec{n})} \int_0^1 dt\, t^{j_2+m_1+m_2+n_1+n_2-2a_1-2a_2-1}(1-t)^{j_3-1} \tag{A.26}$$

$$\times (x^1)^{m_1-a_1}(x^2)^{m_2-a_2}(\overline{x}^1)^{n_1+1-a_1}(\overline{x}^2)^{n_2+1-a_2}$$

$$\times (-2\pi i)^2 (t|z-w|^2 + t(1-t)||x||^2)^{2+a_1+a_2-j_2-j_3} \int_0^\infty \frac{(r^1)^{a_1}(r^2)^{a_2}}{(r^1+r^2+1)^{j_2+j_3}} dr^1 dr^2\,,$$

where we introduced radial coordinates $r^i = |y^i|^2/(t|z-w|^2 + t(1-t)||X-W||^2)$, and we integrated over $d\theta^i$.

Integrating over $dr^i$ and grouping terms, this simplifies to:

$$\mathcal{I}_y(\vec{j}; \vec{m}, \vec{n}) = \left(\frac{(-2\pi i)^2}{\Gamma(j_2)\Gamma(j_3)}\right) \sum_{(a_1,a_2)}^{(\vec{m},\vec{n})} \Gamma(a_1+1)\Gamma(a_2+1)\Gamma(j_2+j_3-2-a_1-a_2)$$

$$\times (x^1)^{m_1-a_1}(x^2)^{m_2-a_2}(\overline{x}^1)^{n_1+1-a_1}(\overline{x}^2)^{n_2+1-a_2}$$

$$\times \int_0^1 dt \frac{t^{2+m_1+m_2+n_1+n_2-a_1-a_2-j_3}(1-t)^{j_3-1}}{(t|z-w|^2 + t(1-t)||x||^2)^{j_2+j_3-2-a_1-a_2}}\,.$$

We now at last have the following integral, which we must integrate over $d^4 x$:

$$\mathcal{I}_x(\vec{j}; \vec{k}, \vec{l}; \vec{m}, \vec{n}) = \int_{\mathbf{C}^2} \frac{(x^1)^{k_1+m_1-a_1}(x^2)^{k_2+m_2-a_2}(\overline{x}^1)^{l_1+n_1+1-a_1}(\overline{x}^2)^{l_2+n_2+1-a_2}}{(||x||^2)^{j_1}(|z-w|^2 + (1-t)||x||^2)^{j_2+j_3-2-a_1-a_2}} d^4 x\,. \tag{A.27}$$

The steps we need to follow to perform this integral are identical to those of the $d^4 y$ integral: Feynman's trick, shifting the integration variable, and only retaining those terms which are invariant under phase rotations of $x^{\dot\alpha}$. We present the final result:

$$\mathcal{I}^1(\vec{j}; \vec{k}, \vec{l}; \vec{m}, \vec{n}) = \left(\frac{(2\pi)^4}{\Gamma(j_1)\Gamma(j_2)\Gamma(j_3)}\right) \frac{\Gamma(j_1+j_2+j_3-4-k_1-k_2-m_1-m_2)}{(|z-w|^2)^{j_1+j_2+j_3-4-k_1-k_2-m_1-m_2}} \delta^{l_i+n_i+1}_{k_i+m_i}$$

$$\times \sum_{(a_1,a_2)}^{(\vec{m},\vec{n})} \Gamma(a_1+1)\Gamma(a_2+1)\Gamma(k_1+m_1+1-a_1)\Gamma(k_2+m_2+1-a_2)$$

$$\times \int_0^1 \int_0^1 ds\, dt \frac{t^{p_1}(1-t)^{j_3-1}s^{p_2}(1-s)^{j_1-1}}{(1-st)^{p_3}}\,, \tag{A.28}$$

where we have made the following definitions:

$$p_1 = 2 + m_1 + m_2 + n_1 + n_2 - a_1 - a_2 - j_3 \,, \tag{A.29}$$

$$p_2 = 1 + k_1 + k_2 + m_1 + m_2 - a_1 - a_2 - j_1 \,, \tag{A.30}$$

$$p_3 = 2 + k_1 + k_2 + m_1 + m_2 - a_1 - a_2 \,. \tag{A.31}$$

To connect to what we have previously determined in Appendices A.1, A.2, let us take several specializations of this general form.

1. Consider $\vec{j} = (2, 3, 2), \vec{k} = (1, 0), \vec{l} = \vec{n} = 0, \vec{m} = (0, 1)$. The integral becomes

$$\mathcal{I}^1(z, w) = \int\limits_{(\mathbb{C}^2)^2} \frac{[\bar{x}, \bar{y}] x_1 y_2}{(||x||^2)^2 (||X - Y||^2)^3 (||y||^2)^2} d^4 x d^4 y \,. \tag{A.32}$$

With these parameters, the general form of our integral becomes

$$\mathcal{I}^1(z, w) = \frac{(-2\pi i)^4}{4} \frac{1}{|z - w|^2} \,. \tag{A.33}$$

This integral is precisely that in equation A.2, except with the anti-holomorphic $(\bar{z} - \bar{w})$ factor from the holomorphic Chern-Simons propagator stripped off. We also must reinstate an overall constant $\frac{1}{2\pi^4}$ coming from the normalization of the propagator and the backreaction field. To get our final answer, we simply reinstate them to recover

$$\mathcal{I}(z, w) = \frac{1}{4(z - w)} \,. \tag{A.34}$$

2. Next consider $\vec{j} = (2, 4, 2), \vec{k} = (1, 0), \vec{l} = \vec{n} = 0, \vec{m} = (0, 1)$:

$$\mathcal{I}(z, w) = \int\limits_{(\mathbb{C}^2)^2} \frac{[\bar{x}, \bar{y}] x_1 y_2}{(||x||^2)^2 (||X - Y||^2)^4 (||y||^2)^2} d^4 x d^4 y \,. \tag{A.35}$$

With these parameters, the general form of our integral becomes

$$\mathcal{I}^1(z, w) = \frac{(-2\pi i)^4}{12} \left( \frac{1}{|z - w|^2} \right)^2 \,. \tag{A.36}$$

This is (up to our stripped off factors) the integral we needed to compute the central term in our Kodaira-Spencer theory, equation A.14. We now simply reinstate the factors that depend on $\bar{z}, \bar{w}$ from the propagator, i.e. $(\bar{z} - \bar{w})^2$. To get the correct normalization for the OPE, we must also reinstate the constant factors which constitute the overall normalizations of $P, \omega$ ($\frac{3}{4\pi^2}, \frac{1}{(2\pi)^2}$, respectively), which we have so far suppressed.

The result may now be plugged into an integral over $dz dw$, with a point-splitting regulator, to complete the determination of the central term in the OPE, as in §6.

# B  Non-central terms in Kodaira–Spencer theory

We choose our notation similarly to Appendix A.2. We fix coordinates $Z = (z, 0), W = (w, 0)$ along the brane. For the diagram in Figure 8, our notation for the bulk coordinates will be $X = (z, x), Y = (y^0, y)$. Similarly, for the diagram in Figure 9, we use $X = (x^0, x), Y = (w, y)$.

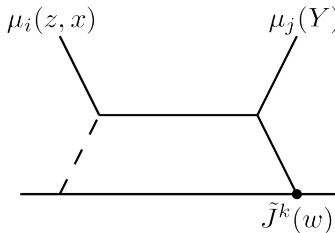

Figure 8

This final diagram type correcting the planar OPE is more involved, and the integrals are more subtle, so we will break down the analysis into simpler steps and summarize the outcome in §6.

First, we shall demand that the integral be well-defined and nonzero, by saturating the correct (antiholomorphic) differential form and polyvector degree.[27] This will enable us to isolate the terms in the weight of the diagram that contribute nontrivially to the integral.

For simplicity, in this section we work in ordinary Kodaira–Spencer theory, meaning the closed-string topological $B$-model on $\mathbf{C}^3$ rather than the $K3$ compactified theory on $\mathbf{C}^3$. To algebraically translate the computations in this section to the $K3$ case one should include the dependence of the backreaction on the Mukai vector $F \in H^2(Y)$; but the analysis is identical.

## B.1 The weight of the diagram

Let $a = (a_1, a_2)$ denote a pair of non-negative integers. The weight of the diagram in Figure 8 is:

$$\mathcal{W}_{ij}(a) = -\int_{z,w} \tilde{J}^k[a](w) \int_{\mathbf{C}^2 \times \mathbf{C}^3} \mu_{BR}(x)\mu_i(z,x)\mathbf{P}(X,Y)\mu_j(Y)D_{a_1,a_2}\mathbf{P}(Y,W)\,, \qquad (\text{B.1})$$

where

$$\mu_{BR}(x) = \left(\frac{1}{2\pi\|x\|^4}\right)\epsilon_{ij}\overline{x}^i d\overline{x}^j \partial_{x^0}\,, \qquad \mathbf{P}(X,Y) = \left(\frac{3}{4\pi^2\|Z\|^8}\right)\epsilon_{ijk}\epsilon_{lmn}\overline{Z}^i\overline{Z}^l d\overline{Z}^j d\overline{Z}^k \partial_{Z^m}\partial_{Z^n}\,,$$

where $Z = X - Y$.

Without loss of generality, we can specialize the external legs to be of the form:

$$\mu_i(z,x) = f(z,x)d\overline{z}\partial_{x^i}\,, \qquad \mu_j(Y) = g(y)d\overline{y}^0\partial_{y^j}\,.$$

To integrate, we need to keep only the terms in the weight that are expressions of the form:

$$\mathcal{W}_i = h(z,x;Y,w)\partial_w \partial_z \partial_x^2 \partial_Y^3 d\overline{z}d\overline{w}d^2\overline{x}d^3\overline{Y}\,. \qquad (\text{B.2})$$

Note that we use the CY form to turn this into a Dolbeault form of type $(7,7)$ on $\mathbf{C}_w \times \mathbf{C}^3_{z,x} \times \mathbf{C}^3_Y$.

Let us first saturate the polyvector field degree, by expanding the numerators of the propagator and backreaction contributions and then isolating the parts of the weight diagram proportional to precisely $\partial_w \partial_z \partial_x^2 \partial_Y^3$.

---

[27]This is equivalent to demanding the correct holomorphic form degree, since polyvector fields can be traded for differential forms using the Calabi-Yau holomorphic volume form, as described in the main text. It turns out to be simpler to instead perform the count directly with the polyvector fields in terms of which we express the propagator.

Note that $\tilde{J}^k(w)dw$ is part of the integrand of any bulk-defect coupling, although we have often left the holomorphic volume form implicit in the main text. For the purposes of holomorphic polyvector counting (i.e. instead of using holomorphic differential forms), the insertion of the current should be thought of as contributing a factor of $\partial_w$. In addition, the coupling of $\tilde{J}^k(w)$ to the propagator $P(Y, W)$ will force us to keep only the $\partial_{(Y-W)^k}$ component of the propagator. This is because in components we have the contraction $\tilde{J}^k P_{kj}$, with $k$ summed over; to keep the notation from being too laden, we have not decomposed the propagator into components in the weight, but will keep this in mind in what follows.

The schematic form of the diagram in Figure 8 allows for various choices of the Kodaira-Spencer fields $\mu^i$ on the external legs. There are four distinct cases to consider, depending on the values of $i$ and $j$.

Case 1: $i = j = 1$

$$\textcircled{1} = \partial_z \partial_{x^1}\left( \epsilon_{j_1 j_2 j_3}(\overline{X}-\overline{Y})^{j_1} \partial_{(X-Y)^{j_2}} \partial_{(X-Y)^{j_3}} \right)\partial_{y^1}\left( \epsilon_{k_1 k_2 k}(\overline{Y}-\overline{W})^{k_1} \partial_{(Y-W)^{k_2}} \right)$$
$$= \partial_z \partial_{x^1}\left( 2\epsilon_{j_1 j_2 2}(\overline{X}-\overline{Y})^{j_1} \partial_{(X-Y)^{j_2}} \partial_{x^2} \right)\partial_{y^1}\left( \epsilon_{k_1 k_2 k}(\overline{Y}-\overline{W})^{k_1} \partial_{(Y-W)^{k_2}} \right)$$
$$= -4\delta_{k,1}(\overline{y}^0-\overline{w})(\overline{x}^1-\overline{y}^1)\partial_z \partial_x^2 \partial_Y^3 .$$

Case 2: $i = j = 2$

$$\textcircled{2} = \partial_z \partial_{x^2}\left( \epsilon_{j_1 j_2 j_3}(\overline{X}-\overline{Y})^{j_1} \partial_{(X-Y)^{j_2}} \partial_{(X-Y)^{j_3}} \right)\partial_{y^2}\left( \epsilon_{k_1 k_2 k}(\overline{Y}-\overline{W})^{k_1} \partial_{(Y-W)^{k_2}} \right)$$
$$= \partial_z \partial_{x^2}\left( 2\epsilon_{j_1 j_2 1}(\overline{X}-\overline{Y})^{j_1} \partial_{(X-Y)^{j_2}} \partial_{x^1} \right)\partial_{y^2}\left( \epsilon_{k_1 k_2 k}(\overline{Y}-\overline{W})^{k_1} \partial_{(Y-W)^{k_2}} \right)$$
$$= -4\delta_{k,2}(\overline{y}^0-\overline{w})(\overline{x}^2-\overline{y}^2)\partial_z \partial_x^2 \partial_Y^3 .$$

Case 3: $i = 1, j = 2$

$$\textcircled{3} = \partial_z \partial_{x^1}\left( \epsilon_{j_1 j_2 j_3}(\overline{X}-\overline{Y})^{j_1} \partial_{(X-Y)^{j_2}} \partial_{(X-Y)^{j_3}} \right)\partial_{y^2}\left( \epsilon_{k_1 k_2 k}(\overline{Y}-\overline{W})^{k_1} \partial_{(Y-W)^{k_2}} \right)$$
$$= \partial_z \partial_{x^1}\left( 2\epsilon_{j_1 j_2 2}(\overline{X}-\overline{Y})^{j_1} \partial_{(X-Y)^{j_2}} \partial_{x^2} \right)\partial_{y^2}\left( \epsilon_{k_1 k_2 k}(\overline{Y}-\overline{W})^{k_1} \partial_{(Y-W)^{k_2}} \right)$$
$$= 4\partial_z \partial_{x^1}\left( -(\overline{x}^1-\overline{y}^1)\partial_{y^0} + (\overline{z}-\overline{y}^0)\partial_{y^1} \right)\partial_{x^2}\partial_{y^2}\left( \delta_{k,2}(\overline{y}^0-\overline{w})\partial_{y^1} - \epsilon_{k_1 k}\overline{y}^{k_1}\partial_{y^0} \right)$$
$$= 4\left( -\delta_{k,2}(\overline{y}^0-\overline{w})(\overline{x}^1-\overline{y}^1) + \epsilon_{lk}(\overline{z}-\overline{y}^0)\overline{y}^l \right)\partial_z \partial_x^2 \partial_Y^3 .$$

Case 4: $i = 2, j = 1$

$$\textcircled{4} = \partial_z \partial_{x^2}\left( \epsilon_{j_1 j_2 j_3}(\overline{X}-\overline{Y})^{j_1} \partial_{(X-Y)^{j_2}} \partial_{(X-Y)^{j_3}} \right)\partial_{y^1}\left( \epsilon_{k_1 k_2 k}(\overline{Y}-\overline{W})^{k_1} \partial_{(Y-W)^{k_2}} \right)$$
$$= \partial_z \partial_{x^2}\left( 2\epsilon_{j_1 j_2 1}(\overline{X}-\overline{Y})^{j_1} \partial_{(X-Y)^{j_2}} \partial_{x^1} \right)\partial_{y^1}\left( \epsilon_{k_1 k_2 k}(\overline{Y}-\overline{W})^{k_1} \partial_{(Y-W)^{k_2}} \right)$$
$$= 4\partial_z \partial_{x^2}\left( (\overline{x}^2-\overline{y}^2)\partial_{y^0} - (\overline{z}-\overline{y}^0)\partial_{y^2} \right)\partial_{x^1}\partial_{y^1}\left( -\delta_{k,1}(\overline{y}^0-\overline{w})\partial_{y^2} - \epsilon_{k_1 k}\overline{y}^{k_1}\partial_{y^0} \right)$$
$$= 4\left( -\delta_{k,1}(\overline{y}^0-\overline{w})(\overline{x}^2-\overline{y}^2) - \epsilon_{lk}(\overline{z}-\overline{y}^0)\overline{y}^l \right)\partial_z \partial_x^2 \partial_Y^3 .$$

We will presently evaluate the integrals for all of these combinations.

Next, we must saturate the antiholomorphic form degree. Happily, that is much simpler, and does not depend on the values of $i, j$.

$$
\begin{aligned}
\text{⑤} &= \left(\epsilon_{i_1 i_2}\overline{x}^{i_1}d\overline{x}^{i_2}\right)d\overline{z}\left(\epsilon_{j_1 j_2 j_3}(\overline{X}-\overline{Y})^{j_1}d(\overline{X}-\overline{Y})^{j_2,j_3}\right)d\overline{y}^0\left(\epsilon_{k_1 k_2 k_3}(\overline{Y}-\overline{W})^{k_1}d(\overline{Y}-\overline{W})^{k_2,k_3}\right)\\
&= \left(\epsilon_{i_1 i_2}\overline{x}^{i_1}d\overline{x}^{i_2}\right)d\overline{z}\left(2(\overline{z}-\overline{y}^0)d(\overline{x}^1-\overline{y}^1)d(\overline{x}^2-\overline{y}^2)\right)d\overline{y}^0\left(2\epsilon_{k_1 k_2}\overline{y}^{k_1}d\overline{y}^{k_2}d\overline{w}\right)\\
&= -4[\overline{x},\overline{y}](\overline{z}-\overline{y}^0)d\overline{z}d\overline{w}d^2\overline{x}d^3\overline{Y}\,.
\end{aligned}
$$

Putting it all together, we find that eq.(B.1) reduces to:

$$
\mathcal{W}_{ij}(a) = -\int_{z,w}\tilde{J}^k[a](w)\left(\frac{1}{2\pi}\right)^5\frac{3^2 4^2(3+a_1+a_2)!}{3!a_1!a_2!}\left(\delta_{i,j}\delta_{k,i}\Lambda_i + |\epsilon_{ij}|\delta_{k,j}\Lambda_i - \epsilon_{ij}\epsilon_{lk}\Phi_l\right),\quad\text{(B.3)}
$$

where $\Lambda_i$ and $\Phi_l$ are defined as follows:

$$
\Lambda_i = \int_{\mathbf{C}^2\times\mathbf{C}^3}\frac{[\overline{x},\overline{y}](\overline{z}-\overline{y}^0)(\overline{y}^0-\overline{w})(\overline{x}^i-\overline{y}^i)f(z,x)g(y)(\overline{y}^1)^{a_1}(\overline{y}^2)^{a_2}}{(||x||^2)^2(||X-Y||^2)^4(||Y-W||^2)^{4+a_1+a_2}}d^4xd^6Y\,,\quad\text{(B.4)}
$$

$$
\Phi_l = \int_{\mathbf{C}^2\times\mathbf{C}^3}\frac{[\overline{x},\overline{y}](\overline{z}-\overline{y}^0)^2\overline{y}^l f(z,x)g(y)(\overline{y}^1)^{a_1}(\overline{y}^2)^{a_2}}{(||x||^2)^2(||X-Y||^2)^4(||Y-W||^2)^{4+a_1+a_2}}d^4xd^6Y\,.\quad\text{(B.5)}
$$

We will next specialize to the test functions $f(z,x)=z^{k_0}(x^1)^{k_1}(x^2)^{k_2}$ and $g(y)=(y^1)^{m_1}(y^2)^{m_2}$. One can also have additional $(y^0)$ dependence, so that test functions which include $(z)^q(y^0)^p$ with $q+p=n$ allows us to access $n+1$ order poles, but all poles beyond second order vanish for scaling reasons; the single pole coming from $q=p=0$ is the usual $\frac{1}{z}\partial J$ term, with half of the coefficient of the double pole, which is easily fixed by symmetry (and at tree-level was already computed explicitly in §5). Therefore, we will focus on these test functions which give us the leading pole.

We will now perform the integrals.

## B.2 Performing the integrals

Both terms in equation B.3 can be computed in the same way, so we will only present the explicit integration of $\Lambda_i$ and then state the result for $\Phi_l$.

Suppose that we are interested in the OPE $\tilde{J}^i[k]\tilde{J}^j[m]$.

$$
\Lambda_i = (z)^{k_0}\int_x\frac{(x^1)^{k_1}(x^2)^{k_2}}{(||x||^2)^2}\int_Y\frac{[\overline{x},\overline{y}](\overline{z}-\overline{y}^0)(\overline{y}^0-\overline{w})(\overline{x}^i-\overline{y}^i)(y^1)^{m_1}(y^2)^{m_2}(\overline{y}^1)^{a_1}(\overline{y}^2)^{a_2}}{(||X-Y||^2)^4(||Y-W||^2)^{4+a_1+a_2}}\,.
$$

For cleanliness, we will introduce the notation $\tau_y, \tau_x$ to denote the portions of $\Lambda_i$ participating in the $Y, x$ integrals, respectively. We first use Feynman's trick,

$$
\begin{aligned}
\tau_y = &\left(\frac{\Gamma(8+a_1+a_2)}{\Gamma(4)\Gamma(4+a_1+a_2)}\right)\int_0^1 dt\, t^3(1-t)^{3+a_1+a_2}\\
&\times\int_Y\frac{[\overline{x},\overline{y}](\overline{z}-\overline{y}^0)(\overline{y}^0-\overline{w})(\overline{x}^i-\overline{y}^i)(y^1)^{m_1}(y^2)^{m_2}(\overline{y}^1)^{a_1}(\overline{y}^2)^{a_2}}{(t||X-Y||^2+(1-t)||Y-W||^2)^{8+a_1+a_2}}\,.
\end{aligned}
$$

We then shift the integration variable $Y \rightarrow Y + tX + (1-t)W$ and impose $U(1)_{y^0}$ equivariance,

$$
\tau_y = \left( \frac{\Gamma(8 + a_1 + a_2)}{\Gamma(4)\Gamma(4 + a_1 + a_2)} \right)(\overline{z} - \overline{w})^2 \int_0^1 dt \, t^4 (1-t)^{4+a_1+a_2}
$$
$$
\times \int_Y \frac{[\overline{x}, \overline{y}]((1-t)\overline{x}^i - \overline{y}^i)(tx^1 + y^1)^{m_1}(tx^2 + y^2)^{m_2}(t\overline{x}^1 + \overline{y}^1)^{a_1}(t\overline{x}^2 + \overline{y}^2)^{a_2}}{(||Y||^2 + t(1-t)||X-W||^2)^{8+a_1+a_2}} \, .
$$

We use the binomial theorem and then impose $U(1)_{y^i}$ equivariance,

$$
\tau_y = \left( \frac{\Gamma(8 + a_1 + a_2)}{\Gamma(4)\Gamma(4 + a_1 + a_2)} \right)(\overline{z} - \overline{w})^2 \sum_{p_n}^{a_n} \binom{a_n}{p_n} \sum_{q_n}^{m_n} \binom{m_n}{q_n}(x^1)^{m_1 - q_1}(x^2)^{m_2 - q_2}(\overline{x}^1)^{a_1 - p_1}(\overline{x}^2)^{a_2 - p_2}
$$
$$
\times \left( \overline{x}^1 \overline{x}^i \delta_{p_1, q_1} \delta_{p_2+1, q_2} - \overline{x}^i \overline{x}^2 \delta_{p_1+1, q_1} \delta_{p_2, q_2} + (-1)^i \overline{x}^i \delta_{p_1+1, q_1} \delta_{p_2+1, q_2} + \epsilon_{i i_1} \overline{x}^{i_1} \delta_{p_1 + u_1(i), q_1} \delta_{p_2 + u_2(i), q_2} \right)
$$
$$
\times \int_0^1 dt \, t^{4 + a_1 + a_2 + m_1 + m_2 - p_1 - p_2 - q_1 - q_2}(1-t)^{6 + a_1 + a_2 - (q_1 - p_1) - (q_2 - p_2)}
$$
$$
\times \int_Y \frac{(|y^1|^2)^{q_1}(|y^2|^2)^{q_2}}{(||Y||^2 + t(1-t)||X-W||^2)^{8 + a_1 + a_2}} \, ,
$$

where $u(i) = 2(\delta_{i,1}, \delta_{i,2})$.

We introduce radial coordinates $r^i = \frac{|y^i|^2}{t(1-t)||X-W||^2}$ and perform the angular integration,

$$
\tau_y = \left( \frac{\Gamma(8 + a_1 + a_2)}{\Gamma(4)\Gamma(4 + a_1 + a_2)} \right)(\overline{z} - \overline{w})^2 \sum_{p_n}^{a_n} \binom{a_n}{p_n} \sum_{q_n}^{m_n} \binom{m_n}{q_n}(x^1)^{m_1 - q_1}(x^2)^{m_2 - q_2}(\overline{x}^1)^{a_1 - p_1}(\overline{x}^2)^{a_2 - p_2}
$$
$$
\times \frac{\left( \overline{x}^1 \overline{x}^i \delta_{p_1, q_1} \delta_{p_2+1, q_2} - \overline{x}^i \overline{x}^2 \delta_{p_1+1, q_1} \delta_{p_2, q_2} + (-1)^i \overline{x}^i \delta_{p_1+1, q_1} \delta_{p_2+1, q_2} + \epsilon_{i i_1} \overline{x}^{i_1} \delta_{p_1 + u_1(i), q_1} \delta_{p_2 + u_2(i), q_2} \right)}{(||X-W||^2)^{5 + a_1 + a_2 - q_1 - q_2}}
$$
$$
\times (-2\pi i)^3 \int_0^1 dt \, t^{m_1 + m_2 - p_1 - p_2 - 1}(1-t)^{1 + p_1 + p_2} \int_0^\infty \frac{(r^1)^{q_1}(r^2)^{q_2}}{(r^0 + r^1 + r^2 + 1)^{8 + a_1 + a_2}} dr^0 dr^1 dr^2 \, .
$$

We integrate over the radial coordinates and over t to obtain

$$
\tau_y = \left( \frac{(-2\pi i)^3)}{\Gamma(4)\Gamma(4 + a_1 + a_2)} \right)(\overline{z} - \overline{w})^2 \sum_{p_n}^{a_n} \binom{a_n}{p_n} \sum_{q_n}^{m_n} \binom{m_n}{q_n}(x^1)^{m_1 - q_1}(x^2)^{m_2 - q_2}(\overline{x}^1)^{a_1 - p_1}(\overline{x}^2)^{a_2 - p_2}
$$
$$
\times \frac{\left( \overline{x}^1 \overline{x}^i \delta_{p_1, q_1} \delta_{p_2+1, q_2} - \overline{x}^i \overline{x}^2 \delta_{p_1+1, q_1} \delta_{p_2, q_2} + (-1)^i \overline{x}^i \delta_{p_1+1, q_1} \delta_{p_2+1, q_2} + \epsilon_{i i_1} \overline{x}^{i_1} \delta_{p_1 + u_1(i), q_1} \delta_{p_2 + u_2(i), q_2} \right)}{(||X-W||^2)^{5 + a_1 + a_2 - q_1 - q_2}}
$$
$$
\times \left( \frac{\Gamma(m_1 + m_2 - p_1 - p_2)\Gamma(2 + p_1 + p_2)\Gamma(1 + q_1)\Gamma(1 + q_2)\Gamma(5 + a_1 + a_2 - q_1 - q_2)}{\Gamma(2 + m_1 + m_2)} \right).
$$

We now integrate over $d^4 x$,

$$
\tau_x = \int_x \frac{(x^1)^{k_1}(x^2)^{k_2}}{(||x||^2)^2(||X-W||^2)^{5 + a_1 + a_2 - q_1 - q_2}}
$$
$$
\times \left( \overline{x}^1 \overline{x}^i \delta_{p_1, q_1} \delta_{p_2+1, q_2} - \overline{x}^i \overline{x}^2 \delta_{p_1+1, q_1} \delta_{p_2, q_2} + (-1)^i \overline{x}^i \delta_{p_1+1, q_1} \delta_{p_2+1, q_2} + \epsilon_{i i_1} \overline{x}^{i_1} \delta_{p_1 + u_1(i), q_1} \delta_{p_2 + u_2(i), q_2} \right).
$$

Using Feynman's trick and imposing $U(1)_x$ equivariance,

$$\tau_x = \delta^2_{k_t+m_t,1+a_t+v_t(i)}\left(\delta_{p_1,q_1}\delta_{p_2+1,q_2}-\delta_{p_1+1,q_1}\delta_{p_2,q_2}+(-1)^i\delta_{p_1+1,q_1}\delta_{p_2+1,q_2}-(-1)^i\delta_{p_1+u_1(i),q_1}\delta_{p_2+u_2(i),q_2}\right)$$

$$\times\left(\frac{\Gamma(7+a_1+a_2-q_1-q_2)}{\Gamma(5+a_1+a_2-q_1-q_2)}\right)\int_0^1 ds\, s(1-s)^{4+a_1+a_2-q_1-q_2}\int_x \frac{(|x^1|^2)^{k_1+m_1-q_1}(|x^2|^2)^{k_2+m_2-q_2}}{(|x|^2+(1-s)|z-w|^2)^{7+a_1+a_2-q_1-q_2}},$$

where $v(t)=(\delta_{i,1},\delta_{i,2})$.

We introduce radial coordinates $r^i=\frac{|x^i|^2}{(1-s)|z-w|^2}$ and perform the angular integration,

$$\tau_x = \left(\frac{(-2\pi i)}{|z-w|}\right)^2\delta^2_{k_t+m_t,1+a_t+v_t}\left(\delta_{p_1,q_1}\delta_{p_2+1,q_2}-\delta_{p_1+1,q_1}\delta_{p_2,q_2}+(-1)^i\delta_{p_1+1,q_1}\delta_{p_2+1,q_2}-(-1)^i\delta_{p_r+u_r(i),q_r}\right)$$

$$\times\left(\frac{\Gamma(7+a_1+a_2-q_1-q_2)}{\Gamma(5+a_1+a_2-q_1-q_2)}\right)\int_0^1 ds\, s(1-s)^{2+a_1+a_2-q_1-q_2}\int_0^\infty \frac{(r^1)^{k_1+m_1-q_1}(r^2)^{k_2+m_2-q_2}}{(r^1+r^2+1)^{7+a_1+a_2-q_1-q_2}}.$$

We integrate over the radial coordinates and over t to obtain

$$\tau_x = \left(\frac{(-2\pi i)}{|z-w|}\right)^2\delta^2_{k_t+m_t,1+a_t+v_t(i)}\left(\delta_{p_1,q_1}\delta_{p_2+1,q_2}-\delta_{p_1+1,q_1}\delta_{p_2,q_2}+(-1)^i\delta_{p_1+1,q_1}\delta_{p_2+1,q_2}-(-1)^i\delta_{p_r+u_r(i),q_r}\right)$$

$$\times\left(\frac{\Gamma(3+a_1+a_2-q_1-q_2)\Gamma(1+k_1+m_1-q_1)\Gamma(1+k_2+m_2-q_2)}{\Gamma(5+a_1+a_2-q_1-q_2)^2}\right).$$

Putting it all together, we find the following expression for $\Lambda_i$

$$\Lambda_i = \left(\frac{(-2\pi i)^5}{\Gamma(4)\Gamma(4+a_1+a_2)}\right)\left(\frac{1}{(z-w)^2}\right)\delta^2_{k_t+m_t,1+a_t+v_t(i)}z^{k_0}\sum_{p_n}^{a_n}\binom{a_n}{p_n}\sum_{q_n}^{m_n}\binom{m_n}{q_n}q_1!q_2! \tag{B.6}$$

$$\times\left(\delta_{p_1,q_1}\delta_{p_2+1,q_2}-\delta_{p_1+1,q_1}\delta_{p_2,q_2}+(-1)^i\delta_{p_1+1,q_1}\delta_{p_2+1,q_2}-(-1)^i\delta_{p_r+u_r(i),q_r}\right)$$

$$\times\left(\frac{(m_1+m_2-1-p_1-p_2)!(1+p_1+p_2)!(2+a_1+a_2-q_1-q_2)!(k_1+m_1-q_1)!(k_2+m_2-q_2)!}{(1+m_1+m_2)!(4+a_1+a_2-q_1-q_2)!}\right)$$

$$\equiv\left(\frac{(-2\pi i)^5}{\Gamma(4)\Gamma(4+a_1+a_2)}\right)\left(\frac{1}{(z-w)^2}\right)\delta^2_{k_t+m_t,1+a_t+v_t(i)}z^{k_0}\gamma_i^a(k,m),$$

where we have defined $\gamma_i^a(k,m)$ for notational convenience.

By completely identical methods, we also obtain the following expression for $\Phi_l$

$$\Phi_l = \left(\frac{(-2\pi i)^5}{\Gamma(4)\Gamma(4+a_1+a_2)}\right)\left(\frac{1}{(z-w)^2}\right)\delta^2_{k_t+m_t,1+a_t+v_t(i)}z^{k_0}\sum_{p_n}^{a_n}\binom{a_n}{p_n}\sum_{q_n}^{m_n}\binom{m_n}{q_n}q_1!q_2! \tag{B.7}$$

$$\times\left(\delta_{p_1,q_1}\delta_{p_2+1,q_2}-\delta_{p_1+1,q_1}\delta_{p_2,q_2}-(-1)^l\delta_{p_1+1,q_1}\delta_{p_2+1,q_2}+(-1)^l\delta_{p_r+u_r(i),q_r}\right)$$

$$\times\left(\frac{(m_1+m_2-p_1-p_2)!(q_1+q_2)!(2+a_1+a_2-q_1-q_2)!(k_1+m_1-q_1)!(k_2+m_2-q_2)!}{(1+m_1+m_2)!(4+a_1+a_2-q_1-q_2)!}\right)$$

$$\equiv\left(\frac{(-2\pi i)^5}{\Gamma(4)\Gamma(4+a_1+a_2)}\right)\left(\frac{1}{(z-w)^2}\right)\delta^2_{k_t+m_t,1+a_t+v_t(i)}z^{k_0}\beta_l^a(k,m),$$

where again we have defined $\beta_i^a(k,m)$ for notational convenience.

We thus find that $\mathcal{W}_{ij}(a)$ is equal to

$$\mathcal{W}_{ij}(a) = 4i\int_{z,w}\tilde{J}^k[a](w)\left(\frac{z^{k_0}}{(z-w)^2}\right)\left(\frac{1}{a_1!a_2!}\right)\left((\delta_{i,j}\delta_{k,i}+|\epsilon_{ij}|\delta_{k,j})\delta^2_{a_t,k_t+m_t-1-v_t(i)}\gamma_i^a(k,m)\right.$$

$$\left.-\epsilon_{ij}\epsilon_{lk}\delta^2_{a_t,k_t+m_t-1-v_t(l)}\beta_l^a(k,m)\right). \tag{B.8}$$

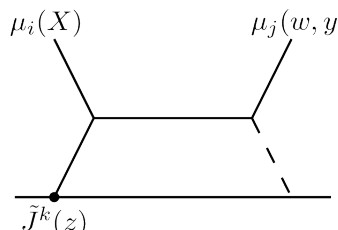

Figure 9

Recall that the important part of the BRST variation of this diagram (to cancel the total BRST variation for Koszul duality) is just the replacement $\mu_i \to \bar{\partial} c_i$ (where the antiholomorphic derivative is along the brane), so that after integration by parts we must perform a contour integral in the defect plane centered at $|z-w|=0$ to extract the OPE from Koszul duality. Since the weight of this diagram produced a double-order pole, we can now fix $k_0 = 1$ to obtain the OPE from the remaining contour integral, which we will see shortly.

## B.3 The second diagram

The OPEs we are interested in also receive contributions from the diagram in Figure 9, of the same topology as Figure 8 but with the other ordering of bulk-defect legs.

The weight of this diagram is

$$\mathcal{W}'_{ij}(a) = \int_{z,w} \tilde{J}^k[a](z) \int_{\mathbf{C}^3 \times \mathbf{C}^2} D_{a_1,a_2} \mathbf{P}(Z,X) \mu_i(X) \mathbf{P}(X,Y) \mu_j(w,y) \mu_{BR}(y). \tag{B.9}$$

As before, we should take the $\tilde{J}^k(z)$ to be implicitly accompanied by $\partial_z$, and we must keep only the $\partial_{(Z-X)^k}$ component of the propagator $P(Z,X)$.

Moving the terms around, this becomes:

$$\mathcal{W}'_{ij}(a) = \int_{z,w} \tilde{J}^k[a](z) \int_{\mathbf{C}^3 \times \mathbf{C}^2} \mu_{BR}(y) \mu_j(w,y) \mathbf{P}(Y,X) \mu_i(X) D_{a_1,a_2} \mathbf{P}(X,Z). \tag{B.10}$$

Relabeling $X \leftrightarrow Y$, and $z \leftrightarrow w$, we find the following equality

$$\mathcal{W}'_{ij}(a) = -\mathcal{W}_{ji}(a). \tag{B.11}$$

Supposed that we are interested in the OPE $\tilde{J}^i[k]\tilde{J}^j[m]$. We specialize the external legs to be of the form:

$$\mu_j(z,x) = z(x^1)^{m_1}(x^2)^{m_2} d\bar{z} \partial_{x^j}, \qquad \mu_i(Y) = (y^1)^{k_1}(y^2)^{k_2} d\bar{y}^0 \partial_{y^i}.$$

Using eq.(B.2), we find that the weight of this diagram is given by

$$\mathcal{W}'_{ij}(a) = -4i \int_{z,w} \tilde{J}^k[a](w) \left( \frac{z}{(z-w)^2} \right) \left( \frac{1}{a_1! a_2!} \right) \Big( (\delta_{i,j} \delta_{k,j} + |\epsilon_{ij}| \delta_{k,i}) \delta^2_{a_t, k_t + m_t - 1 - \nu_t(i)} \gamma^a_j(m,k)$$

$$+ \epsilon_{ij} \epsilon_{lk} \delta^2_{a_t, k_t + m_t - 1 - \nu_t(l)} \beta^a_l(m,k) \Big), \tag{B.12}$$

using the same definitions as the previous subsection.

We may now complete the Koszul duality computation of the OPEs from these contributing diagrams by combining all of these contributions to the off-shell OPEs and performing the brane integrals over $z,w$.

## B.4 Off-shell OPE corrections

We can combine the contribution from both diagrams by noting that $z \sim (z-w)$ and $w \sim -(z-w)$ within the following expressions:

$$\left(\frac{1}{2\pi i}\right)\oint_{|z-w|=0}\left(\frac{zh(z)h'(w)}{(z-w)^2}\right)d(z-w) = \operatorname*{Res}_{(z-w)\to 0}\left((z-w)h(z)h'(w)\right), \tag{B.13}$$

$$\left(\frac{1}{2\pi i}\right)\oint_{|z-w|=0}\left(\frac{wh(z)h'(w)}{(z-w)^2}\right)d(z-w) = -\operatorname*{Res}_{(z-w)\to 0}\left((z-w)h(z)h'(w)\right). \tag{B.14}$$

Using this, we find that the following equality must hold

$$\operatorname*{Res}_{(z-w)\to 0}\left((z-w)\tilde{J}^i[k](z)\tilde{J}^j[m](w)\right) \cong -\left(\frac{1}{2\pi i}\right)\oint_{|z-w|=0}\left(\mathcal{W}_{ij}(a)-\mathcal{W}'_{ij}(a)\right)d(z-w). \tag{B.15}$$

We thus find that the off-shell OPEs are corrected as follows:

$$\begin{aligned}
\tilde{J}^i[k](z)\tilde{J}^j[m](w) \sim -\left(\frac{4i}{(z-w)^2}\right)&\left(\frac{1}{a_1!a_2!}\right)\Big((\delta_{i,j}\delta_{k,i}+|\epsilon_{ij}|\delta_{k,j})\delta^2_{a_t,k_t+m_t-1-\nu_t(i)}\gamma^a_i(k,m)\\
&-\epsilon_{ij}\epsilon_{lk}\delta^2_{a_t,k_t+m_t-1-\nu_t(l)}\beta^a_l(k,m)\\
&+(\delta_{i,j}\delta_{k,j}+|\epsilon_{ij}|\delta_{k,i})\delta^2_{a_t,k_t+m_t-1-\nu_t(i)}\gamma^a_j(m,k)\\
&+\epsilon_{ij}\epsilon_{lk}\delta^2_{a_t,k_t+m_t-1-\nu_t(l)}\beta^a_l(m,k)\Big)\tilde{J}^k[a](w). \tag{B.16}
\end{aligned}$$

Plugging in the four possible $i,j$ combinations, we find that the corrected off-shell OPEs are:

$$\begin{aligned}
\tilde{J}^1[k]\tilde{J}^1[m] \sim -&\left(\frac{4i}{z^2}\right)\left(\frac{1}{(k_1+m_1-2)!(k_2+m_2-1)!}\right)\\
&\times\left(\gamma_1^{(k_1+m_1-2,k_2+m_2-1)}(k,m)++\gamma_1^{(k_1+m_1-2,k_2+m_2-1)}(m,k)\right)\\
&\times\tilde{J}^1[k_1+m_1-2,k_2+m_2-1],\\
\tilde{J}^2[k]\tilde{J}^2[m] \sim -&\left(\frac{4i}{z^2}\right)\left(\frac{1}{(k_1+m_1-1)!(k_2+m_2-2)!}\right)\\
&\times\left(\gamma_2^{(k_1+m_1-1,k_2+m_2-2)}(k,m)++\gamma_2^{(k_1+m_1-1,k_2+m_2-2)}(m,k)\right)\\
&\times\tilde{J}^2[k_1+m_1-1,k_2+m_2-2],\\
\tilde{J}^1[k]\tilde{J}^2[m] \sim -&\left(\frac{4i}{z^2}\right)\left(\frac{1}{(k_1+m_1-2)!(k_2+m_2-1)!}\right)\\
&\times\left(\gamma_1^{(k_1+m_1-2,k_2+m_2-1)}(k,m)-\beta_1^{(k_1+m_1-2,k_2+m_2-1)}(k,m)+\beta_1^{(k_1+m_1-2,k_2+m_2-1)}(m,k)\right)\\
&\times\tilde{J}^2[k_1+m_1-2,k_2+m_2-1]\\
-&\left(\frac{4i}{z^2}\right)\left(\frac{1}{(k_1+m_1-1)!(k_2+m_2-2)!}\right)\\
&\times\left(\gamma_2^{(k_1+m_1-1,k_2+m_2-2)}(m,k)-\beta_2^{(k_1+m_1-1,k_2+m_2-2)}(m,k)+\beta_2^{(k_1+m_1-1,k_2+m_2-2)}(k,m)\right)\\
&\times\tilde{J}^1[k_1+m_1-1,k_2+m_2-2].
\end{aligned}$$

## B.5 On-shell OPE corrections

We can finally use our results from equations (B.17) to obtain the on-shell OPE corrections. For simplicity, we will only pass to on-shell configurations on the left-hand side of the OPE.

It is a straightforward algebraic exercises to express the right hand sides in terms of on-shell generators as well, and in §6 we will do this in some particularly nice examples to see closure of the on-shell algebra explicitly. To proceed, we use the following equality:

$$J[k]J[m] = k_1 m_1 \tilde{J}^2[k_1-1,k_2]\tilde{J}^2[m_1-1,m_2] + k_2 m_2 \tilde{J}^1[k_1,k_2-1]\tilde{J}^1[m_1,m_2-1]$$
$$- k_1 m_2 \tilde{J}^2[k_1-1,k_2]\tilde{J}^1[m_1,m_2-1] - k_2 m_1 \tilde{J}^1[k_1,k_2-1]\tilde{J}^2[m_1-1,m_2]. \quad (B.17)$$

Inserting our findings, we finally obtain the desired OPEs

$$\begin{aligned}
J[k]J[m] \sim &-\left(\frac{4i}{z^2}\right)\left(\frac{\delta_{a_1,k_1+m_1-3}\delta_{a_2,k_2+m_2-2}}{(k_1+m_1-3)!(k_2+m_2-2)!}\right)\Bigg\{k_1 m_1\bigg(\gamma_2^{(a)}(k_1-1,k_2;m_1-1,m_2)\\
&+ \gamma_2^{(a)}(m_1-1,m_2;k_1-1,k_2)\bigg) - k_1 m_2\bigg(\gamma_1^{(a)}(m_1,m_2-1;,k_1-1,k_2)\\
&- \beta_1^{(a)}(m_1,m_2-1;,k_1-1,k_2) + \beta_1^{(a)}(k_1-1,k_2;m_1,m_2-1)\bigg)\\
&- k_2 m_1\bigg(\gamma_1^{(a)}(k_1,k_2-1;,m_1-1,m_2) - \beta_1^{(a)}(k_1,k_2-1;,m_1-1,m_2)\\
&+ \beta_1^{(a)}(m_1-1,m_2;k_1,k_2-1)\bigg)\Bigg\}\tilde{J}^2[k_1+m_1-3,k_2+m_2-2]\\
&-\left(\frac{4i}{z^2}\right)\left(\frac{\delta_{a_1,k_1+m_1-2}\delta_{a_2,k_2+m_2-3}}{(k_1+m_1-2)!(k_2+m_2-3)!}\right)\Bigg\{k_2 m_2\bigg(\gamma_1^{(a)}(k_1,k_2-1;m_1,m_2-1)\\
&+ \gamma_1^{(a)}(m_1,m_2-1;k_1,k_2-1)\bigg) - k_1 m_2\bigg(\gamma_2^{(a)}(k_1-1,k_2;m_1,m_2-1)\\
&- \beta_2^{(a)}(k_1-1,k_2;m_1,m_2-1) + \beta_2^{(a)}(m_1,m_2-1;k_1-1,k_2)\bigg)\\
&- k_2 m_1\bigg(\gamma_2^{(a)}(m_1-1,m_2;k_1,k_2-1) - \beta_2^{(a)}(m_1-1,m_2;k_1,k_2-1)\\
&+ \beta_2^{(a)}(k_1,k_2-1;m_1-1,m_2)\bigg)\Bigg\}\tilde{J}^1[k_1+m_1-2,k_2+m_2-3]. \quad (B.18)
\end{aligned}$$

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
