# Peer review of "Twisted holography on AdS$_3 \times S^3 \times K3 $ & the planar chiral algebra"

_SciPost Physics, doi:SciPost Phys. 17, 109 (2024)_

## Round 1 · Referee Report · Anonymous (Referee 1) · 2024-8-3

Strengths

1) The paper is very beautifully written. The authors do a pretty good job of explaining foundational details concerning their work, so, although it is a long and rather technical paper, it is comparatively readable for a paper on this subject.

2) I think what the authors are doing is a valuable contribution to our understanding of the string theory background that they are studying and its intereesting dualities.

Weaknesses

The paper is certainly not an easy read. But given the nature of the topic, I am not sure that the authors could be expected to do better at making it readable, so I am not sure I would really call that a weakness.

Report

I think the paper does meet the acceptance criteria of the journal and I do recommend publication. I have a couple of comments and a question.

Non-trivial comment: In general, I think that there is something that isn't well expressed in the literature on twisted holography. This is what it really means to ``give a VEV to the superghost.'' In quantum field theory in general, what it means to ``give a VEV'' to a field is the following: first, this only makes sense if one is working on a non-compact space with one or more asymptotic regions ``at infinity.'' Assuming this, what it means to ``give a VEV'' to a field is to specify the asymptotic value of the field at infinity; the theory then decides for itself what will happen in the interior. (If there is more than one asymptotic region at infinity, in general one can specify different asymptotic values at different ends.)
What I have written is no problem for this paper, because the authors are working on the noncompact manifold C^3 x K3, but note that if one is on a compact Calabi-Yau fivefold, I do not believe that ``twisted holography'' can be defined as there is no asymptotic region and no way to ``give a VEV'' to anything. I believe the paper could be improved by more accurately saying at the beginning what it means to ``give a VEV.'' But this isn't a comment just on this paper; it is a comment on the whole literarature on twisted holography.

Trivial comment: On p. 21, is eqn. (2.5.11) written correctly? I think the authors intend to have a first order differential operator acting on Phi, but for this they need parentheses. I also was confused about C^{22} at the top of the page. Please note that this isn't intended as a comprehensive list of possible minor misprints.

And a question: What would happen if T^4 or K3 is replaced by a Hopf surface
S^3 x S^1? Note that string theory on AdS_3 x S^3 x S^3 x S^1 is comparatively not well understood at all. Is there something useful to say?

Recommendation

Publish (surpasses expectations and criteria for this Journal; among top 10%)

---

## Round 2 · Referee Report · Anonymous (Referee 2) · 2024-9-29

Strengths

Twisted holography provides one of the few examples of solvable models of AdS/CFT. This paper makes signifcant advances within this framework.

Weaknesses

The paper is very mathematical and difficult to understand for the general audience.

Report

The paper works out the details of twisted holography for the famous D1D5 system. This is a very important contribution to the field. Twisted holography is one of the few exactly solvable examples of AdS/CFT and it is important to pin down all the details. This may well lead to a better understanding of holography and string theory more generally.

Requested changes

None

Recommendation

Publish (surpasses expectations and criteria for this Journal; among top 10%)

---

## Round 2 · List of Changes

We are grateful to the anonymous referee for their report and feedback on the draft.
In view of their comments, we have made the following changes:

1.) A more detailed discussion about the notion of twisting espoused in this paper, particularly twisting supergravity, on page 1.
2.) Added a footnote on page 2 with comments about the case of AdS3xS3XS3xS1.
3.) Corrected the typo pointed out in equation 2.5.11.
4.) Clarified that the appearance of C^{22} comes from H^2(K3), on the same page.

---

## Editorial Decision

published